# New Visions on Natural Products and Cancer Therapy: Autophagy and Related Regulatory Pathways

**DOI:** 10.3390/cancers14235839

**Published:** 2022-11-26

**Authors:** Alma Martelli, Marzieh Omrani, Maryam Zarghooni, Valentina Citi, Simone Brogi, Vincenzo Calderone, Antoni Sureda, Shahrokh Lorzadeh, Simone C. da Silva Rosa, Beniamin Oscar Grabarek, Rafał Staszkiewicz, Marek J. Los, Seyed Fazel Nabavi, Seyed Mohammad Nabavi, Parvaneh Mehrbod, Daniel J. Klionsky, Saeid Ghavami

**Affiliations:** 1Department of Pharmacy, University of Pisa, Via Bonanno 6, 56126 Pisa, Italy; 2Department of Phytochemistry, Medicinal Plants and Drugs Research Institute, Shahid Beheshti University, Tehran 1983969411, Iran; 3Department of Laboratory Medicine & Pathobiology, University of Toronto Alumna, Toronto, ON M5S 3J3, Canada; 4Research Group in Community Nutrition, Oxidative Stress and Health Research Institute of the Balearic Islands (IdISBa), University of Balearic Islands, 07122 Palma de Mallorca, Spain; 5CIBER Physiopathology of Obesity and Nutrition (CIBEROBN), Instituto de Salud Carlos III (ISCIII), 28029 Madrid, Spain; 6Department of Human Anatomy and Cell Science, Max Rady College of Medicine, Rady Faculty of Health Sciences, University of Manitoba, Winnipeg, MB R3T 2N2, Canada; 7Department of Histology, Cytophysiology and Embryology, Faculty of Medicine in Zabrze, Academy of Silesia, 41-800 Zabrze, Poland; 8Department of Gynaecology and Obstetrics, Faculty of Medicine in Zabrze, Academy of Silesia, 41-800 Zabrze, Poland; 9GynCentrum, Laboratory of Molecular Biology and Virology, 40-851 Katowice, Poland; 10Department of Neurosurgery, 5th Military Clinical Hospital with the SP ZOZ Polyclinic in Krakow, 30-901 Krakow, Poland; 11Biotechnology Centre, Silesian University of Technology, 44-100 Gliwice, Poland; 12Nutringredientes Research Center, Federal Institute of Education, Science and Technology (IFCE), Baturite 62760-000, Brazil; 13Advanced Medical Pharma (AMP-Biotec), Biopharmaceutical Innovation Centre, Via Cortenocera, 82030 San Salvatore Telesino, Italy; 14Influenza and Respiratory Viruses Department, Pasteur Institute of Iran, Tehran 1316943551, Iran; 15Life Sciences Institute, University of Michigan, Ann Arbor, MI 48109, USA; 16Faculty of Medicine in Zabrze, Academia of Silesia, 41-800 Zabrze, Poland; 17Research Institute of Oncology and Hematology, Cancer Care Manitoba, University of Manitoba, Winnipeg, MB R3T 2N2, Canada

**Keywords:** apoptosis, autophagy, autophagy-related diseases, cancer, natural compound, unfolded protein response

## Abstract

**Simple Summary:**

In the past few years, clinical as well as regular scientists have been paying attention to medicinal applications of natural products. There are several investigations focusing on the potential beneficial effects of natural compounds in the treatment of different cancers. In the current comprehensive review paper, we first discuss three major mechanisms that control the effect of cancer therapy agents on tumor cells. This includes autophagy, apoptosis and the unfolded protein response. In the next step, we discuss how different natural compounds (more than 100 compounds) target cancer cells via the autophagy pathway and how this regulates stress response and cell death in cancer cells.

**Abstract:**

Macroautophagy (autophagy) has been a highly conserved process throughout evolution and allows cells to degrade aggregated/misfolded proteins, dysfunctional or superfluous organelles and damaged macromolecules, in order to recycle them for biosynthetic and/or energetic purposes to preserve cellular homeostasis and health. Changes in autophagy are indeed correlated with several pathological disorders such as neurodegenerative and cardiovascular diseases, infections, cancer and inflammatory diseases. Conversely, autophagy controls both apoptosis and the unfolded protein response (UPR) in the cells. Therefore, any changes in the autophagy pathway will affect both the UPR and apoptosis. Recent evidence has shown that several natural products can modulate (induce or inhibit) the autophagy pathway. Natural products may target different regulatory components of the autophagy pathway, including specific kinases or phosphatases. In this review, we evaluated ~100 natural compounds and plant species and their impact on different types of cancers via the autophagy pathway. We also discuss the impact of these compounds on the UPR and apoptosis via the autophagy pathway. A multitude of preclinical findings have shown the function of botanicals in regulating cell autophagy and its potential impact on cancer therapy; however, the number of related clinical trials to date remains low. In this regard, further pre-clinical and clinical studies are warranted to better clarify the utility of natural compounds and their modulatory effects on autophagy, as fine-tuning of autophagy could be translated into therapeutic applications for several cancers.

## 1. Introduction

Autophagy, the unfolded protein response (UPR), and apoptosis are tightly interconnected, typically affect each other and have been extensively investigated in many types of cancer [1,2]; treatment protocols for different cancers may be improved by manipulating these processes. Along these lines, the application of plant derivatives in cancer therapy has a longstanding record [3]. Plants serve as an important root of powerful anti-cancer factors. Greater than 60% of currently used antineoplastic factors are derivatives of natural sources, such as plants, marine organisms and micro-organisms [4]. In this review, we discuss over 100 natural compounds and highlight plant species that have been investigated for their beneficial impact on different types of cancer. We specifically emphasize the impacts of these compounds on the autophagy pathway and outline how the interplay of autophagy with the UPR and apoptosis is therapeutically important. First, we briefly discuss the concepts of autophagy, the UPR, apoptosis and their crosstalk, followed by a detailed explanation of how these processes are involved in the potential anti-cancer effects of natural compounds.

## 2. Autophagy

Over 50 years ago, a new term entered the scientific vocabulary—“autophagy” (self-eating), a process in which intracellular and cytoplasmic constituents are delivered to the lysosomes and degraded [5]. Autophagy is a multi-faceted and complicated molecular and cellular process, extremely important for maintaining cellular homeostasis and preserving cells during nutritional, metabolic and pathological stress [6,7,8,9,10]; however, too much autophagy can be detrimental, and this process is therefore also deemed a type of programmed cell death [11,12,13].

Three main types of autophagy, each modulated by specific autophagy-related (ATG) genes and proteins have been identified in mammals: macroautophagy (herein referred to as autophagy), chaperone-mediated autophagy (CMA) and microautophagy (Figure 1) [14,15,16,17].

Macroautophagy consists of several steps, including initiation, phagophore expansion, autophagosome fusion with the lysosome, developing an autolysosome for proteolytic destruction of the cargo and discharge of the disintegrated products back into the cytosol [18,19,20,21]. Although autophagy is usually not a selective process (e.g., autophagy induced by starvation), it becomes selective under some conditions and has potential importance in clinical diseases. Some of the various types of selective autophagy are called aggrephagy (protein aggregates), mitophagy (mitochondria), xenophagy (invasive microbes) and ciliophagy (cilia) based on the specific targets [22,23,24].

During microautophagy, lysosomes take up small cytosolic constituents (soluble or particulate cellular components) by invagination or protrusion of the lysosomal limiting membrane followed by scission [25,26,27,28]. This type of autophagy is non-selective and does not require autophagosome formation. It has been postulated that microautophagy occurs to maintain the correct size and membrane composition of the lysosome [29,30]. Note that there are microautophagy-like processes (i.e., involving direct uptake at the lysosomal limiting membrane) that are highly selective, including micropexophagy.

Finally, CMA is an autophagy mechanism selective for protein degradation. In fact, only individual proteins that have a specific amino acid motif (KFERQ sequence) [31] bind to a chaperone located in the cytosol (HSPA8/HSC70; heat shock protein family A (Hsp70) member 8). These proteins are unfolded and transferred across the lysosomal membrane through LAMP2A (lysosomal membrane protein 2A) in a process involving lumenal HSPA8 and they are subsequently rapidly degraded [32].

Disruption of autophagy has been correlated with several pathologies, such as cancer, cardiomyopathies, neurodegenerative diseases, diabetes, liver diseases, immune disorders and pathogen infections [33,34,35,36,37,38,39,40,41,42,43,44,45]. Thus, targeting the autophagic process could be a valuable strategy for the identification of new therapeutic avenues. Several studies have highlighted the potential of a broad range of natural products and compounds to modulate autophagy, either leading to pro-survival autophagy or autophagic cell death (ACD) [46]. Natural compounds that modulate autophagy have become an important focus of research that aims to address autophagy-related diseases.

## 3. Autophagy Databases

Advances in understanding the autophagy process have allowed the discovery of genes and their corresponding proteins that are involved in autophagy mechanisms. Currently, more than 40 such genes have been identified, of which many are conserved among different organisms including yeasts, worms, slime mold, flies, plants and mammals [47].

The availability of several databases with detailed information on autophagy-related genes and proteins represents a valuable resource for bioinformatics applications. For example, the human autophagy database (HADb) is a public autophagy resource and is a product of the Luxembourg Institute of Health/LIH, which provides inclusive information on the gene of interest and autophagy-related proteins (http://autophagy.lu/index.html, 22 November 2022).

The autophagy database of the National Institute of Genetics (NIG) provides data on proteins involved in autophagy to modernize current autophagy research; it offers a comparison of homologous proteins among more than forty different species, allowing the user to search for new and established autophagy-related proteins (http://www.tanpaku.org/autophagy/overview.html, 22 November 2022) [48].

Autophagy To Disease/ATD (http://auto2disease.nwsuaflmz.com, 22 November 2022) offers an overview of autophagy-associated diseases, providing a bioinformatic annotation system for chemical sorts, human diseases, and genes and proteins related to autophagy.

The Autophagy Regulatory Network (ARN) (http://autophagyregulation.org, 22 November 2022) is a manually curated database of autophagy components and related connections, integrating resources with known autophagy protein regulators. The database contains 1485 proteins with 4013 different interactions. This database includes possible transcriptional and post-transcriptional regulators (i.e., transcription factors and microRNA/miRNA) of autophagy and its regulatory proteins. All these autophagy effectors have been linked to the most important signaling pathways of the SignaLink 2 resource. The ARN is particularly useful for the identification of potentially relevant therapeutic strategies; by using the online resource, accessible through an easy-to-use interface, it is possible to predict novel regulators and interactions [49].

In addition to databases specifically dedicated to autophagy genes and proteins, several other online resources are available to extract relevant information about autophagy modulators. These include the Autophagic Compound Database (ACDB) (http://www.acdbliulab.com/search.php, 22 November 2022), curated with data retrieved from scientific literature on known compounds and their relationships to autophagy. Specifically, the database contains information on over 350 compounds with more than 160 related signaling pathways and potential targets in various disorders [50]. The Autophagy Small Molecule Database (Autophagy SMDB) (http://www.autophagysmdb.org/ 22 November 2022) is a comprehensive, manually curated, cross-platform database that includes compounds and their associated protein targets, capable of behaving as autophagy modulators. Currently, the Autophagy SMDB lists over 10,000 molecules related to 71 cellular targets. For all entries, detailed information regarding the activity, International Union of Pure and Applied Chemistry (IUPAC) name, canonical Simplified Molecular Input Line Entry System (SMILE), structure, molecular weight and calculated physicochemical properties, is available [51]. The Human Autophagy Modulator Database (HAMdb) (http://hamdb.scbdd.com, 22 November 2022) is a manually curated database containing information on 841 chemicals, 132 microRNAs and 796 proteins, related to the particular impacts on autophagy mechanisms, biological and physicochemical properties, and disorder involvement. Moreover, the HAMdb contains several external links to retrieve additional information that covers extensive biomedical knowledge [52].

All these tools can provide researchers with insightful information about autophagy pathways and related diseases necessary to facilitate the discovery of novel autophagy modulators.

## 4. Physiological and Pathophysiological Roles of the Autophagic Machinery in Cancer

Autophagy is one of the main degradation systems, common to all eukaryotic organisms; during this process, parts of the cytoplasm are isolated and released into lysosomes for degradation [9,53,54]. Mammalian autophagy mechanisms are essential for maintaining a healthy cellular state. Autophagy serves a prominent role in cytoplasmic turnover, removing superfluous, damaged and misfolded organelles and proteins and reducing DNA instability that may lead to oncogenic alterations and ultimately cancer [43,55,56,57]. This process also entails recycling cellular components to provide sources of energy or building blocks (amino acids, fatty acids and sugars) for the biosynthesis of macromolecules [58,59,60]. Thus, autophagy is critically vital for the quality control of intracellular organelles and proteins [61].

Depending on the form of stress stimulus, the autophagic machinery can either support cell survival and adaptation (cell protective) or lead to cell death (cell destructive) [62,63]. Autophagy is induced by nutrient deficiency, hypoxia and oxidative stress, involving the activation of different pathways and processes such as the MTOR (mechanistic target of rapamycin kinase) pathway, stress signals from the endoplasmic reticulum (ER), calcium signaling and the insulin pathway [38,64,65].

A wide diversity of nutrients, growth factors and insulin-like signals inhibit autophagy via the MTOR pathway by directly modulating class I phosphoinositide 3-kinase (PI3K)-AKT/protein kinase B signaling [8,15,55,66,67].

An increasing number of studies suggest an intricate association between autophagy and cancer. In particular, the dualistic behavior of autophagy has been linked to tumor type, staging and nature of the stress signal [56,57,68].

Other autophagy modulators have essential roles in the response to different stimuli. For instance, 5′-adenosine monophosphate (AMP)-activated protein kinase (AMPK) is a modulator that responds to low cellular energy; EIF2A/eIF2α (eukaryotic translation initiation factor 2A) responds to starvation, ER stress and accumulation of double-stranded RNA; and the tumor suppressor protein TP53/p53 together with TP53-related proteins are rapidly activated in the presence of DNA damage [69]. Interestingly, TP53 can inhibit or activate autophagy mechanisms depending on various elements such as the nature of DNA damage or its subcellular localization, DAPK (death associated protein kinase) proteins, MAPK/SAPK, MAPK8/JNK1 (mitogen-activated protein kinase 8), ITPR (inositol 1,4,5-trisphosphate receptor), the ER membrane-associated protein ERN1/IRE1 (endoplasmic reticulum to nucleus signaling 1), GTPases, MAPK1/ERK2-MAPK3/ERK1, ceramide and calcium [37,53,70,71].

Considering its broad association with various cellular processes, the modulation (activation or inhibition) of the autophagic machinery represents a potential approach for treating several diseases [17,42,55,62,71,72,73].

Activation of autophagy has been proposed to promote healthy aging by reducing the incidence of cancer, cardiovascular disease, diabetes and brain decline [74]. In this sense, stimulating autophagy has attracted interest in developing innovative curative approaches for neurodegenerative disorders [35]. However, it has been observed that inducing autophagy processes to treat specific disorders may not be devoid of severe adverse effects. For instance, the activation of autophagy can promote the onset of pathological processes such as the reactivation of dormant tumors [75]. Accordingly, with an improved understanding of the autophagic machinery, it will be possible to improve the selectivity of drugs and specifically target altered autophagy pathways [76]. To date, several examples of molecules able to modulate autophagy have been described in the literature and the number of studies on this topic is constantly growing (https://www.ncbi.nlm.nih.gov/pubmed, 22 November 2022).

### 4.1. Autophagy in the Initial Stages of Cancer

In the course of the primary stages of oncogenesis, autophagy is mainly a tumor suppressor mechanism via the “recycling” of damaged organelles or misfolded proteins, which are involved in cancer initiation [77,78]. Therefore, any disruption in the autophagy pathway is potentially correlated with genetic instability, epigenetic changes, malignant transformation and tumorigenesis [79,80]. For example, an autophagy defect resulting from knocking down BECN1 (beclin 1) promotes the formation of cancer in mice [81,82]. Furthermore, *Becn1* loss is frequently observed in different forms of cancer, such as breast, prostate and ovarian cancers, besides BECN1 loss in malignant epithelial ovarian cancer [82,83,84]. BECN1 is involved in autophagy induction and promotes tumor-suppressive functions by regulating UVRAG (UV radiation resistance associated) and SH3GLB1/Bif-1 (SH3 domain containing GRB2 like, endophilin B1) that positively regulates the activity of BECN1 via supporting the interplay between BECN1 and PIK3C3/VPS34, a component of the class III autophagy-specific phosphatidylinositol 3-kinase (PtdIns3K) complex, resulting in increased autophagy [85,86]. Lower expression of SH3GLB1 and UVRAG mutations that induce its inactivation [87] have been identified in different types of cancers including gastric, prostate, bladder and breast cancers [80].

In addition to *Becn1*, many studies revealed that the deletion of several *Atg* genes is associated with oncogenesis. Indeed, deficiency of *Atg5* and *Atg7* boosts the growth of liver tumors in mice due to an imbalance in oxidative stress and mitochondrial damage [88]. Furthermore, mice lacking *Atg4*, which is necessary for the subsequent interaction between MAP1LC3/LC3 (microtubule associated protein 1 light chain 3) and the ATG12–ATG5-ATG16L1 complex and functions as a tumor suppressor, present a higher risk factor with respect to developing chemical-induced fibrosarcoma [89]. Additionally, autophagy is controlled by signaling pathways that regulate tumorigenesis, such as PI3K-AKT [90,91]. For example, AKT overexpression, PI3K mutations or PTEN (phosphatase and tensin homolog) loss impair autophagy mainly via MTOR activation in glioblastoma cells [90,91]. BCL2 (B-cell lymphoma 2), apoptosis regulator, overexpression can potentially regulate autophagy through an inhibitory binding to BECN1 in mammalian cells [92]. These lines of evidence highlight the fact that autophagy is a crucial biological process that inhibits tumor generation, and its dysfunction could promote tumorigenesis.

### 4.2. Autophagy in Advanced Tumors

Although autophagy has a crucial function in the suppression of malignant transformation in the early stages of cancer, several surveys showed that in the later stages of cancer, autophagy activation can have pro-survival effects, sustaining tumor growth and promoting tumor progression [78,93]. Cancer cells are defined by a significant growth measure and increased metabolic stress, leading to a toxic environment in tumor cells, if it is left unresolved [93]. In this context, autophagy is considered a complex means that reacts to different types of stress, including uncontrolled tumor growth, depletion of the blood supply to tumor cells, starvation and hypoxia. Starvation decreases adenosine triphosphate (ATP) levels and stimulates the AMPK pathway and consequently promotes autophagy [94]. Conversely, the hypoxic condition triggers the stimulation of HIF1A/HIF-1α (hypoxia inducible factor 1 subunit alpha)-dependent autophagy [94]. In both of these conditions, autophagy contributes to tumor survival [94].

Autophagy may also be activated by some oncogenic pathways during cancer cell proliferation, supported by a high frequency of KRAS (KRAS proto-oncogene, GTPase) or HRAS (HRas proto-oncogene, GTPase) mutations that promote the development of cancer cells in an autophagy-dependent mode in different cell types, like pancreatic cancer and colorectal cancer (CRC) cells [89]. The induction of the autophagy process maintains tumor cell development and, consequently, the suppression of KRAS and HRAS could slow down the tumor progression and proliferation partially via autophagy inhibition [89]. Different mechanisms that activate autophagy are linked to several hallmarks of cancer. There are some contradictory data sources concerning the supportive function of autophagy in cancer cell proliferation. For example, some autophagy inducers, such as rapamycin and its derivatives, can block MTOR-cascade-dependent cell growth via cell cycle arrest in MDA-MB-231 breast cancer cells [95]. These studies suggest a fragile equilibrium between autophagy-dependent tumor cell proliferation and cell death.

Another cancer hallmark called “angiogenic switch” is characterized by the ability of tumor cells to induce and maintain angiogenesis during tumor development and provide oxygen and nutrients to cancer cells [96]. Indeed, to sustain growth, cancer cells exploit nearby normal cells for biosynthesis and secretion of VEGF (vascular endothelial growth factor) that triggers the activation of signaling pathways, leading to the establishment of novel blood vessels [96]. The elevated autophagy sustains cancer cells and allows them to overcome oxygen stress [96]. Furthermore, autophagy facilitates tumor invasion and metastasis via regulating the epithelial-to-mesenchymal transition and tumor immune surveillance [20,55,97]. Autophagy is also linked to tumor metabolic reprograming; cancer cells switch their metabolism by exploiting aerobic glycolysis to survive [98]. This phenomenon is called the “Warburg effect” and allows the tumor cells to reduce the need for oxygen in ATP production [98]. Even, in hematological/blood cancers, autophagy can either perform as a chemo-resistance mechanism or have tumor suppressive functions, depending on the context [99].

The plasticity of cancer cell metabolism makes limitations for anticancer treatments that could lead to therapy resistance. Evidence suggests an important role of mitophagy in tumor growth, metastasis and therapy resistance depending on tumor type, stage or metabolic activity. Therefore, pharmacological modulation of mitophagy in tumors could be a promising anticancer strategy [100]. Researchers demonstrated that autophagy plays dual roles in drug resistance of gastric cancer [101] and it is activated in response to chemotherapy in neuroblastoma cells that confers chemoresistance [102]. However, the proposition of inhibition or activation of this pathway is still limited in preclinical models and human tumors [100]. The dual function of autophagy in cancer is shown in Figure 2.

Cancer control may not be achieved by targeting only a single cell death program. Each pathway has its own characteristics. For example, apoptosis exhibits cytoplasmic shrinkage, chromatin condensation, nuclear fragmentation, plasma membrane blebbing and formation of apoptotic bodies; however, autophagy shows extensive cytoplasmic vacuolization and, similar to the culminating phagocytic uptake of apoptotic cells, there is consequent lysosomal degradation [103]. The UPR is a stress pathway that can be triggered by the abnormal accumulation of unfolded proteins in the ER caused by genetic or environmental changes, hypoxia or altered glycosylation. The UPR maintains balance in ER homeostasis by lowering the number of unfolded proteins present in the cell [104]. There is interaction/crosstalk among autophagy, apoptosis and the UPR in tumor cells, which determine their fate. Therefore, we summarized them here to provide a better understanding of the functional mechanism of natural compounds on cancer cells.

### 4.3. Functional Crosstalk between Apoptosis and Autophagy in Cancer

Autophagy and apoptosis are interconnected, and their crosstalk has a crucial function in the response of cancer cells to different chemotherapy compounds [57,105,106]. These procedures often happen concurrently in response to a related trigger and mostly in a way that autophagy negatively or positively regulates apoptosis [9,63,107].

There are several proteins, which are engaged in the regulation of switching between autophagy and apoptosis in malignancy [56]. Some links between these two pathways are mediated via TP53-associated mitochondrial proteins and genes including BBC3/PUMA (BCL2 binding component 3) and PMAIP1/NOXA (phorbol-12-myristate-13-acetate-induced protein 1) [69,71]. Low induction of the mitochondrial membrane permeabilization results in autophagy activation [108]. If the permeabilization passes an irreversible threshold level, it will initiate apoptosis induction [108]. Therefore, the activation of apoptotic or accidental cell death (ACD) may depend on the level of mitochondrial membrane permeabilization [108]. Several mediators can simultaneously induce apoptosis and autophagy, including BECN1 and BCL2 interaction through the BCL2 homology 3/BH3 domain, phosphorylation of BECN1 and PRKD/PKD (protein kinase D) via DAPK and phosphorylation of BCL2 via MAPK8/JNK1 with subsequent activation of the BECN1-PIK3C3/VPS34 complex [56,105,108]. A brief overview of the functional relationship between autophagy and apoptosis pathways that are involved in tumor cell response to chemotherapy is shown in Figure 3.

### 4.4. Functional Relationship between Autophagy and the UPR in Cancer

Autophagy and the UPR are also interconnected and play a strong role in the control of tumor cell response to chemotherapy compounds. There are three arms of the UPR, involving ERN1/IRE1, ATF6 (activating transcription factor 6) and EIF2AK3/PERK (eukaryotic translation initiation factor 2 alpha kinase 3), that are significantly engaged in the initiation of autophagy through different mechanisms, whereas autophagy also modulates the ER stress through the removal of protein aggregates and reduction of ER enlargement [109,110,111]. The UPR is mainly activated to ameliorate the misfolded protein aggregate load in the ER [110,112]. In cases of failure of the UPR to restore proper ER homeostasis, cell death is induced, usually by activating the intrinsic apoptosis pathway [113,114]. Moreover, in order to clear the accumulated misfolded proteins, the UPR may upregulate the autophagy machinery [112,115,116].

There is an interplay between EIF2AK3, EIF2A, ATF4 and DDIT3/CHOP/GADD153 (DNA damage inducible transcript 3) and induction of autophagy [109]. The activated EIF2AK3 helps to phosphorylate EIF2A, causing the inhibition of broad protein synthesis [117]. Phosphorylated EIF2A supports the translocation of ATF4 to the nucleus [118]. ATF4 is in charge of the upregulation of ATG12 [118], a crucial element of the ATG12–ATG5-ATG16L1 complex, which is critical for the expansion of phagophores [109]. The EIF2A-ATF4-DDIT3 axis can promote the SQSTM1/p62 (sequestosome 1) expression at the transcriptional level via binding to the *Sqstm1* promoter to regulate autophagy induction [119]. There is also interplay between ATF6 and autophagy induction. The mechanism includes ATF6 interaction with CEBPB and the formation of a transcriptional heterodimer combination that attaches to ATF/CREB components of the *DAPK1* promoter to stimulate the DAPK1 expression. DAPK1 can drive autophagosome formation through BECN1 phosphorylation [120]. The upregulation of DDIT3, XBP1 (X-box binding protein 1) and HSPA5/GRP78/BiP (heat shock protein family A (Hsp70) member 5) mediated by ATF6 also contributes to ATF6-induced autophagy [121]. The UPR and autophagy are also interconnected via ERN1 in tumor cells. ERN1 RNase produces spliced XBP1 (sXBP1), which controls the expression of genes responsible for folding, entry of proteins into the ER, ER-associated degradation/ERAD and biogenesis of the ER and Golgi apparatus [122]. Moreover, MAPK/JNK activation through the ERN1-MAPK/JNK-BECN1 complex initiates autophagy and helps tumor cells to adapt to UPR conditions [116,123]. The functional relationship between the autophagy and UPR pathways is summarized in Figure 4.

### 4.5. Interaction between the Immune System and Autophagy in Cancer Treatment

It was presumed that autophagy inhibition is a potential opportunity to cure cancer, given that autophagy helps in adaptation to stress conditions including chemotherapy compounds [124]. In the immune-competent and immune-deficient mouse experimental models, it was revealed that autophagy inhibition decreases tumor growth in immune-deficient mice, whereas it reduces the effectiveness of chemotherapy in immune-competent mice [125]. Therefore, autophagy inhibition is not necessarily helpful for tumor growth inhibition in the company of an operative immune system and the help of the immune response is vital for thriving neoplastic treatment [126].

Innate immunity activates autophagy through TLRs (toll-like receptors) and NOD-like receptors (NLRs) by natural killer T (NKT) cell activation, cytokine production and phagocytosis [125]. Adaptive immunity affects autophagy by antigen presentation, lymphocyte development, homeostasis and cytokine release [125].

Autophagy can induce or inhibit tumor progression by modulating immune cell homeostasis. Autophagy can promote tumor development through M2-like tumor-associated macrophages (TAMs), myeloid-derived suppressor cells and regulatory T cells (Tregs) [125]. Autophagy controls tumor progression through MTOR inhibition, which activates CD8+ T cell differentiation to generate cytotoxic T lymphocytes (CTLs) but inhibits T cell differentiation into helper T cells [125]. Autophagy activates dendritic cell and B cell development and specific IgM and IgG production via antigen presentation [127]. Autophagy in association with cytosolic phosphorylated FOXO1 (forkhead box O1) and ATG7 contributes to NKT cell development [128]. In addition, autophagy activates ULK1 (unc-51 like autophagy activating kinase 1) and MAPK/JNK, which are essential for macrophage production and inhibition of tumor progression [125]. MAPK14/p38 can block the development of neutrophils via autophagy induction and impair inhibition of tumor development [125]. The functional relationship between autophagy and immune cells, involved in response to chemotherapy is shown in Figure 5.

Autophagy plays a role in immune cells, T-cells, B cells, macrophages, myeloid derived suppressor cells and dendritic cells. It can alter tumor immunity as well as the efficacy of immunotherapy [129]. The tumor microenvironment (TME), an integral complex component of cancer, can significantly influence the therapeutic response and determine tumor fate by its composition and dynamics [130]. There is a complex interaction between autophagy and TME, which modulates immunotherapy. Autophagy-mediated regulation of tumor-associated immunity may counteract or enhance the efficacy of immunotherapy. On the one hand, autophagy can promote immune response by enhancing the inhibitory role of immune cells on tumor cells and the release of cytokines, which leads to the enhanced antitumor immunotherapy responses. On the other hand, it can reduce immune response by immunosuppressive Tregs and cytokines, contributing to attenuated antitumor immunotherapy effects and accelerated tumor development. The lack of sufficient specificity of autophagy activators or inhibitors limits their clinical applications [129].

## 5. Natural Compounds and Cancer

### 5.1. The Effects of Natural Compounds on the Autophagy Signaling Pathway

Natural compounds are approximately the source of 60% of the routinely found anti-cancer factors [131]. Natural product molecules have been suggested to act as autophagy regulators due to their effects on several cellular signaling pathways and transcription factors both in vitro and in vivo [132]. In the following sections, we introduce these compounds and review their effects on the autophagy pathway. We later highlight how the UPR and apoptosis might be affected by natural compounds via the autophagy pathway.

#### 5.1.1. Alkaloids

Alkaloids are noteworthy chemical compositions that act as a major origin for new medicine. Alkaloids contain nitrogen and heterocyclic rings and are found in different plants. Some alkaloids have only recently managed to become established as anti-cancer medications including vinblastine (1), vinorelbine (2), vincristine (3) and vindesine (4). Vinblastine (1), vincristine (3) and vindesine (4) are indol alkaloids isolated from *Catharanthus roseus* [133]. Vincristine (3) and vinblastine (1) are usually utilized in conjunction with other chemotherapeutic factors for the therapy of lymphoma, advanced testicular cancer, leukemia, breast cancer, Kaposi sarcoma and lung cancer [4].

Numerous alkaloids presented activities against human hepatocarcinoma cell line (HepG2) and human colon cancer cell line (LS174T) by modulating autophagy. Vinblastine (1), which was reported to inhibit autophagy maturation, was combined with an autophagy stimulator, nanoliposomal C6-ceramide; this mixture enhances autophagy vacuole accumulation and decreases the autophagy maturation of the treated cells [134]. Another example of the use of these alkaloids is seen with vincristine and CD24, a protein that is bound to membrane lipid raft microdomains through a glycosylphosphatidylinositol anchor. CD24 is associated with poor prognosis in several forms of tumor tissues [135,136,137]. Furthermore, CD24 was reported to cause chemotherapy resistance [138,139,140] and regulate various signaling pathways in diverse types of tumor cells [141]. Vincristine (3) was reported to suppress the CD24 expression and eliminate autophagy via suppression of the PTEN-AKT-MTOR complex 1 (MTORC1) pathway in WERI-Rb-1 (HTB-169) and Y79 cell lines. Therefore, the sensitivity of retinoblastoma cells to vincristine (3) increases significantly [142].

Camptothecin (5), a natural compound originally derived from the Asian tree *Camptotheca acuminate*, was synthesized in 1966 [143]. A low dose of camptothecin (5) stimulates autophagy through the AMPK-TSC2 (TSC complex subunit 2) MTOR pathway in human CRC cell lines, HCT116 and RKO [144]. Camptothecin (5) and its derivatives have shown anti-cancer function against several types of tumor cells. It was found that camptothecin (5) enhances MYC/c-Myc expression, which in turn upregulates the expression of UPR sensor proteins in human prostate cancer (LNCaP cells), as well as reactive oxygen species (ROS) generation, which induces autophagy through the Ca^2+^-AMPK and MAPK/JNK-AP-1 signaling pathways [145]. Human non-small cell lung cancer (NSCLC) H1299 cells can be sensitized to camptothecin (5) by modulating autophagy. Camptothecin (5) increases autophagosome formation in H1299 cells, in a dose-related mode. The induced autophagy exerts a protective role in camptothecin-treated cells with regard to DNA damage and apoptosis. It was suggested that a combination of camptothecin (5) with an autophagy suppressor may be considered an effective approach for lung cancer treatment [146].

Matrine (6), an alkaloid isolated from *Sophora flavescens* Ait, has been reported to function as an autophagy inhibitor by blocking autophagic degradation and modulating the maturation procedure of lysosomal proteases in the gastric cancer (GC) SGC-7901 cell line [147].

Berberine (7) is an isoquinoline quaternary alkaloid isolated from many well-known medicinal plants, such as *Rhizoma coptidis* [148]. Some synthesized derivatives of berberine (7) with improved efficacy and bioavailability (NAX053, NAX056, NAX057, NAX080 and NAX081) have been reported to increase LC3 lipidation (LC3-II) and consequently are able to stimulate autophagy in human colon carcinoma (HCT116) cells, expressing WT TP53 and in TP53-mutated drug-resistant (SW613-B3) cell lines [149].

Isorhynchophylline (8) is a tetracyclic oxindole alkaloid isolated from *Uncaria rhynchophylla*. A study showed that this alkaloid results in an enhancement of autophagy markers including LC3-II and stimulates autophagy in an MTOR-independent but BECN1-dependent manner in neuronal cells including SH-SY5Y, PC12 cells and N2a, as well as primary cortical neurons. Moreover, isorhynchophylline (8) can trigger autophagy in vivo in the fat bodies of Drosophila L3 larvae [150].

Tetrandrine (9) is a bisbenzylisoquinoline alkaloid isolated from the stem of the creeper *Stephania tetrandra* S. Moore [151]. The effect of tetrandrine (9) on human hepatocellular carcinoma (HCC) metastasis was evaluated both in vivo and in vitro. The obtained data showed that this compound stimulates autophagy in human liver cells, which modulates cancer cell metastasis by inhibiting WNT-CTNNB1/β-catenin pathway activity and decreasing MTA1 (metastasis-associated 1) expression [152].

Hernandezine (10) is an isoquinoline alkaloid isolated from *Thalictrum glandulosissimum.* Hernandezine (10) demonstrates potent cytotoxic effects toward different cancer cell types (A549, HeLa, PC3, MCF-7, Hep3B, HepG2 and H1299), while it shows low cytotoxicity toward normal liver hepatocytes. This compound activates AMPK directly and consequently induces ACD in apoptosis-resistant or drug-resistant cancer. Moreover, the resulting alkaloid-induced cellular toxicity is related to the upregulation of ATG7-dependent autophagy [153].

Bisleuconothine A (11) is a bisindole alkaloid isolated from *Leuconotis griffithii.* This compound increases LC3 lipidation in lung cancer A549 and in breast cancer MCF-7 cells, which might be due to the induction of autophagy via the AKT-MTOR signaling pathway. Bisleuconothine A (11) induces formation but inhibits the degradation of autophagosomes [154].

The utmost common cancer of the oral cavity is oral squamous cell carcinoma (OSCC), which is a serious global well-being issue. The major constituents of *Murraya koenigii* (L.) Sprengel are carbazole (12) alkaloids. The anti-cancer activities and mechanism of action of methanol and dichloromethane extract of the leaves of *M. koenigii* were investigated in OSCC cell line, CLS-354. The obtained results indicated that both isomahanine and carbazole alkaloids-mahanine cause increased accumulation of SSQTM1/p62, with a coordinated expression of LC3B-II and cleaved CASP3 (caspase 3), implying inhibition of autophagy flux. It was suggested that suppression of autophagy flux is related to carbazole-induced apoptosis [155].

Capsaicin (8-methyl-N-vanillyl-6-nonenamide) (13) is a functional element of chili peppers. Capsaicin (13) triggers mutated TP53 protein degradation through autophagy. Treatment with capsaicin (13) results in a reduction in mutated TP53 levels and reactivated WT TP53 protein in mutant TP53-carrying cells and the TP53 reactivation contributes to capsaicin-induced autophagy [156].

Alkaloids that are mentioned in this section are presented in Figure 6. Moreover, the schematic role of these natural compounds on different proteins involved in the autophagy pathway is demonstrated in Figure 7.

#### 5.1.2. Terpenoids

In 2005, the effect of some saponins isolated from soybeans was evaluated on the inhibition of colon tumor cell growth. According to the obtained results, a refined soybean B-group saponin extract caused the accumulation of human colon adenocarcinoma HCT-15 cells in the S phase along with a substantial increase in LC3-II levels and numerous autophagy vacuoles, showing autophagy induction [157]. Neoalbaconol (14), a terpenoid isolated from the fungus *Albatrellus confluens*, targets PDPK1/PDK1 (3-phosphoinositide-dependent protein kinase 1) and inhibits the subsequent PI3K-AKT signaling in the nasopharyngeal carcinoma (NPC) nude mouse model, resulting in activation of autophagy [158].

Guttiferone K (15) is isolated from *Garcinia yunnanensis* Hu. This compound results in stress-induced cell death through an AKT-MTOR-dependent autophagy pathway in cervical cancer HeLa cells by MAPK/JNK activation and AKT-MTOR inhibition, causing LC3-II accumulation and SQSTM1 degradation [159].

Oridonin (16), an active diterpenoid isolated from *Rabdosia rubescens* [160], induces autophagy via deactivation of p-AMPK and downregulation of SLC2A1/GLUT1 (solute carrier family 2 member 1) in the CRC cell line, SW480 [161].

Paclitaxel (17) is a taxane class diterpenoid, which was first discovered in the Pacific yew tree (*Taxus brevifolia* Nutt). Paclitaxel (17), with the trademark of Taxol, has been identified as a promising anti-cancer drug [162]. The anti-cancer effects of paclitaxel (17) have been extensively studied [163,164]. The accumulation of *ATG* genes (*ATG7, ATG12*, *ATG5*), *BECN1* and *MAP1LC3B* has been found in drug-resistant HeLa cells (HeLa-R) treated with paclitaxel (17). HeLa-R cells may undergo a shift in cellular metabolism from the tricarboxylic acid cycle to glycolysis; this activates HIF1A in HeLa-R cell lines. Suppression with koningic acid or glycolysis by 2-deoxy-D-glucose and downregulation of *HIF1A* by specific siRNA can reduce autophagy and increase the sensitivity of HeLa-R cells to paclitaxel (17) [165]. In 2016, it was found that paclitaxel (17) activates BECN1 in 2 NSCLC lines, A549 and Calu-3 in a dose-dependent manner. The microRNA *MIR216B* levels are significantly downregulated in paclitaxel-treated cells. In fact, paclitaxel (17) modulates *MIR216B*, which targets BECN1 upregulation to increase autophagy in NSCLC [166]. Research shows that paclitaxel (17) can suppress the proliferation of cervical cancer SiHa cells in a dose-related mode. However, following treatment with the long non-coding RNA *RP11-381N20.2* substantial changes are found in the expression of key regulators of cell death pathways and autophagy is reduced significantly [167].

In 2011, it was proposed that gelomulide K (18), a diterpene, can induce PARP1 (poly(ADP-ribose) polymerase 1) hyperactivation, AIFM/AIF nuclear translocation and cytoprotective autophagy in cancer cells. Gelomulide K (18) can synergize with paclitaxel (17); together, it shows low toxicity in normal cells. This compound induces autophagy through an ROS-mediated downstream signaling pathway. Moreover, the significance of the 8,14-epoxy group in the structure of gelomulide K (18) was highlighted for induction of cell death and autophagy [168].

Tanshinone IIA (19), an abietane diterpenoid, isolated from *Salvia miltiorrhizae*, was reported to trigger autophagy in KBM-5 leukemia cells through AMPK activation and suppression of MTOR and RPS6KB/p70S6K. Moreover, it significantly increases the expression of LC3-II and activates the MAPK/ERK signaling pathway, such as RAF, MAPK/ERK and RPS6KA1/p90 RSK in a dose- and time-related mode [169].

A sesquiterpene lactone named helenalin (20) was reported to trigger autophagy via inhibition of RELA/NF-κB p65 expression in the RKO colon carcinoma, MCF-7 breast adenocarcinoma and A2780 human ovarian cancer cell lines. According to the acquired results, helenalin (20) treatment induces autophagy markers (*MAP1LC3B* and *ATG12*) and caspase activation, showing caspase activity is vital for ACD triggered by helenalin (20) [170].

Elevated MAPK/JNK signaling association with autophagy induction has been demonstrated [171]. Ursolic acid (21), a natural triterpenoid, was reported to modulate autophagy by stimulating the accrual of LC3B-II and SQSTM1/p62 with the engagement of the MAPK/JNK pathway in apoptosis-resistant CRC cells [172].

The results from an experiment on HCT116 colon cancer cells showed that ginsenoside compound K (22) stimulates autophagy mediated by the ROS-JNK1 pathway. Ginsenoside compound K (22) therapy of tumor cells causes sensitizing TNFRSF10A/TRAIL/APO2L1 (TNF receptor superfamily member 10a) and TNFRSF10A-triggered apoptosis through autophagy-related and -independent (TP53-DDIT3 pathway) TNFRSF10B upregulation [173].

Frondoside A (23), extracted from the sea cucumber *Cucumaria frondosa*, is a triterpenoid saponin with a sugar-steroid structure [174], which inhibits autophagy in human urothelial carcinoma cell lines. Accumulation of LC3-I/II and SQSTM1 and LC3-positive organelles (autophagosomes), time- and dose-dependently, indicates inhibition of autophagy [175].

Oblongifolin C (24) is a natural compound isolated from *Garcinia yunnanensis* Hu. It is an effective autophagy flux inhibitor. Contact with oblongifolin C (24) leads to an elevated quantity of autophagosomes and defective degradation of SQSTM1 [176].

The chemical structures of all autophagy-modulator terpenoids are mentioned in Figure 8. The schematic role of the mentioned terpenoids on the proteins involved in the autophagy pathway is shown in Figure 9.

#### 5.1.3. Flavonoids and Phenolic Compounds

Quercetin (25), a flavonoid molecule discovered in vegetables, grains, leaves and fruits has exhibited anti-cancer effects on different cancer cells [177,178,179]. Quercetin (25) causes preferential degradation of oncogenic RAS and leads to autophagy in HRAS/Ha-RAS-transformed human colon cells, DLD-1 and HT-29. Treatment with 3-methyladenine (3-MA), an autophagy inhibitor, inhibits quercetin-induced cell death. Moreover, the formation of EGFP-LC3 is detected in quercetin-treated cells. Moreover, zVAD-FMK, a general caspase inhibitor, fails to suppress vacuolization, demonstrating that the induced autophagy is caspase-independent [180]. Moreover, quercetin (25) induces autophagy in GC cells. The results indicate the engagement of PI3K-AKT-MTOR and HIF1A signaling in quercetin-triggered autophagy development. In the presence of quercetin (25), formation of acidic vesicular organelles (AVOs), autophagic vacuoles, conversion of LC3-I to LC3-II, activation of autophagy genes and recruitment of LC3-II to phagophores are detected. Accordingly, quercetin (25) triggers autophagy progression in GC cells, such as AGS, BGC-823, SGC-7901 and MKN-28 [181]. Quercetin (25) suppresses the proteasome function in breast cancer cells, resulting in blocking MTOR activity by a decrease in the MTOR substrates EIF4EBP1/4E-BP1 and RPS6KB/p70S6 kinase phosphorylation, consequently triggering autophagy [182]. *Emblica officinalis* extract reduces cell proliferation in ovarian cancer OVCAR3 cells, dose- and time-dependently, by significantly increasing the expression of the autophagy proteins BECN1 and LC3B-II in vitro. Quercetin (25) as one of the components of the extract also increases the expression of these proteins and consequently reduces cell proliferation in ovarian cancer cells under in vitro conditions [183].

Apigenin (26), a common dietetic flavonoid available in vegetables and fruits, shows promising biological effects against different cancer cells [184,185,186]. This compound initiates autophagy through the suppression of MTOR and RPS6KB/p70S6K of the JAK-STAT pathway in leukemia TF1 cells [187].

Genistein (27), a naturally derived isoflavonoid discovered in soy products [188], can trigger cell death through autophagy in ovarian cancer cells via a caspase-independent pathway in a starvation-like signaling response. Treatment with 25–100 µM genistein (27) results in reduced AKT phosphorylation [189]. In 2010, another investigation showed that this isoflavonoid as a tyrosine kinase inhibitor activates autophagy in ovarian cancer cells through the PDE4A/PDE4A4 (phosphodiesterase 4A) and SQSTM1 pathways. Genistein (27) inhibits both PRKC/PKC and MAPK/ERK inhibitors via suppression of PDE4A mass formation, which results in the induction of autophagy [190].

Salvigenin (28), a natural polyphenolic compound, increases the levels of autophagy factors, including the LC3-II:LC3-I ratio, ATG12 and ATG7 in human neuroblastoma SH-SY5Y cells [191].

Chrysin-organotin is a synthetic compound based on a natural flavonoid named chrysin (29). Research has shown that treatment with chrysin (29) and chrysin-organotin/Chry-Sn in breast cancer MCF-7 cells results in an enhancement of LC3-II levels, giving rise to the induction of autophagy in the treated cells [192].

Another natural flavonoid named baicalin (30) is present in several medicinal plants including *Scutellaria baicalensis* Georgi [193]. Repolarization of tumor-associated macrophages towards the M1 phenotype characterizes the immunocompetent microenvironment in favor of tumor regression. Baicalin (30) treatment in hepatocellular carcinoma induces this repolarization, which is implemented by autophagy induction through activation of the RELB-p52 pathway and degradation of TRAF2 (TNF receptor associated factor 2) [194].

Juglanin (31) is a flavonol compound isolated from *Polygonum aviculare*, which triggers autophagy and apoptosis in breast cancer cells via MAPK/JNK activation and ROS production. A MAPK/JNK inhibitor attenuates juglanin (31) and significantly induces apoptosis and autophagy, whereas an ROS scavenger can reverse them [195].

Resveratrol (32) is a stilbenoid that is found in significant concentrations in berries, red wine, grapes and some other herbal sources [196]. A connection between autophagy and resveratrol (32) has been shown in the treatment of several cancer cells. It has been demonstrated that this compound triggers autophagy in ovarian tumor cells. Morphology and ultra-structural changes and even a high amount of the anti-apoptotic proteins BCL2L1/Bcl-xL and BCL2 indicate that resveratrol (32) stimulates cell death in ovarian cancer A2780 and CaOV3 cells via a different mechanism, compared to apoptosis [197]. Resveratrol (32) significantly suppresses breast cancer cell development in breast cancer stem-like cells separated from SUM159 and MCF-7 by repressing the WNT-CTNNB1/β-catenin signaling pathway. Moreover, LC3-II, BECN1 and ATG7 are significantly upregulated by resveratrol (32) in breast cancer cells in a dose-related mode [198]. The resveratrol (32) impact was investigated in C33A (TP53 mutant), HeLa and CaLo (HPV18+), and CaSki and SiHa (HPV16+) cervical cancer cell lines. An elevated lysosomal permeability (autophagy) was observed in HeLa, C33A and CaLo cell lines in a concentration-independent manner [199]. Resveratrol (32) also induces autophagy in melanoma B16 cells via ceramide accumulation and suppression of the AKT-MTOR pathway [200]. Moreover, resveratrol (32) triggers autophagy-mediated cell death in prostate tumor cells (PC3 and DU-145) through regulation of a store-operated calcium entry (SOCE) mechanism, including downregulating STIM1 (stromal interaction molecule 1) expression, which triggers ER stress and therefore activates AMPK and inhibits the AKT-MTOR pathway [201]. The ability of resveratrol (32) to trigger a starvation-like signaling reaction through decreasing phosphorylated AKT and MTOR to start autophagy was reported in ovarian tumor cells [202]. However, the AKT-MTOR pathway is not engaged in resveratrol-triggered autophagy in resveratrol-treated U373 glioma cells. MAPK14/p38 or MAPK1/ERK2-MAPK3/ERK1 inhibitors decrease the autophagy pathway, suggesting that resveratrol-induced autophagy is positively controlled by MAPK14/p38 and the MAPK1-MAPK3 pathways [203]. Resveratrol (32) was concluded to trigger ACD in chronic myelogenous leukemia cells through both AMPK activation and MAPK/JNK-mediated SQSTM1 overexpression [204].

Another study reported that resveratrol (32) activates a novel type of autophagy in breast tumor cells. Resveratrol (32) can inhibit the activation of AKT and MTOR in the MCF-7 cell line. Resveratrol (32)-induced ACD is impeded by CASP3 expression, except for its enzymatic function. Overexpression of BCL2 cannot reverse the non-canonical autophagy induced by resveratrol (32) in these breast tumor cells and resveratrol (32)-induced autophagy acts as a caspase-independent cell death mechanism in breast tumor cells [205]. Moreover, in some studies, resveratrol (32) was reported to induce premature senescence in squamous cell carcinoma (SCC) A431, which is related to a blockage in autolysosome formation, as evaluated by the lack of colocalization of essential markers of lysosomes and autophagosomes LAMP2 (lysosomal associated membrane protein 2) and LC3, respectively. RICTOR (RPTOR independent companion of MTOR complex 2) is a crucial element of the MTORC2 complex. Resveratrol (32) is able to downregulate the amount of RICTOR, which in turn decreases RHOA GTPase activity, alters the actin cytoskeleton network, elevates the senescence-associated GLB1/β-gal function and attenuates the autophagy process, which may be the mechanism of tumor inhibition related to early senescence [206].

The phosphatidylinositol-3-phosphate (PtdIns3P) effectors, WIPI1 (WD repeat domain, phosphoinositide interacting 1) and WIPI2 (WD repeat domain, phosphoinositide interacting 2) act downstream during the start of autophagosome creation and following resveratrol (32) treatment. The involvement of WIPI1 with the phagophore membrane was confirmed by the localization of WIPI1 at the ER and the plasma membrane, upon triggering autophagy [207]. An investigation was conducted to assess the autophagy-inducing properties of resveratrol (32) in four human tumor cell lines (G361, U2OS, MCF-7 and HeLa). The results indicate that WIPI1 is specifically bound to PtdIns3P and becomes a membrane-associated protein of the phagophore. Moreover, WIPI1 increases the LC3-II level, upon nutrient starvation and functions upstream of both ATG5 and ATG7. It was suggested that resveratrol (32)-mediated autophagy occurs through the non-canonical pathway and is BECN1-independent but ATG7- and ATG5-dependent [208]. Resveratrol (32) was reported to increase the protein expression of ATG5, ATG7, ATG9 and ATG12 in a human hepatitis C-triggered Huh-7 hepatoma cell line inducing ACD [209].

Glioblastoma is one of the increasingly prevalent main brain cancers [210]. Resveratrol (32) induces the formation of autophagosomes in three human glioblastoma cell lines (U-251, U-87 MG and U-138 MG) via upregulation of autophagy proteins ATG5, BECN1 and LC3-II [211]. SIRT1 (sirtuin 1) is a NAD-dependent deacetylase that has a key function in regulating autophagy; it forms a complex with some autophagy machinery components ATG5, ATG7 and Atg8-family proteins and directly deacetylates these components in a NAD-dependent fashion [212]. Resveratrol (32) is able to regulate SIRT1 and PARP and consequently regulates induced autophagy in lung cells with cigarette smoke-mediated oxidative stress [213]. SERPINB4/SCCA1 (serpin family B member 4), an endogenous CTSL (cathepsin L) inhibitor, is widely expressed in various uterine cervical cells. Autophagy and the CTSL-SERPINB4/SCCA1 lysosomal pathway are engaged in resveratrol-induced cytotoxicity in cervical tumor cells. Moreover, resveratrol (32) increases the level of LC3-II and autophagosomes in these cells [214].

Curcumin (33) is a natural curcuminoid with high anti-cancer efficacy. Curcuminoids are the major polyphenolic-derived compounds found in the rhizome of many *Curcuma* species, which have been applied for centuries in conventional Indian and Chinese medicine. The primary study on the anti-cancer activities of curcumin (33) was published in 1987 [215]. The antineoplastic activity of curcumin (33) has been comprehensively investigated in several cancers, including melanoma, breast [216], colon and prostate cancers [217]. There are numerous studies demonstrating the beneficial effect of curcumin (33) in modulating autophagy. Curcumin (33) suppresses the AKT-MTOR-RPS6KB/p70S6K pathway and stimulates the MAPK1-MAPK3 pathway, thereby inducing autophagy [218]. Curcumin (33) exerts anti-proliferative effects on two human malignant glioma cell lines (U373-MG and U87-MG) by arresting the cell cycle in the G_2_/M phase and promoting autophagy by inhibiting the AKT-MTOR-RPS6KB/p70S6K pathway and stimulating the MAPK1-MAPK3 pathway [219]. In 2011, the participation of ROS in curcumin-triggered ACD was investigated. The data indicate that curcumin (33) leads to ROS production, which results in ACD in human colon cancer HCT116 cells. However, curcumin-triggered autophagy is not implicated in the ROS-dependent stimulation of MAPK1-MAPK3 and the MAPK14/p38 signaling pathway. Thus, a combination of curcumin (33) and agents that result in ROS production might enhance the cytotoxic effect of this compound [220].

Additionally, this compound exhibits anti-cancer activity against OSCC through induction of autophagy, which was confirmed by detecting a significant increase in LC3-I conversion to LC3-II in curcumin-treated cells. In this research, a substantial level of AVOs can be identified in spite of a brief duration of exposure (6 h) to curcumin (33) [221]. This compound also induces autophagy in the human lung adenocarcinoma cell line, A549, as confirmed by the analysis of LC3 and SQSTM1. In addition, it enhances the phosphorylation of AMPK and ACAC (acetyl-CoA carboxylase) in A549 cells, although it can also cause blocking and disruption in the AMPK signaling pathway [222]. Curcumin (33) induces autophagy in SK-UT-1 and human uterine leiomyosarcoma SKN cell lines through stimulation of the MAPK1-MAPK3 pathway; accordingly, suppression of the MAPK1-MAPK3 pathway decreases curcumin-induced autophagy [223].

Lysosomes have a vital function in the maturation and degradation of autophagosomes. A major molecular mechanism in the control of lysosomal action involves nuclear TFEB (transcription factor EB), which is the immediate downstream focus of MTOR. TFEB activity is regulated by the MAPK1/ERK2 pathway [224,225]. Data obtained from one study revealed that curcumin (33) triggers autophagy in human colon cancer HCT116 cells by activation of lysosomal function via suppression of the AKT-MTOR signaling pathway and induces TFEB transcriptional activity by direct binding to TFEB [226]. The effect of curcumin (33) was investigated on H2O2-triggered apoptosis and autophagy in EA.hy926 cells. In this case, curcumin (33) induces autophagy and LC3-II expression and suppresses the phosphorylation of AKT-MTOR, indicating its protective effect against oxidative stress damage by triggering autophagy via AKT-MTOR pathway inhibition [227]. Curcumin (33) also triggers autophagy and suppresses proliferation by downregulating the AKT-MTOR signaling pathway in human melanoma A375 and C8161 cell lines [228]. Curcumin (33) potentiates autophagy in pancreatic cancer PANC1 and BxPC3 cells, as confirmed by the promotion of autophagosome formation, upregulation of LC3-II and downregulation of the MTOR protein expression [229].

Finally, inhibition of apoptosis following curcumin (33) treatment in osteosarcoma MG63 cells increases autophagy due to upregulation of the MAPK/JNK signaling pathway [171]. In addition, the impact of curcumin (33) on the proliferation, apoptosis and autophagy of GC cells (SGC-7901 and BGC-823) shows that this compound inhibits PI3K via downregulating the PI3K-AKT-MTOR pathway and activates the TP53 signaling pathway by upregulating TP53 and CDKN1A/p21, both of which regulate autophagy [230].

Some derivatives of curcumin (33) also possess anti-cancer activities. A derivative of curcumin named *bis*-dihydroxy-curcumin (36) increases autophagosome formation and autophagy flux in human colon cancer HCT116 cells. An enhancement of ER stress and poly-ubiquitinated proteins is detected upstream of autophagy initiation and results in cell death. Thus, this compound induces autophagy through an ER-stress pathway [231]. Moreover, tetrahydrocurcumin (37) is one of the primary catabolic products of curcumin (33). This compound suppresses the PI3K-AKT-MTOR-RPS6KB/p70S6K, GSK3B **(**glycogen synthase kinase 3 beta) and MAPK14/p38 pathways and activates the MAPK1-MAPK3, MAPK8/JNK1-MAPK9/JNK2 pathways. These changes cause the initiation of autophagy in human leukemia HL-60 cells [232].

Honokiol (34), a biphenolic compound isolated from *Magnolia officinalis*, acts as a potent inhibitor of melanoma B16-F10 cells by targeting melanoma stem cell signaling pathways by either suppressing NOTCH1 (notch receptor 1) signaling or increasing autophagosome formation, suggesting that honokiol (34) can also induce autophagy in melanoma cells via attenuating AKT-MTOR [233].

Magnolol (35), a lignin compound, induces autophagy via the PI3K-AKT signaling pathway in human GC cells SGC-7901 at elevated amounts, but it is not engaged in cell death [234].

Rottlerin (38) is a polyphenol isolated from *Mallotus phillippinensis* (the monkey-faced tree, Kamala) [235]. Rottlerin (38) triggers autophagy in human fibrosarcoma HT1080 cells through a PRKCD/PKCδ-independent pathway. The LC3-II protein level also increases following the administration of rottlerin (38); early induction of autophagy acts as a survival mechanism versus late apoptosis [236]. Another pathway through which autophagy is regulated via rottlerin (38) involves PRKCD-TGM2 (transglutaminase 2) and downregulation of PRKCD results in ACD [237]. Moreover, this phytochemical inhibits MTORC1 signaling through its negative controller TSC2 in breast tumor cells in a high nutritional environment. Rottlerin (38) causes the assembly of EGFP-LC3-II and free EGFP, along with proteolytic fragments of EGFP [238]. Moreover, rottlerin (38) induces ACD in apoptosis-resistant breast cancer MCF-7 cells, as confirmed by an elevated LC3-II:LC3-I ratio and degraded SQSTM1 protein. The autophagic death appears to be triggered by non-canonical signaling cascades because it is independent of BCL2, BECN1, AKT and MAPK/ERK pathways [239].

Another investigation using breast cancer stem cells revealed that rottlerin (38) treatment increases the expression of ATG12, BECN1 and LC3 proteins. Rottlerin (38)-triggered autophagy is mediated through stimulation of the AMPK pathway [240]. Rottlerin (38) treatment results in BCL2- and BECN1-independent autophagic death in apoptosis-resistant breast cancer MCF-7 cells. Interestingly, it was reported that rottlerin (38)-triggered autophagy is mediated by suppression of MTORC1; however, any role for AMPK was excluded and it was suggested that rottlerin (38) interacts directly with MTORC1 and induces autophagy through a novel AMPK- and MTORC1 phosphorylation-independent mechanism [241]. Rottlerin (38) triggers autophagy via the PI3K-AKT-MTOR signaling pathway in prostate tumor stem cells; as a result, the formation of autophagosomes, an increase in LC3-II and induction of BECN1, ATG7, ATG5 and ATG12 are observed [242].

Gossypol (39) is a natural phenol isolated from the cotton plant (genus *Gossypium*) [243]. An enhancement in LC3-II levels and autophagosome numbers in mutant TP53 pancreatic adenocarcinoma PANC-1 cells are observed in the presence of a combination of gossypol (39) and BRD4770, an inhibitor of histone methyltransferase EHMT2/G9a (euchromatic histone lysine methyltransferase 2). This combination acts in a BNIP3 (BCL2 interacting protein 3)-dependent manner. It seemed that gossypol (39) enhances BRD4770 cytotoxicity significantly in TP53-mutant cells and induces autophagy-related cell death. The upregulation of BNIP3 upon EHMT2 inactivation might be responsible for the synergistic ACD effect. Thus, EHMT2 inhibition may solve drug resistance issues in certain cancer cells [244].

Salvianolic Acid B (40) is a phenolic acid isolated from the dried root and rhizome of *Salvia miltiorrhiza* that triggers autophagy via the AKT-MTOR pathway suppression in CRC cell lines, HCT116 and HT-29 [245].

Isoliquiritigenin (41), a natural flavonoid derived from the root of licorice (Glycyrrhiza uralensis), triggers autophagy in drug-resistant breast tumor MCF-7 cells through *MIR25* inhibition, thus functioning as a regulator of chemoresistance-associated autophagy by promptly elevating ULK1 expression [246].

Summaries of the flavonoids and phenolic compounds that act as autophagy modulators are shown in Figure 10 and Figure 11. The schematic role of these compounds on the proteins involved in the autophagy pathway is shown in Figure 12.

#### 5.1.4. Other Natural Compounds

Elaiophylin (42) has been identified as an autophagy inhibitor. It promotes anti-tumor activity in human ovarian cancer cells by autophagosome collection but stops autophagosome maturation by attenuation of lysosomal cathepsin function, which destabilizes lysosomes. This compound causes the accumulation of SQSTM1 [247].

Lipoic acid (43) is a natural disulfide complex. Lipoic acid (43) suppresses MGMT (O-6-methylguanine-DNA methyltransferase) by interfering with its catalytic Cys145 residue, inducing autophagy in a TP53-independent manner in isogenic colorectal carcinoma HCT116 cell lines. Moreover, dephosphorylation of AKT is observed, which is accompanied by an increase in the autophagy marker LC3B-II. Furthermore, lipoic acid (43) enhances the cytotoxic effects of temozolomide, an alkylating anti-cancer drug [248].

A combination of resveratrol (32) and spermidine (44), a polyamine found in soybean and citrus fruit [249], triggers autophagy in human colon cancer HCT116 cells through AMPK-MTOR-independent pathways; although the major targets of resveratrol (32) and spermidine (44) are different, both factors activate converging pathways. They both stimulate MTOR-independent autophagy and elicit quite comparable alterations in the phosphoproteome and acetylproteome. This result proposes that both factors synergistically activate autophagy via a multi-faceted mechanism that engages a considerable amount of deacetylation responses [250]. 

Allicin (45), a diallylthiosulfinate, is an active compound in freshly crushed garlic extract [251]. Allicin (45) possesses an anti-cancer effect against different cancer cell lines by proliferation inhibition [252]. This compound activates ACD in thyroid cancer SW1736 and HTh-7 cells by suppressing the AKT-MTOR signaling pathway. Moreover, combined usage of allicin (45) and rapamycin induces more cell death, compared to that induced by either compound alone [253].

Trehalose (46), a natural disaccharide sugar, is a known autophagy enhancer. The disassembly effect of trehalose on stress granules, however, is mediated via p-EIF2A instead of autophagy. Upregulation of basal p-EIF2A by trehalose (46) may, nonetheless, be a mechanism behind its autophagy-stimulating capacity [254]. Moreover, benzyl isothiocyanate (47) causes autophagy in lung cancer A549 cells through EIF2AK3 and EIF2A pathways, and preliminary treatment with the ER stress suppressor 4-PBA attenuates the autophagy induction [255]. (+)-Grandifloracin (48), derived from Uvaria dac, shows better toxicity to human pancreatic cancer, PANC-1 cells. This compound strongly inhibits AKT activation and induces autophagy in PANC-1 cells [256].

Natural products that are mentioned in this section are presented in Figure 13. Moreover, the schematic role of these compounds on the proteins involved in autophagy is shown in section A of Figure 14. The autophagy-mediated anti-cancer activity exerted by natural compounds is summarized in Table 1.

### 5.2. The Effects of Natural Compounds on the Apoptosis Signaling Pathway

#### 5.2.1. Alkaloids

Vinblastine (1) induces apoptosis in KB-3 carcinoma cells via JUN auto-amplification and TP53-independent downregulation of CDKN1A/p21/WAF1/CIP1 [257]. Besides induction of JUN phosphorylation, this compound activates MAPK/JNK and promotes the inactivation of MAPK/ERK in KB-3 cells. Moreover, Vinblastine (1) decreases TP53 protein expression [258]. Vinblastine (1) can induce apoptosis through activation of MAPK/JNK, which might be mediated by BCL2 inhibition in leukemia cell lines [259].

Vincristine (3) induces apoptosis in Jurkat acute lymphoblastic leukemia cells by activation of CASP3 and CASP9. Moreover, the production of ROS is found early in Jurkat cells by exposure to vincristine (3), which confirms a mitochondrial role in vincristine (3)-induced apoptosis [260]. In melanoma cell lines vincristine (3)-triggered apoptosis is related to the activation of MAPK/JNK. Moreover, phosphorylation of BCL2 results in the activation of BAX-BAK1, discharge of CYCS from the mitochondria and stimulation of caspases [261]. A small, anti-apoptotic heat shock protein, HSPB1/HSP27, is an anti-apoptotic molecule involved in the survival of cells exposed to several types of stress, including anti-cancer drugs. Vincristine (3) regulates the phosphorylation of HSPB1 in breast cancer cells [262,263]. Moreover, this compound activates AMPK in B16 melanoma cells through ROS-dependent STK11/LKB1 (serine/threonine kinase 11) activation. Furthermore, activation of TP53 and inhibition of MTORC1 are detectable in vincristine (3)-treated B16 melanoma and this treatment causes apoptosis in this cell line [264]. Moreover, this compound induces apoptosis in SH-SY5Y neuroblastoma cells and promotes the expression of CASP3, CASP9 and CCNB (cyclin B), while decreasing the expression of CCND (cyclin D) [265].

Matrine (6) was previously mentioned as an autophagy inhibitor. This compound induces apoptosis in different cancer cell lines. The antineoplastic activity of matrine (6) was investigated against human BxPC-3 and PANC-1 pancreatic tumor cells. The results showed this compound induces apoptosis by decreasing the ratio of BCL2:BAX, upregulating FAS (Fas cell surface death receptor) and increasing the activation of CASP3, CASP8 and CASP9 in a dose-related mode. Moreover, matrine (6) suppresses the growth of pancreatic cancer cells [266].

In 2004, a study demonstrated that tetrandrine (9) reduces the survival of U937 leukemia cells in a dose- and time-related mode. This compound induces apoptosis in U937 cells through oxidative stress, as well as activation of caspases and PRKCD. Moreover, it seems that there might be a mutual regulation between PRKCD and caspases for their activation [151]. Tetrandrine (9) induces apoptosis in the cultured and subcutaneous mouse colon cancer CT-26 cells, which can be attributed to stimulation of the MAPK14/p38 signaling pathway [267].

Subditine (49) is a monoterpenoid indole alkaloid isolated from the bark of the *Nauclea subdita*. This compound triggers apoptosis in human prostate cancer LNCaP and PC-3 cell lines by downregulation of BCL2 and BCL2L1. Its apoptotic effect is TP53-dependent in LNCaP (WT TP53), but not in PC-3 (*TP53*-null) cells [268].

Caffeine (50) is a well-known alkaloid found in tea, coffee and the cacao plant [269]. It was suggested that caffeine (50) triggers apoptosis by an increase of autophagy through PI3K-AKT-MTOR-RPS6KB/p70S6K suppression and MAPK1-MAPK3 pathway activation in various cell lines including HeLa, PC12D and SH-SY5Y. Apoptosis is partially attenuated by blocking caffeine-induced autophagy via 3-MA treatment or *atg7* knockout [270]. Chrysin-organotin and chrysin (29) increase the level of ROS and CASP3, indicating that these compounds induce apoptosis to suppress the expansion of human breast cancer MCF-7 cells [192].

Camptothecin (5) induces autophagy in H1299 and H460 lung cancer cells, which can protect these cells from apoptosis. The results indicate that activation of CASP9 is enhanced in response to the suppression of autophagy by 3-MA [146]. A derivative of camptothecin (5), named hydroxycamptothecin (51), increases apoptosis in human Tenon’s capsule fibroblasts. It is suggested that an ER stress reaction and mitochondrial dysfunction have participated in apoptosis, which might be mediated by the PERK pathway [271]. Low-dose camptothecin (5) induces autophagy; however, suppression of autophagy causes an increase in apoptosis and attenuation in senescence possibly by blocking the TP53-CDKN1A/p21 pathway in human CRC cell lines, HCT116, and RKO. In this case, there may be a positive circuit between BECN1-mediated autophagy and TP53, through which autophagy plays a switched role between premature senescence and apoptosis [144].

The chemical structures of alkaloids are presented in Figure 6. The schematic role of these alkaloids on different proteins involved in apoptosis can be seen in Figure 15.

#### 5.2.2. Flavonoids and Phenolic Compounds

As mentioned above, quercetin (25) is a naturally accruing flavonoid molecule that exhibits anti-cancer properties in different cancer cells [177,178,179]. There is a wide range of publications on the cytostatic and pro-apoptotic effects of quercetin (25) [272,273,274]. In 2006, an investigation reported that TNFRSF10A/TRAIL-induced apoptosis is increased by quercetin (25) via AKT dephosphorylation and activation of caspases in human prostate cancer LNCaP and DU-145 cells [275]. Later, in 2016, the results from a study showed that the complex of quercetin (25) and TNFRSF10A triggers apoptosis in breast cancer cells, as indicated by increased PARP cleavage and activation of the executioner caspases; CASP3 and CASP7, which have therapeutic potential for all types of hormone-dependent and triple-negative breast cancer (TNBC) cells [276]. Moreover, quercetin (25)-triggered apoptosis in human prostate cancer LNCaP cells is mediated by dissociation of BAX from BCL2L1 and the activation of caspases [277]. Quercetin (25) possesses inhibitory activities against GC cells, BGC-823. This compound induces apoptosis via the mitochondrial pathway, as confirmed by a reduction in the BCL2:BAX ratio and an elevation in the CASP3 expression [278]. Quercetin (25) induces apoptosis in the drug-resistant sphere model of OSCC via inhibition of the MAPK14/p38-HSPB1 pathway [278]. Moreover, quercetin (25) induces apoptosis via PI3K-AKT-MTOR signaling pathway inhibition in MCF-7 breast tumor stem cells. This effect is achieved by upregulating the BAX expression and downregulating the BCL2 expression [279].

Epigallocatechin gallate (52) is a flavonoid found in green tea. Administration of this compound in combination with *TP53* silencing, leads to an anti-tumoral effect in the Hs578T cell culture model of a TNBC cell line, by inhibition of autophagy and activation of pro-apoptotic genes. Epigallocatechin gallate (52) along with *TP53* siRNA altered gene expression, promotes apoptosis, decreases cell survival and reduces angiogenesis and autophagy [280].

Genistein (27), a naturally occurring flavonoid, triggers apoptosis in human colon cancer HT-29 cells, mainly via upregulation of CDKN1A/p21 and BAX-BCL2 expression [281]. Moreover, apigenin (26), another flavonoid, triggers caspase-related apoptosis in myeloid leukemia HL60 cells through inhibition of the PI3K-AKT pathway. This effect is cell-dependent and is not observed in erythroid leukemia TF1 cells. Moreover, downregulation of the JAK-STAT pathway was observed in both cells [187].

In contrast to the case with normal tissues, curcumin (33) by itself and in conjunction with chemotherapy or radiation can induce apoptosis in most cancers. Curcumin (33) stimulates apoptosis via modulating various signaling pathways in cancer cells. In pancreatic cancer PCa cells in high concentration, this compound can upregulate pro-apoptotic genes such as *BAX*, whereas it downregulates anti-apoptotic genes such as *BCL2* [229]. Inhibition of NFKB is reported to be one of the most important targets for curcumin (33) [282]. Curcumin (33) results in an enhancement in cell death of CRC cells and inhibits cell growth in conjunction with 5-fluorouracil (5-FU), which itself downregulates NFKB [283]. In addition, curcumin (33) reduces chemoresistance via inhibition of NFKB and anti-apoptotic genes, such as *BCL2* and *BCL2L1* [284]. Other synergistic effects on NFKB suppression by curcumin (33) and 5-FU have also been reported on additional cancer cell types like esophageal squamous cell carcinoma [285].

Inhibition of PI3K-AKT is another way to simplify apoptosis initiation in cancer cells, in specific in cancers with a mutation in PTEN. The administration of curcumin (33) in breast cancer MCF-7 cells results in apoptosis via downregulation of AKT and upregulation of PTEN. Moreover, curcumin (33) inhibits *MIR21* by upregulating the PTEN-AKT signaling pathway. Thus, the *MIR21*-PTEN-AKT signaling pathway is an important mechanism for the anti-cancer impact of curcumin (33) [286].

One of the top crucial roles of TP53 is the induction of apoptosis in pre-cancerous and cancer cells. Thus, activation of TP53 has been suggested as a mechanism for ameliorating the tumor reaction to chemotherapy and radiotherapy [287]. Curcumin (33) triggers apoptosis in human breast cancer MCF-7 cells through TP53 activation and upregulation of BAX [288]. The same result was reported about the impact of curcumin (33) on the upregulation of TP53 in a human multiple myeloma MM RPMI 8226 cell line [289]. In 2017, curcumin (33) was reported to affect apoptosis via a decrease in TP53 and ESR1/ERα in hormone-dependent breast cancer T-47D cells. However, it was suggested that in this type of cell, the curcumin (33) effect may not be dependent on TP53 and the downregulation of protein levels was due to its cytotoxicity to the cells [290]. Curcumin (33) downregulates the expression of pro-apoptotic proteins BCL2 and BCL2L1, upregulates the expression of pro-apoptotic proteins BAX, BAK1, BBC3/PUMA and PMAIP1/NOXA and activates CASP3 and CASP9 in prostate cancer cells, which enhances the apoptosis-triggering ability of TNFRSF10A in androgen-unresponsive PC-3 cells and sensitized androgen-responsive TNFRSF10A-resistant LNCaP cells. In addition, curcumin (33) upregulates the expression of TP53 and downregulates AKT when combined with TNFRSF10A/TRAIL [291]. Another investigation showed that curcumin (33) induces apoptosis in head and neck squamous cell carcinoma (HNSCC) through suppression of NFKB mediated by NFKBIA/IκBα kinase inhibition and, consequently, results in the suppression of cell survival and cell proliferative genes, including *BCL2, CCND1*, *MT-CO2/COX-2*, *IL6* (interleukin 6) and *MMP9* (matrix metallopeptidase 9) [292].

Furthermore, curcumin (33) is able to activate SMPD (sphingomyelin phosphodiesterase) and inhibit sphingolipid-modifying enzyme activity, leading to ceramide generation and, accordingly, resulting in apoptosis in multidrug-resistant human leukemia HL60 cells [293]. Further study by the same group demonstrated that ceramide formation by curcumin (33) is controlled by crosstalk among BCL2, BCL2L1, caspases and glutathione. It was suggested that curcumin (33) triggers apoptosis in human leukemia HL60 cells via suppression of glutathione, leading to ceramide generation. Moreover, it was shown that activation of SMPD by curcumin (33) is accompanied by reduction of glutathione that is required for the stimulation of CASP8 and suppression of BCL2L1 [294]. Similarly, curcumin (33) is proposed as a powerful anti-cancer factor for uterine leiomyosarcoma (LMS) due to its noticeable impact on the induction of apoptosis and autophagy. Following curcumin (33) treatment of leiomyosarcoma SK-UT-1 and SKN cells, the expression of LC3B-II increases, SQSTM1 levels decrease and the MAPK1-MAPK3 pathway is activated [223]. Moreover, considerable evidence has demonstrated that curcumin (33) induces cancer cell apoptosis via autophagy activation. For instance, curcumin (33) enhances the levels of BECN1 and LC3-II in a leukemia cell line (K562). Additionally, an autophagy inhibitor, bafilomycin A_1_, represses curcumin-triggered K562 cell death. In this instance, autophagy is not the major reason for the death of K562 cells; curcumin (33) treatment might promote apoptosis [295].

Another study indicated that the combination of curcumin (33) with tamoxifen promotes autophagy and apoptosis in human melanoma cells more effectively than each drug alone. Curcumin (33) induces high levels of apoptosis in A375 and the G361 chemoresistant malignant human melanoma cell lines. There is a significant induction in autophagy and apoptosis following combination treatment with curcumin (33) and tamoxifen in low doses [296]. The expression levels of p-MAPK1-MAPK3 in curcumin (33)-treated TNBC cells decrease significantly, in comparison to the control group. Consequently, curcumin (33) might be able to suppress the proliferation of TNBC cells through EGFR signaling pathway inhibition and apoptosis induction [297].

Upregulation of *TNFRSF10A* has an important function in the sensitization of cancer cells to apoptosis [298]. ROS generation, stimulation of TP53 and suppression of NFKB are proposed to upregulate the expression of *TNFRSF10A* [299]. Curcumin (33) enhances TNFRSF10A-triggered apoptosis even in TNFRSF10A-resistant breast tumor cells, activating MAPK/ERK and AKT [300]. Furthermore, curcumin (33) can trigger TNFRSF10A-induced apoptosis in TNFRSF10A-resistant breast cancer cell lines via regulating apoptosis-related proteins such as MCL1, MAPK/ERK and AKT and genes such as *MCL1* [300,301]. Curcumin (33) was reported to increase the susceptibility of prostate cancer LNCaP cells to TNFRSF10A by inhibiting NFKB stimulation through hindering phosphorylation of NFKBIA/IκBα and its degradation [302]. Moreover, when a combination of curcumin (33) was administered in hormone-refractory prostate cancer cells and TNFRSF10A, both NFKB and p-AKT are markedly decreased, causing apoptosis induction [303]. In 2005, a study showed that curcumin (33) sensitizes TNFRSF10A-triggered apoptosis in human renal cancer cells through ROS-mediated and DDIT3-independent upregulation of TNFRSF10B [304]. It was reported that curcumin (33) and TNFRSF10A together enhance the induction of apoptotic cell death in chemoresistant ovarian cancer SKOV3 and ES-2 cells by activating both the intrinsic and extrinsic apoptotic pathways [305]. Triazolyl curcumins (53,54) alone or in combination with TNFRSF10A induce apoptosis in a dose-related mode. These compounds induce more apoptosis, compared to the TNFRSF10A treatment alone; apoptosis increases by more than 80% when cells are treated with these compounds [306].

Curcumin (33) was tested on ovarian carcinoma cells with differing TP53 status (wild type (WT) *TP53*: HEY and OVCA429; mutant *TP53*: OCC1; null *TP53*: SKOV3). The results indicate cell death by apoptosis via both the extrinsic and intrinsic pathways by activating CASP3, CASP8 and CASP9, CYCS release and BID (BH3 interacting domain death agonist) cleavage into truncated BID. This activity is TP53-independent and involves MAPK14/p38 activation, downregulation of BCL2, BIRC5/survivin expression and AKT signaling [307]. Curcumin (33) can also induce apoptosis in breast cancer cells transformed by radiation in the presence of estrogen by promoting ROS [308]. ROS accumulation leads to TP53-CDKN1A/p21- and CDKN2A/p16-RB1-mediated breast cancer suppression [216]. Curcumin (33) could also trigger breast cancer cell apoptosis by activating ROS. Curcumin (33)-triggered ROS collection results in TP53-CDKN1A/p21 and CDKN2A/p16-RB1-mediated breast cancer suppression [216].

As mentioned above, curcumin (33) triggers autophagy in human GC cell lines. Treatment with the autophagy inhibitor 3-MA significantly promotes apoptotic cell death triggered by curcumin (33). This treatment increases the population of the cells at the initiation and end stages of apoptosis by BCL2 downregulation, BAX upregulation and CASP3 and CASP9 activation [255]. Moreover, curcumin (33) in combination with quercetin (25), inhibits the growth of GC cells, MGC-803 via initiation of apoptosis through the mitochondrial pathway, following the release of CYCS and decreased phosphorylation of AKT and MAPK/ERK [309].

Demethoxycurcumin (DMC) (55) and bisdemethoxycurcumin (BDMC) (56) are curcuminoids discovered in the rhizome of many *Curcuma* species. The antineoplastic impact of these compounds on human lung cancer NCI H460 cells was investigated. The results indicate that BDMC (56) increases the activities of CASP3, CASP8 and CASP9, ROS levels and Ca^2+^ production in NCI H460 cells. Moreover, it was suggested that BDMC (56) induces apoptosis through multiple signaling mechanisms such as extrinsic, intrinsic and ER stress pathways [310]. Another curcuminoid, DMC (55), suppresses cell growth and triggers apoptosis in the same human lung cancer cell line via a mitochondrial-dependent pathway. DMC (55) activates CASP3, CASP8 and CASP9 and promotes AIFM, ENDOG and PARP expression [311].

Moreover, some analogs of curcumin (33) show anti-cancer activities against different cancer cell lines. A mono-carbonyl analog of curcumin (33), WZ35 (57), triggers apoptosis in NSCLC H1975 cells via the production of ROS, mitochondrial dysfunction and an ER stress pathway [312]. Along these lines, curcumin (33) triggers both autophagy and apoptosis in osteosarcoma MG63 cells. This study provided an important insight into the interplay between autophagy and apoptosis in osteosarcoma cells and medical administration approaches using curcumin (33). When autophagy is inhibited by 3-MA, curcumin-induced apoptosis increases; consequently, curcumin-triggered autophagy plays an anti-apoptotic function in osteosarcoma cells [171]. A novel bioactive curcumin (33) analog (PAC: 3,5-Bis [4-hydroxy-3-methoxybenzylidene]-N-methyl-4-piperidone) suppresses cell survival and induces apoptosis in oral cancer Ca9-22 cells, involving MAPK1-MAPK3, MAPK14/p38-MAPK/JNK, NFKB, CASP3, CASP9 and WNT cellular signaling pathways [313].

Resveratrol (32), a stilbenoid, triggers apoptosis via autophagy in human colon cancer HT-29 and COLO 201 cells. This compound increases the intracellular ROS level, which correlates with CASP3 and CASP8 activation and the elevation of LC3-II; these effects are diminished by the ROS scavenger N-acetyl cysteine (NAC) [314]. Autophagy lags apoptosis and protects cells from death, whereas apoptosis is a major reason for death in glioma U251 cells. It was suggested that suppressing resveratrol (32)-induced autophagy significantly affects its antineoplastic effects. In other words, autophagy suppresses resveratrol-triggered apoptosis [315]. Resveratrol (32) triggers cell cycle arrest and apoptosis. The CDKN1A/p21 expression is upregulated in a TP53-independent mode but it downregulates CCNE (cyclin E), CCNA (cyclin A) and CDK2 (cyclin dependent kinase 2) expression. Moreover, the ratio of pro-apoptotic to anti-apoptotic proteins increases, which is related to mitochondrial membrane depolarization and the enhancement in caspase activity. Resveratrol (32) shows no impact on FAS, FASLG (Fas ligand), MAPK1-MAPK3 and MAPK14/p38 expression; however, it negatively affects phospho-MAPK/ERK and phospho-MAPK14/p38 expression [209]. Another experiment demonstrated that low amounts of resveratrol (32) trigger death in the androgen-resistant prostate cancer DU-145 cell line; this is possibly an apoptotic response involving increased CASP3 activity via HSPA8/HSP70 [316]. Resveratrol (32) has a strong effect on CASP3 and CASP7 activation in C6 rat glioma cells by a decrease in phosphorylation of AKT when administered in combination with quercetin (25), whereas low amounts of resveratrol (32) (10 µM) or quercetin (25) (25 µM) individually have no impact on apoptosis initiation [315].

As discussed above, Rottlerin (38) is an ACD inducer. In some studies, rottlerin-induced autophagy is reported to be followed by apoptosis. For instance, this phytochemical triggers the intrinsic apoptotic pathway in *CASP3*-transfected breast cancer MCF-7 cells, based on the detection of activated CASP3 and CASP9, and PARP cleavage [239]. Prolonged exposure of breast cancer stem cells to rottlerin (38) results in apoptosis, which is correlated with phosphorylated AKT and MTOR suppression, phosphorylated AMPK upregulation and anti-apoptotic BCL2, BCL2L1, XIAP and BIRC2/cIAP1 downregulation. Thus, rottlerin (38)-triggered autophagy results in apoptosis in breast cancer stem cells [240]. Another investigation showed that prolonged exposure of human fibrosarcoma HT1080 cells to rottlerin (38) eventually causes caspase-independent apoptosis via a PRKCD-independent pathway. Conversely, the pre-administration of a specific inhibitor of autophagy (3-MA) to cells accelerates rottlerin (38)-triggered apoptosis. Consequently, it was suggested that rottlerin (38)-triggered autophagy possibly has several roles as a survival mechanism versus late apoptosis [236].

In pancreatic cancer stem cells, rottlerin (38) induces early autophagy based on the formation of autophagosomes, conversion of LC3-I to LC3-II, expression of ATG7 and accumulation of BECN1. Rottlerin (38)-induced autophagy results in apoptotic cell death via suppression of the PI3K-AKT-MTOR and AMPK pathways [317]. Moreover, it was reported that suppression of AMPK by shRNA blocks rottlerin (38)-triggered autophagy, representing that AMPK plays a critical function in rottlerin (38)-triggered apoptosis. These discoveries firmly propose that rottlerin (38)-triggered autophagy possibly also has several functions in apoptosis induction in prostate cancer stem cells [242].

Chrysophanol (58), an anthraquinone, triggers apoptosis in breast cancer MCF-7 and BT-474 cells by ROS production and blocking the PI3K-AKT and MAPK signaling pathways [318].

Magnolol (35) triggers mitochondrial-mediated apoptosis in human GC cells, SGC-7901, which engages mitochondria and PI3K-AKT-related pathways, as shown by an enhanced ratio of BAX:BCL2, activation of CASP3 and inhibition of PI3K-AKT [234].

The chemical structures of flavonoids and phenolic compounds can be found in Figure 10 and Figure 11. The schematic role of these compounds on the proteins involved in the apoptosis pathway is shown in Figure 16.

#### 5.2.3. Terpenoids

In 2007, a research study demonstrated that Isobavachalcone (59), a naturally occurring chalcone complex isolated from *Psoralea corylifolia*, remarkably reduces the levels of pro-CASP3 and CASP9. Moreover, it increases the level of cleaved CASP3 and CASP9 in both neuroblastoma IMR-32 and NB-39 cell lines. BAX is also induced by isobavachalcone (59) application. Therefore, this compound induces apoptotic cell death via the mitochondrial pathway and exhibits no cytotoxicity against normal cells [319]. This compound also triggers apoptosis- and autophagy-dependent cell death in myeloma H929 cells. Autophagy suppression through knocking down *BECN1* or applying the autophagy inhibitors 3-MA, bafilomycin A_1_ and CQ, significantly enhances isobavachalcone (59)-triggered cell death, suggesting that the suppression of autophagy in isobavachalcone (59)-administered myeloma cells may change the pro-survival impact of autophagy to apoptosis. Isobavachalcone (59)-induced apoptosis is suggested to be through the mitochondrial pathway and the proteolytic activation of PRKCD contributes to this effect [320].

Cucurbitacins (including cucurbitacin A, B, E, I and Q) have inhibited tumor growth and induced apoptosis via the suppression of STAT3 activation [321]. Cucurbitacin B (60) is a tetracyclic triterpene compound, which is a bitter principle of the family *Cucurbitaceae* [322,323]. *STAT3* siRNA induces either early or late stages of apoptosis in Hep2 laryngeal cancer cells [324]. Moreover, in 2008, the cucurbitacin B (60) was reported to exhibit anti-cancer activity on the same cell line through inhibition of STAT3 phosphorylation and its activation [325]. As discussed above, ginsenoside compound K (22) was considered an autophagy inducer. Moreover, this compound and TNFRSF10A can function cooperatively against colon cancer HCT116 and HT-29 cells. Ginsenoside compound K (22) enhances the pro-apoptotic impact of TNFRSF10A on colon cancer cells by inhibiting the expression of cell survival proteins, triggering expression of pro-apoptotic proteins (apoptosis regulator BAX, tBID and CYCS) and autophagy-dependent and autophagy-independent (TP53-DDIT3 pathway) upregulation of TNFRSF10B [173].

Frondoside A (23) was discussed as an autophagy modulator. This compound induces apoptosis in human urothelial carcinoma cell lines, RT4 (WT *TP53*), HT-1197 (WT *TP53*), TCC-SUP (*TP53* mutant), and T-24 (*TP53* mutant), which is related to the control of several pro-apoptotic elements, including CASP3, CASP8, CASP9, PARP, BAX, and CDKN1A/p21, externalization of phosphatidylserine and DNA fragmentation. Interestingly, caspase suppression and inhibition of *TP53* by gene silencing or pifithrin-α pretreatment do not inhibit the apoptotic function of frondoside A (23) [175].

The taxane paclitaxel (17) induces apoptosis through the following: (a) control of AKT-MAPK and ROS signaling in canine mammary gland cancer cells [326]; (b) MAPK14/p38 pathway inhibition in paclitaxel-resistant ovarian cancer cells [327]; (c) activation of MAP3K5/Ask1-MAPK14/p38 and MAPK/JNK-MAPK/SAPK pathways and inhibition of AKT and MAPK1/ERK2-MAPK3/ERK1 in U-937 human macrophage cells [328]; (d) downregulation of NFKB and AKT in human ovarian cancer OVCAR-3 cells [329]; and (e) MAPK/JNK-MAPK/SAPK signaling pathway activation in glioblastoma U373MG and leukemia cells [328,330,331]. Paclitaxel (17) significantly suppresses the growth of NSCLC and elevates the expression of *MEG3* (maternally expressed 3) and *TP53*. Moreover, it was proposed that the MEG3-TP53 pathway is engaged in the apoptosis of A549 cells induced by paclitaxel (17) [332]. Finally, the combination of paclitaxel (17) and pro-CASP3-activating compound-1, a pro-apoptotic agent, increases the antineoplastic activity of paclitaxel (17) against NSCLC cells, A-549 and H322m via activation of MSMP/PC-3, and thus the apoptotic pathway [333].

All the terpenoids mentioned in this part of the review are presented in Figure 8. Figure 17 represents the schematic role of these terpenoids on the proteins involved in the apoptosis pathway.

#### 5.2.4. Other Natural Compounds

Allicin (45), a diallyl thiosulfinate, presents antineoplastic properties. The activation of CASP3, CASP8 and CASP9 and subsequent induction of apoptosis upon treatment with allicin (45) are observed in the human cervical cancer SiHa cell line [334]. Allicin (45) can induce apoptosis in the colon cancer HCT116 cell line via the mitochondrial-dependent pathway, as it results in an increase in CYCS and apoptosis regulator BAX protein level but a reduction in the expression of the anti-apoptotic protein BCL2. Allicin-induced apoptosis occurs via a mechanism associated with nuclei translocation of the transcription factor *NFE2L2/NRF2* (NFE2-like bZIP transcription factor 2) [335]. Moreover, overexpression of *NFE2L2* results in the abolishment of allicin-triggered cell apoptosis in cervical cancer cells, showing that the anti-tumor role of allicin (45) is mainly increased by suppressing *NFE2L2* [336,337]. Allicin (45) induces apoptosis in GC cells, SGC-7901 through stimulation of both extrinsic (FAS-FASLG-mediated) and intrinsic (mitochondrial-dependent) pathways simultaneously [338]. An investigation indicated that allicin (45) induces apoptosis in breast cancer cells (MCF-7 and HCC-70) mediated by CASP3, CASP8 and CASP9 activation and loss of mitochondrial membrane potential (∆Ψm). These pro-apoptotic pathways (intrinsic and extrinsic) are also accompanied by upregulation of *CDKN1A/p21*, *PMAIP1/NOXA* and *BAK1* expression, as well as downregulation of *BCL2L1* [339].

Diallyl disulfide, another component of garlic, was reported to induce apoptosis via the activation of CASP3, degradation of PARP, generation of hydrogen peroxide and fragmentation of DNA in human leukemia HL-60 cells [340] and via CASP3 activation in human bladder cancer T24 cells [341] and activation of pro-CASP9 and CASP3 in colorectal adenocarcinoma HT-29 cells, especially in combination with sodium butyrate [342].

Chemical structures of natural compounds that are mentioned in this part as apoptosis inducers are presented in Figure 13. In part B of Figure 14, the effect of these compounds on the proteins involved in apoptosis is depicted.

Moreover, the anti-cancer activity of natural compounds by inducing apoptosis is summarized in Table 2.

### 5.3. The Effects of Natural Compounds on the UPR Signaling Pathway

The ER is engaged in protein synthesis and secretion and calcium (Ca^2+^) signaling [343]. Under ER stress many proteins including HSPA5, HSP90B1/GRP94, ER-associated degradation, PDI (protein disulfide isomerase), ATF6, ERN1, XBP1, EIF2AK3 and EIF2A are overexpressed in many forms of tumor cells [344,345]. ER stress conditions lead cancer cells to utilize the UPR, as a survival strategy and for progression [346]. Moreover, the UPR has a significant function in resistance to chemotherapy or radiation in tumor cells [347]. Numerous studies report that ER stress and activation of the UPR are related to pathological processes, like cancer, neurodegenerative diseases and cardiovascular disease [348,349,350].

The use of natural product-isolated compounds can be an encouraging therapy for different cancers [351,352] and modulation of ER stress can be an antineoplastic approach. Thus, the interaction between natural products and the antineoplastic impact through ER stress in malignant cells has been investigated in several studies [353].

#### 5.3.1. Alkaloids

Piperine (61) is an amide alkaloid isolated from the fruits of black and long pepper plants (*Piper nigrum* Linn and *Piper longum* Linn) [354]. Piperine (61) exposure generates ROS and triggers apoptosis by induction of ER stress in colon cancer HT-29 cells. This alkaloid promotes DDIT3 expression, which permeabilizes mitochondrial membranes and increases the expression of MAPK/JNK in HT-29 cells. Moreover, piperine (61) increases the expression of some proteins, which are related to ER stress like ERN1, DDIT3 and HSPA5 and activation of MAPK/JNK and MAPK14/p38. These findings indicate that piperine (61) triggers ER stress-mediated apoptosis in HT-29 cells [355].

According to the results of numerous investigations, camptothecin (5) is an important anti-cancer agent. Camptothecin (5) results in the accumulation of ER stress-controlling proteins, such as EIF2AK3, EIF2A, ATF4 and DDIT3 in androgen-sensitive prostate cancer LNCaP cells by enhancing the MYC expression and ROS generation. Furthermore, the transfection of *EIF2A*-targeted siRNA reduced triggering autophagy and diminished the levels of BECN1 and ATG7, indicating that camptothecin (5) upregulates ER stress-mediated autophagy. MYC results in the upregulation of UPR sensor protein expression in LNCaP cells, along with ROS generation, inducing autophagy through the Ca^2+^ and MAPK/JNK signaling pathways [145].

Irinotecan (62), another alkaloid, is a derivative of natural camptothecin (5). Irinotecan (62) individually or in conjunction with chemotherapy improves the survival rate of patients with advanced CRC [356]. The effect of this compound (62) on human CRC cell lines, LoVo and HT-29, is enhanced by curcumin (33) through induction of cell cycle arrest and apoptosis mediated via the production of ROS and stimulation of the ER stress pathway. The expression of HSPA5, DDIT3 and two markers of ER stress is attenuated by blocking ROS production [357].

Another camptothecin (5) derivative, hydroxycamptothecin (51), stimulates the expression of UPR sensor proteins such as EIF2AK3, ATF6, ERN1, HSPA5, DDIT3, BAX and MAPK/JNK and decreases the expression of BCL2. Moreover, according to quantitative real-time PCR, the mRNA levels of *DDIT3* and *HSPA5* are increased. According to the obtained results, the ER stress reaction and mitochondrial dysfunction are engaged in hydroxycamptothecin-triggered apoptosis in human Tenon’s capsule fibroblast cells [271].

These alkaloids are presented in Figure 6. The schematic role of alkaloids on the proteins engaged in the UPR and ER stress pathways is shown in Figure 18.

#### 5.3.2. Flavonoids and Phenolic Compounds

Epigallocatechin gallate (52) is a flavan-3-ol by-product extracted from green tea. The possible impact of epigallocatechin gallate (52) was investigated on HSPA5 in malignant mesothelioma (MMe) cell lines. According to qRT-PCR analysis, this compound (52) causes strong dose-dependent induction of *EDEM*, *ATF4* and *DDIT3,* as well as a limited induction of *HSPA5/GRP78* expression. Moreover, a significant elevation was found in the ratio of spliced to total *XBP1*. Moreover, western blot analysis of cell lysates demonstrated a significantly higher expression of HSPA5/GRP78, XBP1 and DDIT3. This compound results in the activation of CASP3 and CASP8. Epigallocatechin gallate (52) converts the constitutive UPR of MMe cells to pro-apoptotic ER stress [358].

Casticin (63) is a major component obtained from *Fructus viticis*. This compound (63) supports the sensitivity of tumor cells to TNFRSF10A by using multiple mechanisms. Western blot analysis was used to assess the cells for TNFRSF10A receptor expression. Casticin (63) enhanced TNFRSF10B expression, suggesting that TNFRSF10B upregulation is a potential mechanism via which this compound (63) increases the apoptotic impacts of TNFRSF10A in GC cells, BGC-823. Casticin (63) increases DDIT3, EIF2A and HSPA5 expression and acts as an ER stress inducer. Moreover, this compound upregulates TNFRSF10B receptors in MGC-803 and SGC-7901 cell lines via acting on the ROS-ER stress-DDIT3 pathway [359].

Ampelopsin (64) is one of the most important flavonoids found in *Ampelopsis grossedentata.* The expression of ER stress-dependent proteins, like HSPA5, p-ElF2A, spliced ATF6, DDIT3 and EIF2AK3 increases in breast cancer MCF-7 cells and MDA-MB-231, after treatment with ampelopsin (64). Moreover, this compound triggers ROS generation in both of these breast cancer cell lines [360].

Curcumin (33) was discussed above as an anti-cancer factor that can trigger cancer cell death in different cancer cell lines through different pathways. This compound induces ER stress and mitochondrial dysfunction, leading to apoptosis in AGS and HT-29 cell lines. The expression of DDIT3, phosphorylation of MAPK/JNK, CYCS, BAX and FADD are observed in HT-29 cells in the presence of curcumin (33) [359]. In 2013, the cytotoxicity effect of curcumin (33) was investigated in the murine myelomonocyte leukemia WEHI-3 cell line. Based on the increase of DDIT3, ATF6, ERN1, CASP12 and the anti-apoptotic protein BCL2, it was concluded that curcumin-triggered apoptosis is done via ER stress signaling pathways. Moreover, curcumin (33) increases ROS production and cytosolic Ca^2+^ discharge and induces DNA damage but decreases the level of mitochondrial membrane potential (ΔΨm) in WEHI-3 cells [361]. In 2017, it was shown that curcumin (33) promotes ER stress-mediated apoptosis in PC3 prostate cancer cells. According to the results, increased expression of ER stress-dependent proteins ERN1, PDI and CALR (calreticulin) together with a set of chaperones, play a key role to activate and promote a pro-apoptotic mechanism in PC3 cells. Moreover, the anti-cancer effect of curcumin (33) is related to ROS generation, UPR induction and autophagy [362]. Roberts et al. reported that curcumin (33) in interaction with sildenafil kills gastrointestinal (GI) cancer cells through ER stress, ROS and reactive nitrogen species (RNS). The induced ER stress signal is related to the formation of EIF2A, which is in part responsible for reducing MTORC1 and MTORC2 activity, increasing BECN1-dependent PtdIns3K activity, and enhancing autophagosome and autolysosome formation in an EIF2AK3-EIF2A-dependent manner [363]. Following curcumin (33) treatment, pro-apoptotic proteins BAD and BAX are upregulated and anti-apoptotic proteins BCL2, BCL2L1 and XIAP are downregulated. Moreover, ROS, Ca^2+^, ER stress and CASP3 are increased but mitochondrial membrane potential (ΔΨm) decreases [364]. Moreover, as mentioned, curcumin (33) alone or in combination with irinotecan (62) increases the expression of ER stress-associated proteins such as PDI, HSPA5 and DDIT3 in LoVo and HT-29 cells [357].

The anti-cancer activity of DMC (55) occurs through promoted expression of ER stress-associated proteins HSPA5, DDIT3, ERN1, ATF6A, ATF6B, CASP4 and CAPS12 in human lung cancer NCI-H460 cells. DMC (55) increases the level of ROS, and Ca^2+^ decreases the mitochondrial membrane potential (ΔΨm) and promotes the activities of CASP3, CASP8 and CASP9 [311]. Furthermore, BDMC (56), a curcuminoid, significantly induces apoptotic cell death. BDMC (56) increases ER stress-associated proteins expression such as HSPA5, DDIT3, ERN1, ERN2/IRE-1β, ATF6, ATF6B and CASP4, which results in cell apoptosis in the human lung cancer NCI H460 cell line [310].

WZ35 (57), a mono-carbonyl equivalent of curcumin (33), activates the ER stress pathway by activating the collection of ROS in the H1975 human NSCLC cell line [312]. Moreover, another analog of curcumin, (1E, 4E)-1, 5-bis (2-methoxyphenyl) penta-1,4-dien-3-one (referred to as B19) (65) is a new mono-carbonyl, which induces apoptosis through activation of ER stress and the autophagy signaling pathway in human ovarian cancer HO-8910 cells. The expression of the ER stress-related proteins, HSPA5, DDIT3, ATF6 and XBP1 and CASP3 increases in the cells treated with B19 (65) [359].

The powerful anti-cancer activity of the chalcone flavokawain B (66) isolated from *Alpinia pricei* Hayata was investigated in colon cancer HCT116 cells. Flavokawain B (66) exercises its apoptotic function via ROS formation and both *DDIT3* mRNA and DDIT3 protein upregulation, resulting in the regulation of the expression of BCL2 family members, leading to the induction of mitochondrial dysfunction and apoptosis. Moreover, pretreatment with the ROS scavenger NAC reduces the inhibitory impacts of flavokawain B (66), indicating that ROS production is necessary for ER stress-mediated apoptosis [365].

Resveratrol (32), as discussed previously, is a stilbenoid that shows antineoplastic activities toward different forms of cancers and blocks cancer development by focusing on diverse molecules and pathways engaged in developing cancer [366]. Some investigations demonstrated the ability of resveratrol (32) to trigger ER stress-dependent apoptosis in diverse cancer cell forms. For example, resveratrol (32) triggers ER stress-dependent apoptosis through suppression of the transcriptional activity of *XBP1* via SIRT1 in human multiple myeloma ANBL-6 cells. Resveratrol (32) suppressed expression levels of *VEGFA*, while expression levels of *PPP1R15A/GADD34* mRNA and *DDIT3* mRNA are vigorously triggered. This compound (32) also activates cell death signals through ERN1 stimulation of MAPK/JNK and EIF2AK3 and *ATF6* stimulation of *DDIT3*. Although ERN1 activation leads to increased XBP1s, resveratrol (32) suppresses the transcriptional activity of *XBP1*s, resulting in disabling cell survival functions and promoting cell death [367].

Resveratrol (32) triggers apoptosis in the malignant melanoma A375SM cell line by enhancing ER stress and ROS production, simultaneously. In the presence of resveratrol (32), the expression levels of MAPK14/p38, TP53 and BAX increase and the level of BCL2 decreases. Moreover, resveratrol (32) increases the levels of ER stress-related proteins p-EIF2A and DDIT3. Resveratrol (32) may also induce ROS generation and ER stress to enhance the apoptosis of melanoma cells [368].

Another investigation showed that resveratrol (32) enhances the toxicity of palmitate via ER stress-dependent apoptosis and increases palmitate-triggered ER stress in the human hepatoblastoma HepG2 cell line. Resveratrol (32) increases *XBP1* splicing and *DDIT3* expression but decreases ROS generation in palmitate-treated cells [369]. Bai et al. reported that resveratrol (32) demonstrates a pro-apoptotic function in *MIR200C*-positive cells. According to the analysis of the resveratrol (32) mechanism, ER stress-dependent proteins, such as RECK, a membrane-bound protein or ER stress molecules, such as HSPA5 and DDIT3, are increased in *MIR200C*-transfected cells. Collectively, based on these data, RECK increases the *MIR200C* expression, which sensitizes H460 cells to resveratrol (32). Thus, resveratrol and *MIR200C* act synergistically as anti-tumor agents in human lung cancer cells, NCI-H460 [370].

ER stress-mediated cell death induced by resveratrol (32), regulates the UPR events, non-canonical autophagy and caspases in two human nasopharyngeal carcinoma cell lines, NPC-TW039 and NPC-TW076. ER stress is a potent trigger of autophagy and resveratrol (32) induces autophagy, mediated by ER stress, as demonstrated by elevated expression of the UPR signaling molecules ATF6, DDIT3, ERN1 and p-EIF2AK3 [371]. Moreover, mitochondrial dysfunction and ROS-mediated ER stress are involved in apoptosis in human lung adenocarcinoma A549 cells, induced by a combination of resveratrol (32) and arsenic trioxide. The expression of ER stress-related including HSPA5, CASP12 and DDIT3 is increased in this cell line [372].

A powerful analog of resveratrol (32), (Z)3,4,5,4′-trans-tetramethoxystilbene (TMS), was investigated in human lung cancer A579 and H1975 cells. This compound induces caspase-independent apoptosis and autophagy by directly attaching to ATP2A/SERCA, a Ca^2+^ ATPase and leading to ER stress and AMPK activation. TMS remarkably enhances the cytosolic Ca^2+^ level. Moreover, the key members of the ER stress pathway, EIF2AK3, EIF2A and DDIT3 are upregulated [373]. Another resveratrol derivative, 3,5,4′-trihydroxy-trans-stilbene (Res-006), was reported to trigger ER stress, mediated by mitochondrial ROS in human hepatoma HepG2 cells. Co-administration of ER stress regulator 4-phenylbutyrate or the ROS inhibitor NAC significantly reduces Res-006-induced HepG2 cell death. The inhibitory effect of this compound on cell viability is more effective than resveratrol [374].

Xanthones are a group of polyphenolic complexes with a xanthene-9-one backbone [375]. A xanthone named α-mangostin (67), isolated from mangosteen (*Garcinia mangostana* L.), shows pro-apoptotic activity in the rat pheochromocytoma PC12 cells, which seems to be related to the ER stress pathway. Furthermore, this compound inhibits the ER ATP2A/SERCA Ca^2+^-ATPase significantly. A relationship is observed between the ATP2A/SERCA ATPase inhibitory and the apoptotic impacts of the xanthone by-product through the mitochondrial pathway by induction of CYCS. Conversely, α-mangostin (67) was reported to activate MAPK/JNK-MAPK/SAPK, which is one of the signaling molecules of ER stress [376]. Moreover, garcinone E (68), another xanthone found in mangosteen fruit, is able to trigger ER stress in ovarian cancer HEY, A2780 and A2780/Taxol cells and activates the ERN1 pathway. This compound (68) significantly enhances the protein expression levels of ERN1, XBP1, HSPA5, DDIT3 and CASP12 [377]. Moreover, gartanin (69), an isoprenylated xanthone of the mangosteen fruit, was reported to activate apoptosis through modulation of the expression of ER stress chaperones and markers in the human prostate cancer LNCaP and 22RV1cell lines. In this investigation, the ER stress pathway has shown a role in the degradation of AR (androgen receptor). Gartanin (69) reduces the AR expression in prostate cancer cells and suppresses AR function. Gartanin (69) modulates ER stress proteins including ERN1, HSPA5, DDIT3 and EIF2AK3. It was found that *DDIT3* knockdown partially restores gartanin (69)-triggered AR degradation, which proposes a potential interaction between UPR activation and AR signaling [378].

Honokiol (34), a biphenolic compound, was mentioned above as an autophagy inducer [233]. Honokiol (34) triggers apoptosis through CAPN2-mediated *HSP90B1/GRP94* cleavage in MKN45, SCM-1, AGS and N87 cells. Moreover, honokiol (34) increases PARP cleavage and DDIT3 (a protein that mediates ER stress-triggered apoptosis) levels in MKN45 cells. MKN45 and SCM-1 cells are more resilient to these reactions than AGS and N87 cells [379].

Chrysophanol (58) (1,8-dihydroxy-3-methyl-9,10-anthraquinone), a natural anthraquinone by-product, triggers ROS production and ER stress in breast cancer BT-474 and MCF-7 cell lines. This compound activates pro-apoptotic proteins and increases the deprivation of mitochondrial membrane potential (ΔΨm) and intracellular Ca^2+^ levels with the stimulation of UPR regulatory proteins such as DDIT3, EIF2AK3, EIF2A and ERN1. Furthermore, ER stress-dependent protein expression is reduced by co-administration of chrysophanol (58) and NAC in cells, which shows that ER stress and ROS production are major pathways in the pro-apoptotic function of chrysophanol (58) in both breast cancer cell lines [318].

Chemical structures of flavonoids and phenolic compounds are presented in Figure 10 and Figure 11. Figure 19 shows the schematic role of the flavonoids and phenolic compounds on the proteins involved in the UPR and ER stress pathways.

#### 5.3.3. Terpenoids

Garcinol (70), a polyisoprenylated benzophenone isolated from *Garcinia indica*, induces ROS generation and increases the level of both *DDIT3* mRNA and DDIT3 protein in human hepatocellular cancer Hep3B cells. Consequently, garcinol (70) results in downstream apoptosis activation with an elevation in the BAX:BCL2 ratio. Moreover, a reduction in the mitochondrial membrane potential (ΔΨm) and discharge of CYCS are detected. Moreover, the activation of CASP3, CASP9 and their target PARP is observed. Taken together, this compound activates not just the death receptor and the mitochondrial apoptosis pathways but also the ER stress modulator DDIT3 [380].

Polyphyllin D (71), a powerful cytotoxic saponin extracted from *Paris polyphylla,* induces a cytotoxic effect via a mechanism triggered by ER stress, accompanied by the mitochondrial apoptosis pathway. Polyphyllin D (71) treatment results in the upregulation of the mRNA levels of *HMOX1*, *ATF3*, *STC*2 (stanniocalcin 2) and *DDIT3*. According to western blot analysis, there were increases in HSPA5 and PDI proteins. Moreover, the expression of DDIT3 and CASP4 is increased upon polyphyllin D (71) treatment in an NSCLC cell line (NCI-H460) [381].

Saxifragifolin D (72) is an oleanane form of pentacyclic triterpenoid derived from *Androsace umbellate.* This compound induces ER stress in both breast cancer MDA-MB-231 and MCF-7 cells. Moreover, saxifragifolin D (72) increases the expression of DDIT3, MAPK/JNK and CYCS via activation of Ca^2+^ and ROS in the cytoplasm. In fact, according to the obtained results, saxifragifolin D (72) suppresses breast cancer cell development and initiates the interaction between autophagy and apoptosis via ROS-mediated ER stress [382].

Dehydrocostuslactone (73) is a sesquiterpene lactone isolated from the *Saussurea lappa* and *Aucklandia lappa.* Dehydrocostuslactone (73) triggers ER stress via EIF2AK3-DDIT3 and ERN1-MAPK/JNK signaling pathways in human NSCLC A549 and NCI-H460 cell lines, through induction of ROS in the cytoplasm. This compound causes an increase in cytosol calcium levels, EIF2AK3 phosphorylation, ERN1 and DDIT3 levels, *XBP1* mRNA splicing and CASP4 activity [383].

Cryptotanshinone (74) is a constituent of the stem of *Salvia miltiorrhiza Bunge*, which is shown to act as a powerful activator of ER stress. This compound results in the activation of ElF2A, HSP90B1/GRP94, HSPA5 and DDIT3 in HepG2 hepatoma and breast carcinoma MCF-7 cells. Moreover, cryptotanshinone (74) generates ROS, which has a critical function in ER stress-triggered apoptosis [384].

Pimpinelol (75), a linear sesquiterpene lactone isolated from *Pimpinella haussknechtii*, triggers ER stress in the human breast cancer MCF-7 cell line. This compound increases protein aggregation and mRNA expression of *ATF4*, *DDIT3*, *PPP1R15A/GADD34* and *TRIB3* (tribbles pseudokinase 3) [385].

Furanodiene (76) is a natural terpenoid derived from a well-known Chinese medicinal herb, named *Curcumae rhizome*. This compound increases DDIT3 expression at the gene and protein levels and induces both *HSPA5* mRNA and HSPA5 protein in lung cancer 95-D and A549 cells. Moreover, the synergetic activities of furanodiene (76) with paclitaxel (17) in NIH-H1299 and 95-D cells were reported [386].

Parthenolide (77) is a sesquiterpene lactone derived from the feverfew plant. This compound induces apoptosis through ER stress by stimulating EIF2A-ATF4-DDIT3 signaling in lung cancer Calu-1, A549, H1299 and H1792 cells. Parthenolide (77) also upregulates the expression of ATF4 and DDIT3 in a dose- and time-related mode. Knockdown of *DDIT3* reduces parthenolide-triggered TNFRSF10B and PMAIP1/NOXA expression and apoptosis. In addition, knockout of *ATF4* by siRNA reduces apoptosis, demonstrating that the expression of DDIT3 mediates parthenolide (77)-triggered apoptosis [387].

Zerumbone (78) is a bioactive sesquiterpene refined from the *Zingiber zerumbet* Smith [388]. Zerumbone (78) in combination with celecoxib causes the activation of the UPR pathway. Both compounds together activate the EIF2AK3-EIF2A arm of the UPR to induce the expression of ATF4, DDIT3, ATF3 and TNFRSF10B at the gene and protein levels, in both HCT116 *TP53* null and SW480 cells. Conversely, ROS scavengers abolish zerumbone-induced TNFRSF10B expression, which shows that the generation of ROS results in ER stress and activates the UPR in TP53-deficient human colon tumor cells [389].

The effect of a natural vitamin E derivative named α-tocopheryl succinate (79) was investigated on ER stress and the UPR and the apoptosis triggered by this compound in human GC cells, SGC-7901. α-Tocopheryl succinate (79) significantly increases the expression of HSPA5 and *DDIT3* and activates CASP4 and MAPK/JNK. This compound (79) triggers the expression of endogenous ER stress-associated HSP90B1/GRP94 at both protein and mRNA levels. Moreover, this compound triggers the activation of UPR components, including EIF2AK3, ATF6, *XBP1* and *ATF4* in a concentration- and time-related mode. In summary, α-tocopheryl succinate (79) induces apoptosis in human GC cells, which is mediated by ER stress and the UPR [390]. Moreover, α-tocopheryl succinate (79) activates *DDIT3*, HSPA5 and MAPK/JNK and upregulates the cleavage of CASP4 in SGC-7901 cells. Moreover, NAC decreases ER stress in SGC-7901 cells treated with this compound. Consequently, ER stress-mediated apoptosis induced by α-tocopheryl succinate (79) is related to ROS production [391].

4-nerolidylcatechol (80) is a catechol complex isolated from *Pothomorphe umbellata* L. This compound triggers apoptosis through ER stress both in SK-MEL-28, BRAfi- and MEKi-resistant melanoma cell lines. Increased HSPA5 and DDIT3 indicate the stimulation of the UPR in all melanoma cell lines. Moreover, *DDIT3* knockdown enhances cellular viability after administration of 4-nerolidylcatechol (80), demonstrating that apoptosis is related to DDIT3 [392].

7-acetylsinumaximol B (7-AB) (81) is a cembranoid initially isolated from aquaculture soft coral *Sinularia sandensis.* The apoptotic intrinsic pathway is stimulated in the 7-AB-induced cell death in the human GC cell line, NCI-N87. Moreover, 7-AB (81) triggers ER stress, leading to the stimulation of the EIF2AK3-ElF2A-ATF4-DDIT3 apoptosis pathway. Moreover, the expression of autophagy-dependent proteins is increased in NCI-N87 cells, in the presence of 7-AB (81). Taken together, by activation of the EIF2AK3-EIF2A-ATF4-DDIT3 pathway, 7-AB (81) induces ER stress-autophagy crosstalk in GC cells, NCI-N87 [393].

Tocotrienols are a subgroup of vitamin E, which shows a cytotoxic effect on breast cancer MCF-7 and MDA-MB-231 cell lines. γ-Tocotrienol (82) promotes DDIT3 and MAPK/JNK expression, cleavage of CASP3 and CASP8 and upregulation of TNFRSF10B at the gene and protein levels. Following the γ-tocotrienol (82) treatment, the level of HSPA5/GRP78 increased at the gene and protein levels. Activation of MAPK/JNK and MAPK14/p38, which is mediated by ER stress, increases the level of TNFRSF10B, whereas silencing *MAPK*/*JNK* and *MAPK14/p38* reduces the increase in TNFRSF10B and DDIT3 and partially blocks the apoptosis induced by γ-tocotrienol (82) [394].

Chemical structures of terpenoids with an effect on the UPR signaling pathway are represented in Figure 8. The schematic role of terpenoids on the proteins engaged in the UPR and ER stress pathway is shown in Figure 20.

#### 5.3.4. Other Natural Compounds

The anti-cancer function of garlic-based and garlic extracts has been investigated in different assays. An ethanol-based garlic extract stimulates apoptosis in several human cancer cells (DU145, U2OS and 67NR) through an ER stress pathway. Moreover, in support of elevated ER stress, an increased GST interaction with HSPA5 is detected [395]. Ajoene (83), an allyl sulfur compound found in crushed garlic, induces an ER accumulation of misfolded proteins and activates the UPR in human breast cancer MDA-MB-231 cells and human esophageal cancer WHCO1 cells. Ajoene (83) induces time-related expression of the ER stress sensor protein HSPA5 in MDA-MB-231 cells. The results show that this compound targets protein folding in the ER of cancer cells [396]. Moreover, BisPMB (84), a synthetic ajoene (83) analog, has superior cytotoxicity in WHCO1 cells, compared to ajoene (83). BisPMB (84) increases the expression of the ER stress sensor HSPA5 in WHCO1 cells. This compound also induces the expression of the ER stress marker DDIT3, which has a central function in the cytotoxic action of BisPMB (84) [397].

Polyphenon E^®^ is a standardized green tea extract. In the presence of polyphenon E^®^, the gene expression of ERS markers, *DDIT3 and* unspliced (*u)XBP1* but not *HSPA5* is triggered in human prostate cancer PC3 cells. While in PNT1a cells, gene expression of *HSPA5* and *DDIT3* was observed relative to control and no change in the level *(u)XBP1* was evident. The increased expression of *DDIT3* causes cells to resensitize to chemotherapy compounds. Moreover, treatment with polyphenon E^®^ increases the protein levels of EIF2AK3 targets specifically and p-EIF2A. Moreover, ATF4 is slightly upregulated. In PC3 cells, these proteins especially p-EIF2A is elevated significantly up to 48 h [398]. The chemical structures of natural compounds mentioned in this part with the effect on the UPR signaling pathway are presented in Figure 13. Moreover, these natural compounds are summarized in Table 3. The schematic role of these compounds on important proteins that participated in the UPR and ER stress pathways is represented in part C of Figure 14.

## 6. Animal Studies

As discussed above, the anti-tumor activities of different natural compounds are mediated through several cell signaling pathways. Numerous in vivo investigations have been carried out to assess the impact of these compounds on animals.

In 1997, the first sign of autophagy induction by rottlerin (38) was described in apoptosis-resistant rat and human gliomas. Rottlerin (38) decreases growth and induces cytotoxicity in the rat (C6) and two human glioma cells (U138MG and T98G) [399].

The daily intake of 1–5 mg/kg body weight of resveratrol (32) declined to inhibit the metastasis or growth of breast cancer in mice, even though in vitro results were promising. Resveratrol (32) has no impact on organ histology, body weight or estrous cycling of the tumor-bearing mice [400]. Conversely, another assay showed that oral administration of trans-resveratrol (32) (20 mg/kg twice per day or added to the drinking water at 23 mg/L) decreases hepatic metastatic intrusion of B16M cells, injected intrasplenically in rabbits, rats and mice. However, oral administration of this compound does not suppress the development of B16M, injected into the footpad of mice [401].

Tetrandrine (9) treatment in RT-2 gliomas in rats leads to slower development of subcutaneous tumors, longer survival time and increased survival grade; a high dose (150 mg/kg/day) is more potent than a low dose. Moreover, tetrandrine (9) is more effective against small subcutaneous gliomas than large ones. Furthermore, tetrandrine (9) affects intracerebral tumors and prolongs animal survival with no impact on the survival rate [402]. The anti-tumor effect of tetrandrine (9) is also seen in cultured mouse CT-26 colon cancer cells and when used subcutaneously. Animals treated early with tetrandrine (9) at a higher dose (150 mg/kg/day) live significantly longer. The higher dose of tetrandrine (9) and earlier treatment had more powerful impacts than a smaller dose (50 mg/kg/day) and delayed treatment. Moreover, the tumor size for the group administered with the higher dose of tetrandrine (9) was smaller than the group treated with a lower dose of this compound (9). Induced apoptosis in tetrandrine (9) treatment is suggested to be related to MAPK14/p38 signaling pathway activation [267].

Matrine (6) in nude BALB/c mice inhibits tumor development of pancreatic cancer cells in a dose-related mode by inducing apoptosis through stimulation of CASP3, CASP8 and CASP9, reducing the BCL2:BAX ratio and upregulating FAS. However, matrine (6) has no significant effect on the body weight of mice, compared to the controls [266].

Treatment with quercetin (25) (50 mg/kg daily intraperitoneal injection) results in apoptotic and ACD in gastric xenograft tumors in BALB/c mice and results in a remarkable reduction in tumor size. The average tumor size of quercetin-administered mice decreases by ~70%, in comparison to the vehicle-administered control mice. Moreover, the accumulation of LC3-II increases significantly in gastric xenograft tumors, in reaction to quercetin (25) intake [181].

Some compounds can synergistically induce autophagy. For instance, resveratrol (32) and spermidine (44) initiate autophagy via diverse mechanisms. These compounds activate converging pathways that peak in concurring alterations of the acetylproteome. Both factors support concurrent acetylation and deacetylation reactions in the nucleus and cytosol, respectively. Both resveratrol (32) and spermidine (44) trigger autophagy in cytoplasts (enucleated cells). At doses of 1/10 of the obtained optimal dose of these agents (resveratrol at 25 mg/kg or spermidine at 50 mg/kg), neither resveratrol (32) nor spermidine (44) stimulate autophagy alone. Nevertheless, the combined form of low doses of both factors is very effective in inducing autophagy in vivo in C57BL/6 mice. Comparable findings are acquired when these factors are administered to WT mice, as demonstrated using LC3 lipidation and SQSTM1 degradation [250].

Curcumin (33) suppresses tumor formation in intracranial glioma-initiating cells (GICs), following implantation into mice by inducing autophagy and increasing animal survival significantly. Nearly, all mice that received curcumin (33) endured the investigation course of 120 days, whereas all animals in the untreated groups died after three months [403]. The combined form of curcumin (33) and 5-fluorouracil (5-FU) induces an extremely strong anti-tumor function against esophageal squamous carcinoma, compared to the use of each treatment individually. All animals were exterminated on day 28 and the tumor weights were evaluated. The suppression rates in curcumin (33), 5-FU and combined curcumin (33) + 5-FU groups were 34.78%, 45.22% and 83.48%, respectively, in comparison to the control. Moreover, it was suggested that the inhibitory impact of curcumin (33) on the development of in vivo tumors could be via stopping the NFKB signaling pathway and triggering cell apoptosis [285].

Neoalbaconol (14) is a terpenoid that suppresses tumor development in the NPC nude mouse model by suppressing the PI3K-AKT-energy metabolic pathway. The tumor volumes in the neoalbaconol (14)-administered group were significantly smaller than those in the control group. Moreover, immunohistochemical examination of the tumor sections shows downregulation of phosphorylated AKT, MTOR and the HK2 expression in the neoalbaconol (14)-administered group [158].

In 2013, a substantial amount of rottlerin (38) was detected in the tumor and plasma of mouse xenografts fed with this compound based on RP-HPLC. In addition, rottlerin (38) was efficiently absorbed in the cells and tissues both in vivo and in vitro in pancreatic cancer cells. The obtained results show that mice fed with rottlerin (38) grow smaller tumors than the control group [404].

Mice with tumor xenografts of 100 mm^3^ and treated with salvianolic acid B (40) (80 mg/kg, intraperitoneal injection, once daily, for 28 days), displayed a strong anti-tumor activity against CRC. A combined mix of the autophagy inhibitor 3-MA and salvianolic acid B (40) significantly increases tumor growth, compared to salvianolic acid B (40) alone, indicating that this compound induces pro-death autophagy in vivo. Moreover, no significant hepatic toxicity or weight loss is found in the group of salvianolic acid B (40) or the combined group of 3-MA and salvianolic acid B (40) [245].

In 2017, the anti-tumor effect of trehalose (46) was investigated on mice-bearing Ehrlich ascites carcinoma (EAC). Treatment with this disaccharide induces antineoplastic impacts toward EAC, as demonstrated by a significant decrease in tumor volume, body weight, number of viable tumor cells and the *BCL2* expression as the anti-apoptotic gene. Trehalose (46) also significantly increases life span, average survival time and expression of the apoptotic gene *CASP3*. Moreover, co-administration of trehalose (46) and methotrexate, normally applied as standard chemotherapy for several cancers, shows the maximum antineoplastic effect on EAC [405].

## 7. Clinical Trials

The first natural compounds to progress into clinical settings were vinblastine (1) and vincristine (3). These compounds belong to the alkaloids family and are utilized by different cultures for the cure of diabetes [406]. Vindesine (4) has similar effects to vinblastine (1) and the anti-cancer function of vindesine (4) has been stated in blast crisis of chronic myeloid leukemia, acute lymphocytic leukemia pediatric solid tumors, malignant melanoma and metastatic renal, breast, esophageal and colorectal carcinomas [133]. In a phase II trial, 50 patients with progressed solid tumors were attended to, 43 of whom had obtained at least one earlier chemotherapy treatment, to investigate the effect of vinblastine (1) (3 mg/m^2^) weekly and rofecoxib (25 mg) daily. The median progression-free survival for all the patients was 103 days and for the subjects who received the clinical treatment, was 289 days. Moreover, the results from this trial show that the combined intake of daily cyclophosphamide and weekly vinblastine (1), administered simultaneously with daily rofecoxib, provides a medium anti-cancer function in a group of patients with progressed solid tumors. The admissibility of this treatment is excellent. In addition, major adverse reactions were not notable, indicating a favorable therapeutic index [407].

Cabazitaxel (85) (XRP6258) is a paclitaxel (17) derivative that was confirmed by the US Food and Drug Administration (FDA) in 2010 as the first therapy for the hormone-refractory metastatic prostate cancer [408]. In the phase I clinical study, the primary evidence of anti-cancer activity of cabazitaxel (85) was observed. Twenty-five patients with progressive solid malignancies were prescribed 102 courses of cabazitaxel (85) at four dose levels, spanning 10 to 25 mg/m^2^. Three patients obtained incomplete reactions, including two patients with castration-resistant prostate cancer (CRPC), one of whom had formerly taken docetaxel, but two patients had a diseased state beyond 4 months. Moreover, according to the results, the suggested phase II dose of this compound was 20 mg/m^2^ [409]. A phase II study was conducted on 71 patients with metastatic breast cancer to assess the dose of cabazitaxel (85) for the phase III trial. Intravenous (i.v.) cabazitaxel (85) was administered at 20 mg/m^2^ every 3 weeks. The median overall survival was 12.3 months and the median time to progression was 2.7 months after a follow-up of 20 months. Patients experienced hypersensitivity reactions (6%), vomiting (18%), sensory neuropathy (17%), leucopenia (55%), diarrhea (30%), nausea (32%), fatigue (35%) and neutropenia (73%). Conversely, 18 patients (30%) had a medical condition for at least 3 months. The objective response rate was 14% in the treated patient population with two complete and eight partial responses [410].

An open-label randomized phase III trial was carried out to examine the efficacy and safety of cabazitaxel (85) in men with metastatic CRPC, who were under hormone treatment with docetaxel but had progressive disease during and post-treatment. A total of 755 patients (377 mitoxantrone, 378 cabazitaxel [85]) were prescribed 10 mg oral daily prednisone and were arbitrarily appointed to take either 12 mg/m^2^ mitoxantrone intravenously for 15–30 min or 25 mg/m^2^ cabazitaxel (85) intravenously, for 1 h every 3 weeks. The most frequent clinically significant grade 3 or greater adverse reaction was neutropenia (82%). The non-hematological toxicities seen in the cabazitaxel (85) arm were peripheral neuropathy, asthenia, fatigue and diarrhea. Therefore, therapy with cabazitaxel (85) plus prednisone was concluded to be clinically effective and improved overall survival in the patients with metastatic CRPC, with progressive disease in and post-docetaxel treatment [411].

The increased expression of *ABCB1/MDR1*, encoding an ATP-dependent drug efflux pump, reduces the intracellular concentration of taxanes and results in resistance to these drugs. The poor affinity of cabazitaxel (85) to ABCB1 because of the presence of extra methyl groups makes this compound superior to paclitaxel (17). Thus, cabazitaxel (85) seems to be efficacious in docetaxel-resistant cancers. Cabazitaxel (85) shows the anti-tumor function in the preclinical and phase I, II and III clinical studies in docetaxel-resistant cancers [412]. Cabazitaxel (85), as a third-generation taxane, is suggested for the treatment of metastatic CRPC [413].

TPI 287 (86) is a synthetic taxane by-product. In contrast to taxane, TPI-287 (86) is permeable through the blood–brain barrier [414]. A phase I study of TPI 287 (86), combined with temozolomide for patients with metastatic melanoma, was conducted; a total of 21 patients aged 15 years or older with histologically approved metastatic melanoma participated in the trial. There were two partial reactions and seven diseased states (9.5% overall reaction rate and 42.9% disease control rate). Three patients had brain disease despite the advanced extra-cranial disease. The highest acceptable dose of the blend was 125 mg/m^2^ (i.v.) of TPI 287 (86) and 110 mg/m^2^ of oral temozolomide. The rate-limiting toxicity was neuropathy and five patients had grade III neuropathy [415].

Camptothecin (5) progressed to clinical trials by the National Cancer Institute in the 1970s, but it was not approved because of serious bladder toxicity. Later, more effective derivatives of camptothecin (5), topotecan (87) and irinotecan (62) (CPT-11; Camptosar) were developed through extensive research. Camptothecin (5) by-products have demonstrated anti-cancer function toward a broad spectrum of malignancies. Topotecan (87) is applied for the therapy of small cell lung and ovarian cancers, while irinotecan (62) is employed for the therapy of CRCs [4]. Veliparib (88) was supplemented with the weekly topotecan (87) treatment for 58 patients with solid tumors. The results showed that hematological toxicity was dose-limiting but both drugs could be administered at three-fourths of their single treatment with the highest acceptable doses. The goal reaction rate was 10% with four partial responses and one complete response. Twenty-two patients (42%) had a diseased state, spanning 4 to 26 cycles. This treatment showed a compliant safety profile and exhibited signals, indicative of possible action [416]. In another clinical trial, adult patients (>19 years; median age of 61 years) with relapsed/refractory small cell lung cancer received a combination of topotecan (87) and doxorubicin. The results showed hematological adverse reactions as the most frequent side effects. Moreover, no therapy-related grade 5 toxicities were found. This combination seems to be safe in subjects with relapsed small-cell lung cancer and it needs to be further investigated.

A combination of topotecan (87) (1.35 mg/m^2^) on days 1–5 of a 3-week cycle and weekly doxorubicin (20 mg/m^2^) starting on day 6 was proposed as a safe dose for future trials. The reaction rate was 20% (4/20); median progression-free survival and overall survivals were 3.6 months and 6 months, respectively [417]. Gimatecan (89) is a lipophilic oral camptothecin (5) analog that has been used for a clinical trial for recurrent glioblastoma. In phase I of the clinical trial, oral gimatecan (89) was administered for 5 consecutive days every 28-day cycle in ascending doses. The dose was increased using a modified Fibonacci method. Weekly intake of oral gimatecan (89) results in ongoing systemic vulnerability to possible efficacious concentrations of the dynamic complete lactone type of the drug. The subjects under this treatment did not experience significant alopecia or diarrhea. The frequency and severity of the hematological toxicities appear to increase with dose escalation [418]. In 2013, a multicenter phase II trial was conducted to assess the efficacy of gimatecan (89) in adults with recurring glioblastoma. In this trial, subjects taking enzyme-triggering anti-seizure drugs were ruled out. According to the results, 6-month progression-free survival for the subjects with recurring glioblastoma undergoing gimatecan (89) monotherapy, given on a 5 days every 28 days cycle, was not remarkably superior to the historical 6-month progression-free survival for the subjects with recurring glioblastoma who did not take the treatment. Therefore, gimatecan (89) in this treatment regimen showed minimal efficacy [419].

Other clinical trials exploring the effectiveness of camptothecin (5) derivatives in patients with high-grade gliomas were disappointing. Karenitecin (90) is another lipophilic oral camptothecin (5) analog of preclinical function in glioma models. The effect of karenitecin (90) was investigated in subjects with recurrent malignant gliomas. The data showed no objective response in 11 patients treated at or above the maximum tolerated doses who had previously received only one or a few prior chemotherapy regimens. Thirty-two patients (median age 52 years; median Karnofsky performance scale (KPS) score of 90) were collected. Among them, 78% had glioblastoma and 22% had anaplastic gliomas. Moreover, with a response rate of less than 2%, it appears that karenitecin (90) given on this schedule is not efficacious for the treatment of subjects with recurrent malignant glioma [420].

Irinotecan (62) or camptothecins-11 is another camptothecin (5) derivative. A phase I study was initiated to establish the highest permitted dose, major toxicities and pharmacokinetics of irinotecan (62). Sixty-four patients, fulfilling the standard phase I admission requirements were enrolled (median age 51 years; primary sites: head and neck, colon, lung and pleura; 60 of 64 had been formerly cured). Grade 3 to 4 non-hematological toxicities including nausea and vomiting (9%), diarrhea (16%; three hospitalizations), alopecia (53%), asthenia (14%), the elevation of hepatic transaminases (8%), and one case of skin toxicity were observed. The maximum-tolerated dose was given as a 30-min i.v. infusion every 3 weeks at 600 mg/m^2^. For safety reasons, the appropriate dose was presented as 350 mg/m^2^ every 3 weeks. Nonetheless, when close observation of GI toxicities is feasible, a large dose of 500 mg/m^2^ can be prescribed in subjects with acceptable risk [421]. The therapeutic capacity of irinotecan (62) was investigated for the therapy of metastatic CRC in two large, randomized phase III trials. The effect of the combination of irinotecan (62) and FU-leucovorin calcium (LV) was compared with FU-LV individually. Considerable effectiveness was shown for the combined irinotecan (62) and FU-LV concerning reaction rate, the median time to disease progression and median survival time. Thus, the combined FU-LV and irinotecan (62) was confirmed as the forefront and reference therapy for metastatic CRC [422]. In 2003, a multi-center phase II and the pharmacokinetic study examined the impacts of irinotecan (62) (350 mg/m^2^ every 3 weeks), either prior to (group A) or after relapse following radiotherapy (group B), in chemotherapy-naïve subjects with glioblastoma. Fifty-two patients (25 patients in group A and 27 patients in group B) were given 191 cycles of irinotecan (62). The used dose of irinotecan (62) showed small clinical function as a single regimen in subjects with recently diagnosed and reoccurring glioblastoma, following radiotherapy. This dose of irinotecan (62) resulted in a median time-to-progression of 9 weeks in group A and 14.4 weeks in group B, with 6-month progression-free survival rates of 26% and 43% in groups A and B, respectively. Although the response rate of irinotecan (62) was a limiting factor, it is still an attractive drug for combination therapy against glioblastoma [423].

In 2004, irinotecan (62) was again investigated in a phase II study in adults with reoccurring malignant glioma. The reaction rate in this study was 6% (1/18). One patient out of 18 patients had a complete response, 28% (5/18) of these subjects progressed in health condition while taking irinotecan (62), which means 5 patients showed radiographic progression. Moreover, 5 subjects had stable disease, 6 patients were excluded from the investigation because of the toxicity and 1 patient declined additional treatment. According to the results, irinotecan (62) represented minimal efficacy in this investigation [424].

Natural compounds use different mechanisms to affect the autophagy pathway and the distinction between the survival-supporting and/or death-promoting roles of them on autophagy process need more deep study for therapeutic response. For example, magnolol (35) that can induce autophagy can affect the morphological and cellular events such as ATP level, cell blebbing and DNA fragmentation without leading to cell death in itself [425]. Most of the natural compounds mentioned here for clinical trials are alkaloids. As mentioned previously, alkaloids like vinblastine (1) and vincristine (3) act as autophagy inhibitors [128,142], whereas camptothecin (5), another alkaloid, was reported to induce autophagy [144]. Paclitaxel (17), an important anticancer agent, is a diterpenoid. This natural compound inhibits the progression of cervical cancer by inhibiting autophagy [167]. However, cabazitaxel (85) is a derivative of paclitaxel (17) that has been used in different phases of clinical trial. This compound was reported as an autophagy [426].

The chemical structures of natural compounds that have been used in clinical trials are shown in Figure 21.

## 8. Drawbacks of Natural Compounds

Although natural compounds were proved to represent promising anti-cancer activities against different types of cancers, some disadvantages should be considered. Paclitaxel (17), Taxol, is an important anti-cancer agent. Despite several clinical successes [427], there is a major issue with the administration of Taxol, which is the availability of enough drugs for therapy [428,429]. Only 2 mg of this complex can be extracted from each adult yew plant, while each cancer subject requires 2.5 to 3 g of Taxol [430]. Moreover, according to some studies, Taxol meets the threshold to trigger considerable mitotic arrest and apoptosis in tumor cells because the intra-tumoral concentration of paclitaxel (17) is significantly smaller than that in serum. Moreover, resistance to paclitaxel (17) is observed in some cancer cells, with unknown underlying molecular mechanisms [431]. Vinblastine (1), another natural compound with anti-cancer activities, has been used for the chemotherapy of human neuroblastoma, but it shows dose-limiting side effects [432].

One study indicated that trace levels of curcumin (33) and its by-products are found in the peripheral circulation and tissues far from the GI tract of patients with CRC who ingested curcumin (33), which could be due to its low systemic bioavailability [433]. However, some work has shown that quercetin (25) can enhance the bioavailability of curcumin (33) by increasing its uptake into human carcinoma cells, which might be related to an ALB (albumin)-binding interaction [434]. On a cautionary note, a comprehensive review performed by Nelson et al. in 2017 challenged the efficacy of curcuminoids, according to more than 100 failed human clinical trials for several disease states. Therefore, curcuminoids relatively in the natural proto-form may not be adequately pharmacologically active. It has been pointed out that no form of curcumin (33) seems to have the properties essential for a good drug candidate [435].

Moreover, genistein (27) was shown to trigger autophagy and apoptotic death in tumor cells. According to absorption, distribution, metabolism and excretion/ADME studies, this compound has intrinsically small oral bioavailability due to metabolic enzymes and efflux transporters. Thus, further investigations are needed to ameliorate its effectiveness in the therapy of resilient cancers [436].

Some compounds consumed in the normal diet may not achieve fairly high concentrations to moderate pharmacological impacts. For example, concentrations of neither resveratrol (32) nor spermidine (44) reach the required level to exhibit pharmacological effects [250].

Clinicians should be conscious that high doses of resveratrol (32) necessary for the most therapeutic impacts may not be used in the clinical setting, as was demonstrated in a population study [437]. The use of resveratrol (32) was shown again to be constrained by its low bioavailability and inadequate solubility [438,439].

Camptothecin (5), another natural compound with an anti-cancer effect, is also limited to treating cancers because of poor solubility and severe side effects. Therefore, many modifications and derivatives have been developed [440].

## 9. Conclusions

Autophagy is one of the most intriguing biological “machineries” that has emerged from biological research in recent decades. In fact, this process plays several critical roles in modulating principal features of tissue and cell metabolism, controlling their development and homeostasis, and enhancing their potential to respond to stress conditions. The complexness of the mechanisms involved in autophagy and several factors and effectors that participated in this process have allowed us to hypothesize that autophagy can be pharmacologically modulated, and that this pharmacological intervention may depict a novel medicinal approach for a broad set of oncological, cardiovascular, metabolic, immune, inflammatory, neurological, and even infectious diseases. Indeed, by virtue of the intensive attempts that have currently been conducted by biomedical investigation in this field, we currently understand that autophagy is indeed a “druggable process”. Consistently, many molecules that act as inhibitors or promoters on different levels of the “autophagic machinery” are the object of intense drug discovery studies. Certainly, the pharmacological modulation of a process that plays such a ubiquitous and capillary homeostatic role is not devoid of serious toxicological concerns. Therefore, looking for the molecules that can indicate selectivity for the specific molecular targets involved in autophagy and/or for tissue-specific drugs is a difficult and timely issue. However, it is worth mentioning that numerous medications are only under examination in reassuring clinical investigations [441].

Among various molecules with recognized modulatory effects on different molecular targets involved in autophagy, a very remarkable section is depicted by the compounds of biological sources, which have often given us important contributions to understanding the biological meaning and mechanisms of autophagy. Alongside their importance in basic research, these compounds could demonstrate an important pharmacological means to be applied in therapy but could also offer a valuable pattern for the development of novel molecules with ameliorated pharmacodynamic and/or pharmacokinetic features, aimed at making autophagy an increasingly appropriate, effective and safe drug target. Hopefully, in the coming years, selective modulators of autophagy and related pathways based on natural compounds, which are able to activate or inhibit the process only in the target tissues, will provide effective and safe therapeutic options for numerous diseases.

## Figures and Tables

**Figure 1 cancers-14-05839-f001:**
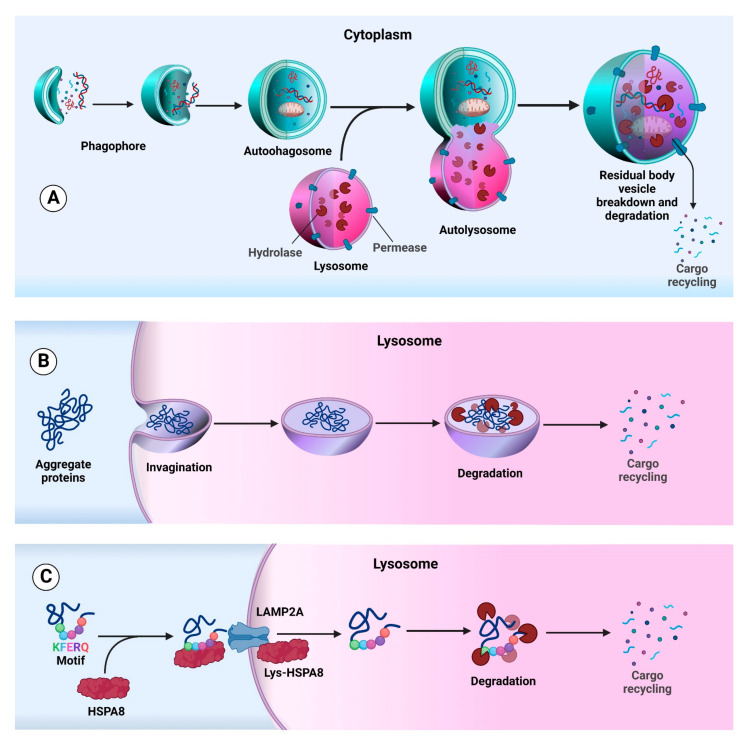
Schematic illustration of the main distinct autophagy mechanisms in mammals. (**A**) macroautophagy (autophagy), which includes initiation, phagophore expansion, autophagosome fusion with the lysosome, developing an autolysosome and cargo degradation steps; (**B**) microautophagy, in which the lysosome takes up small soluble particulates by invagination or protusion, followed by scission of the lysosomal membrane; and (**C**) CMA, which is a selective mechanism for proteins with specific amino acid motif (KFERQ sequence). The figure was created with BioRender.com. Licensing Right NX24OB5K3I, 21 November 2022.

**Figure 2 cancers-14-05839-f002:**
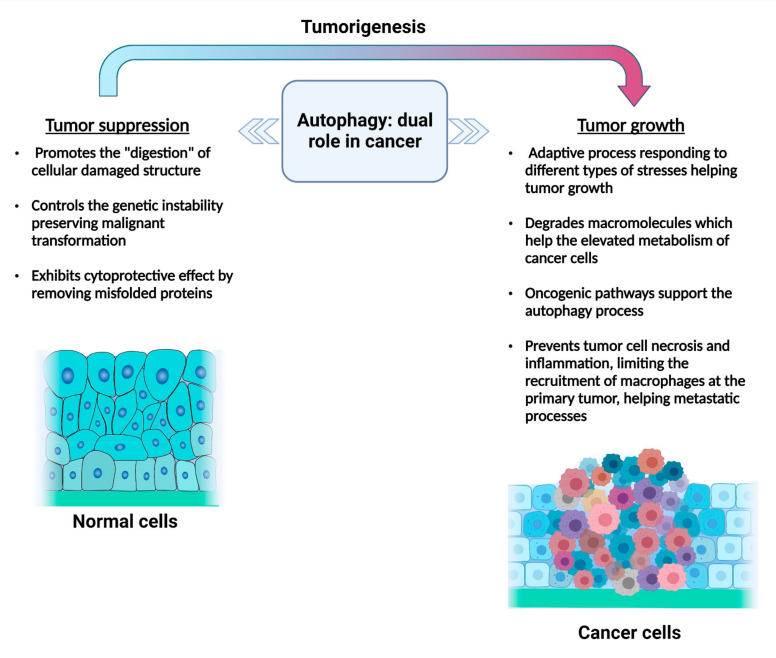
Dual function of autophagy in cancer. The dualistic function of autophagy in cancer. Autophagy in the initial stages of cancer promotes tumor suppression by digesting cellular damaged structures. In contrast, in the late stages of cancer, autophagy sustains cancer cell growth by supporting elevated metabolism and by limiting the recruitment of macrophages. The figure was created with BioRender.com. Licensing Right GD24OB603C, 21 November 2022.

**Figure 3 cancers-14-05839-f003:**
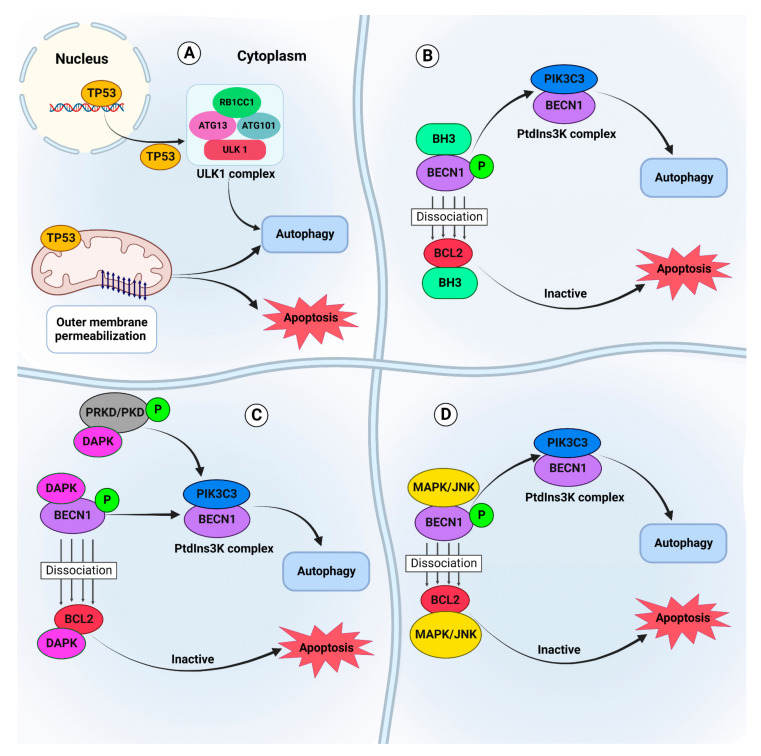
The crosstalk between signals that induce both apoptosis and autophagy. (**A**) The tumor suppressor TP53 has both pro-apoptotic and pro-autophagic functions depending on its subcellular localization. (**B**) BH3 (BCL2 homology domain 3) proteins interact with BECN1 and BCL2 to activate autophagy and apoptosis. (**C**) DAPK (death associated protein kinase) can stimulate autophagy and apoptosis by phosphorylating BECN1, driving its dissociation from BCL2. DAPK can also activate PRKD/PKD, which phosphorylates and activates PIK3C3/VPS34. (**D**) MAPK/JNK induces autophagy and apoptosis by phosphorylating and inactivating BCL2 and activating the BECN1-PIK3C3/VPS34 complex. The figure was created with BioRender.com. Licensing Right DV24OB65T6, 21 November 2022.

**Figure 4 cancers-14-05839-f004:**
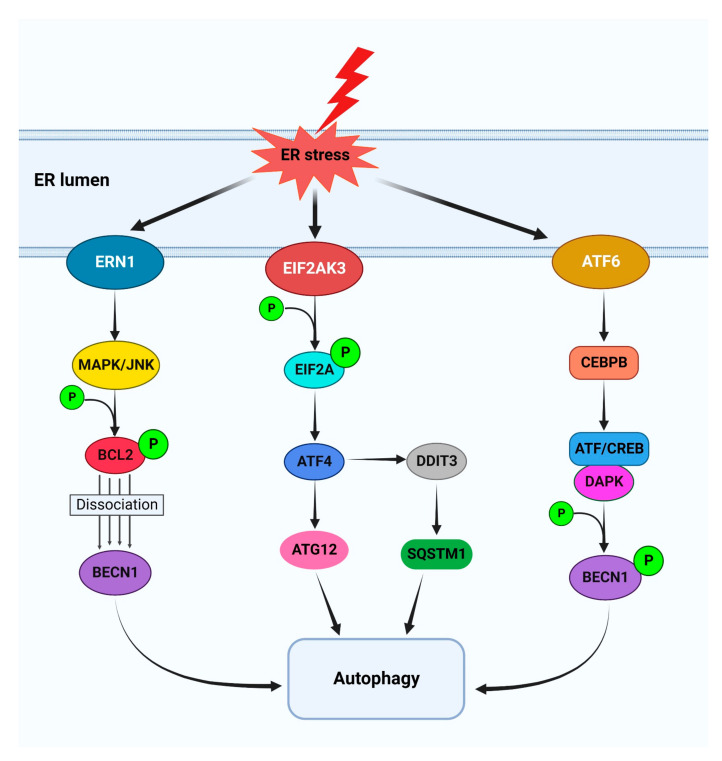
The interplay between ER stress (UPR) and autophagy. UPR signaling pathways and ER stress activate autophagy. The ERN1 arm of the ER stress response causes MAPK/JNK activation and phosphorylation of BCL2, which stimulates its dissociation from BECN1, leading to autophagy activation. Increased phosphorylation of EIF2A by EIF2AK3, another arm of the ER stress reaction, in response to distinct ER stress factors can result in autophagy via an ATF4-related increase in the ATG12 expression. The EIF2A-ATF4-DDIT3 axis can promote the SQSTM1 expression and autophagy induction. ATF6, the third arm of the ER stress reaction, activates autophagy not only through CEBPB, induction of DAPK1 expression and BECN1 phosphorylation, but also through the ATF4-DDIT3-SQSTM1 pathway. The figure was created with BioRender.com. Licensing Right JA24OB6AMR, 21 November 2022.

**Figure 5 cancers-14-05839-f005:**
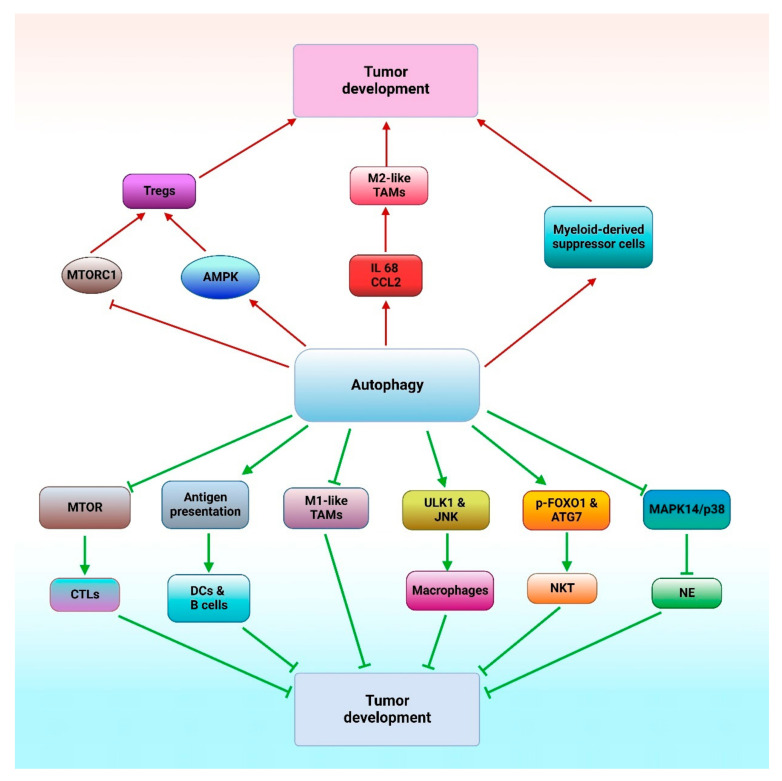
The relationship between autophagy and immune cells. Autophagy activation can increase or prevent tumor development by modulating the immune system. Autophagy inhibits tumor development through MTOR inhibition, antigen presentation, M1-like TAM polarization, ULK1 and MAPK/JNK activation, phosphorylation of FOXO1 working with ATG7 and inhibition of MAPK14/p38. Autophagy can promote tumor development by M2-like TAMs, regulatory T cells (Tregs) and myeloid-derived suppressor cells. The figure was created with BioRender.com. Licensing Right RN24OB6GKQ, 21 November 2022.

**Figure 6 cancers-14-05839-f006:**
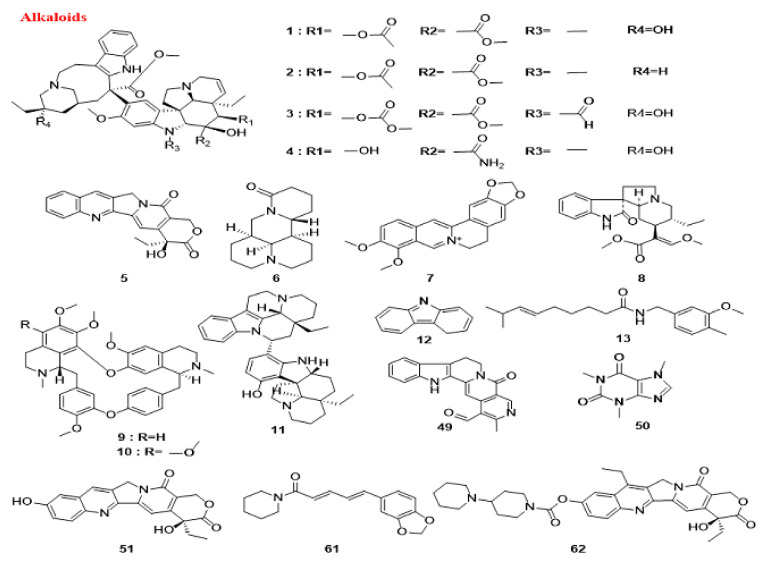
Chemical structures of alkaloids. Vinblastine (1), vinorelbine (2), vincristine (3), vindesine (4), camptothecin (5), matrine (6), berberine (7), isorhynchophylline (8), tetrandrine (9), hernandezine (10), bisleuconothine A (11), carbazole (12), capsaicin (13), subditine (49), caffeine (50), hydroxycamptothecin (51), piperine (61), irinotecan (62). The images were prepared using ChemDraw Professional 15.0.

**Figure 7 cancers-14-05839-f007:**
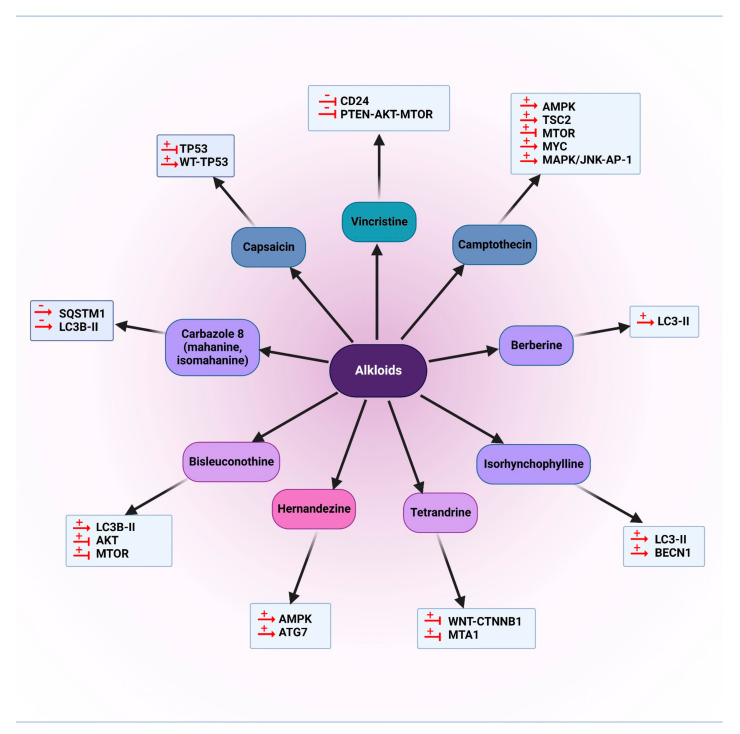
The schematic role of alkaloids on different proteins involved in the autophagy pathway. Alkaloids regulate autophagy in different ways. Vincristine and carbazoles have been demonstrated to inhibit autophagy by suppression of CD24, suppression of the PTEN-AKT-MTOR pathway and enhancement of SSQTM1/p62. Other alkaloids induce autophagy via activation of AMPK, TSC2, MAPK/JNK-AP-1 and BECN1, and enhance MYC/c-Myc, LC3-II and ATG7 expression. Moreover, inhibiting the AKT-MTOR pathway has a key function in induced autophagy by alkaloids. Moreover, another alkaloid named tetrandrine induces autophagy by suppressing the WNT-CTNNB1/β-catenin pathway and decreasing MTA1 expression. →: Indicates activation or an increase in the mentioned protein expression. 
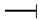
: Indicates deactivation or a decrease in the mentioned protein expression. − and +: Indicate the effect of the proteins on different pathways (− means negative effect and + means positive effect). The figure was created with BioRender.com. Licensing Right VD24OB6Q8E, 21 November 2022.

**Figure 8 cancers-14-05839-f008:**
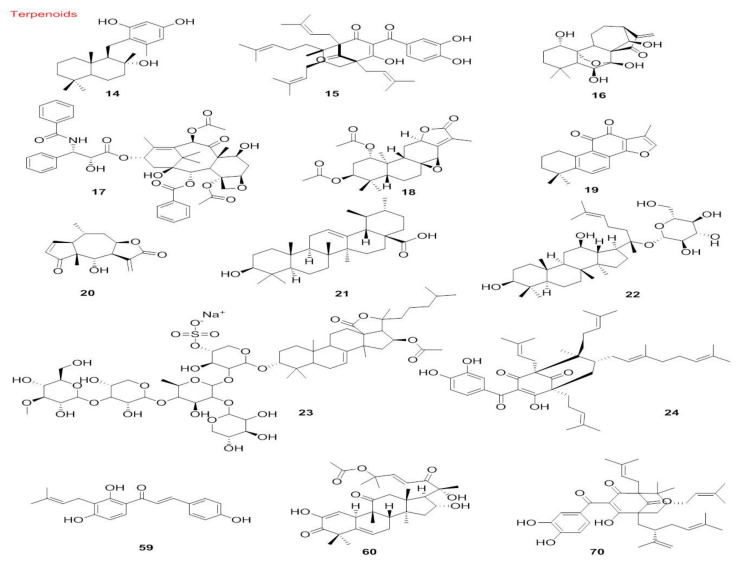
Chemical structures of terpenoids. Neoalbaconol (14), guttiferone K (15), oridonin (16), paclitaxel (17), gelomulide K (18), tanshinone IIA (19), helenalin (20), ursolic acid (21), ginsenoside compound K (22), frondoside A (23), oblongifolin C (24), isobavachalcone (59), cucurbitacin B (60), garcinol (70), polyphyllin D (71), saxifragifolin D (72), dehydrocostuslactone (73), cryptotanshinone (74), pimpinelol (75), furanodiene (76), parthenolide (77), zerumbone (78), α-tocopheryl succinate (79), 4-nerolidylcatechol (80), 7-acetylsinumaximol B (7-AB) (81), γ-tocotrienol (82). The images were prepared using ChemDraw Professional 15.0.

**Figure 9 cancers-14-05839-f009:**
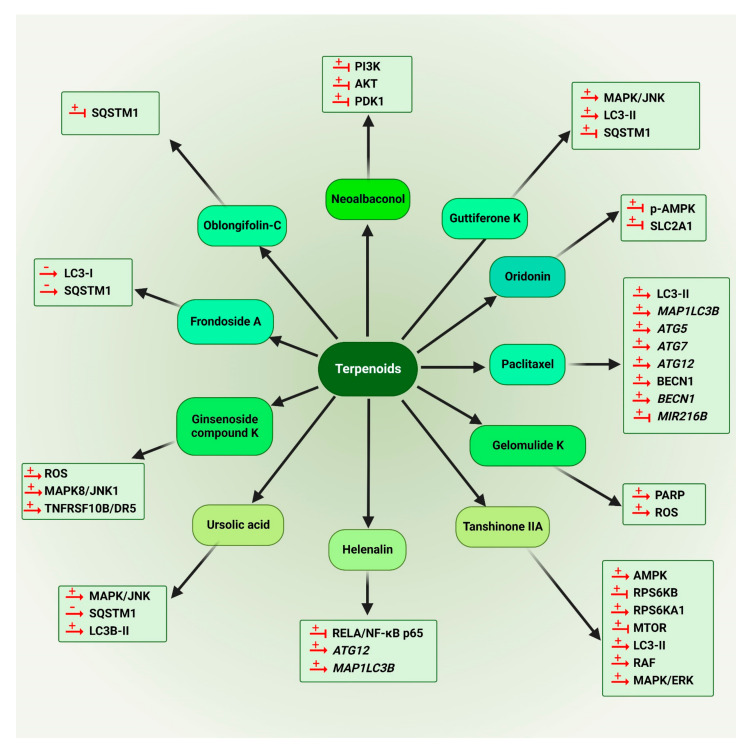
The schematic role of terpenoids on the proteins involved in the autophagy pathway. Terpenoids induce autophagy via different signaling pathways such as MAPK/ERK, PI3K-AKT-MTOR, MAPK/JNK, NFKB, RAF, RPS6KA1/p90RSK, RPS6KB/p70S6K and reactive oxygen species generation. Moreover, these compounds upregulate the expression of some proteins such as LC3-II and BECN1, ATG genes and TNFRSF10B/DR5 while suppressing the expression of RELA/NF-κB p65, SQSTM1, SLC2A1/GLUT1 and *MIR216B*. However, accumulation of SQSTM1 is observed in the treatment with ursolic acid and frondoside A. →: Indicates activation or an increase in the mentioned protein expression. 
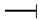
: Indicates deactivation or a decrease in the mentioned protein expression. − and +: Indicate the effect of the proteins on different pathways (− means negative effect and + means positive effect). The figure was created with BioRender.com. Licensing Right AI24OB6U1K, 21 November 2022.

**Figure 10 cancers-14-05839-f010:**
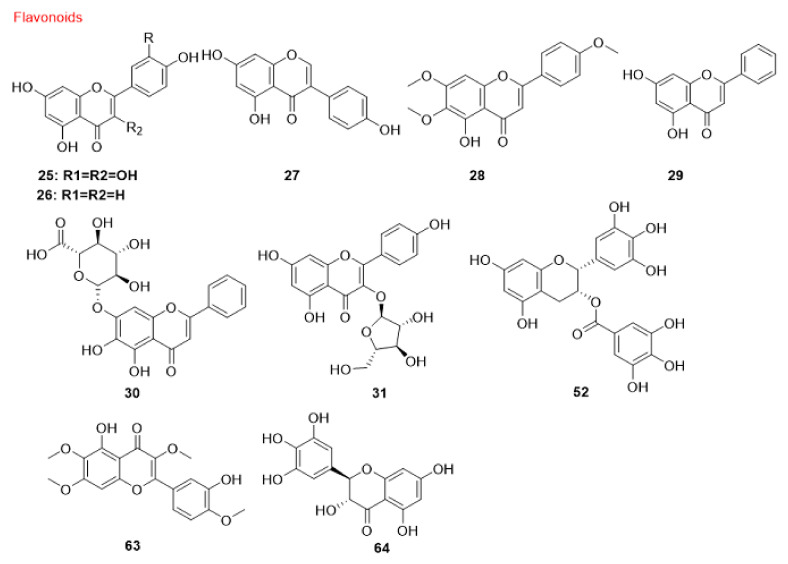
Chemical structures of flavonoids. Quercetin (25), apigenin (26), genistein (27), salvigenin (28), baicalin (30), juglanin (31), epigallocatechin gallate (52), casticin (63), ampelopsin (64). The images were prepared using ChemDraw Professional 15.0.

**Figure 11 cancers-14-05839-f011:**
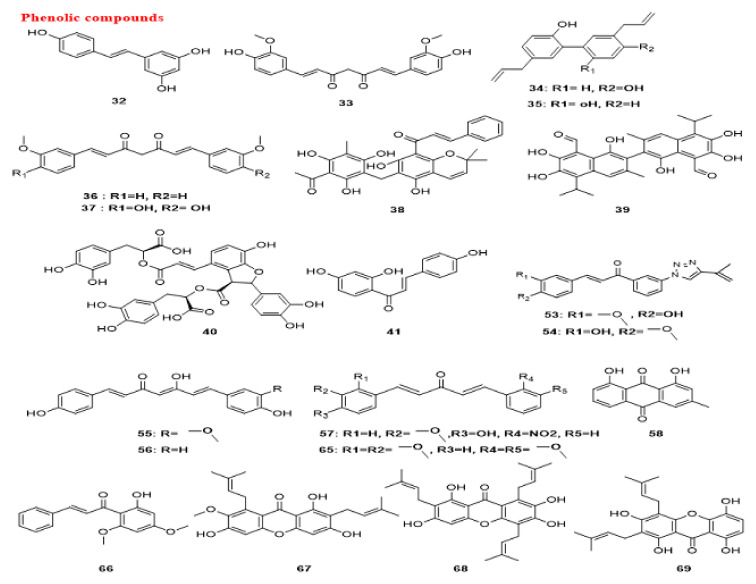
Chemical structures of phenolic compounds. Resveratrol (32), curcumin (33), honokiol (34), magnolol (35), bis-dihydroxy-curcumin (36), tetrahydrocurcumin (37), rottlerin (38), gossypol (39), salvianolic Acid B (40), isoliquiritigenin (41), epigallocatechin gallate (52), triazolyl curcumins (53, 54), demethoxycurcumin (DMC) (55), bisdemethoxycurcumin (BDMC) (56), WZ35 (57), chrysophanol (58), B19 (65), flavokawain B (66), α-mangostin (67), garcinone E (68), gartanin (69). The images were prepared using ChemDraw Professional 15.0.

**Figure 12 cancers-14-05839-f012:**
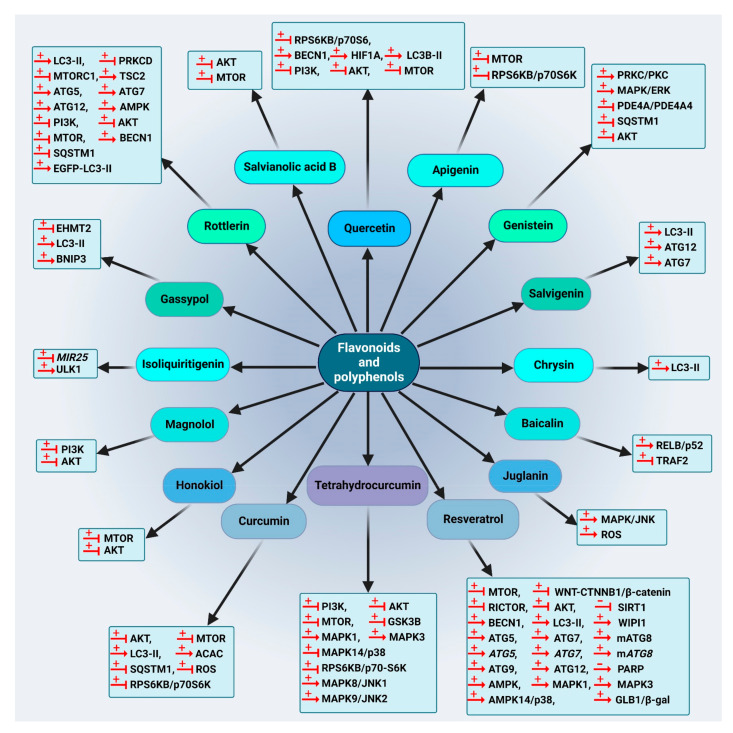
The schematic role of flavonoids and phenolic compounds on the proteins involved in the autophagy pathway. mATG8, mammalian Atg8-family proteins. →: Indicates activation or an increase in the mentioned protein expression. 
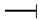
: Indicates deactivation or a decrease in the mentioned protein expression. − and +: Indicate the effect of the proteins on different pathways (− means negative effect and + means positive effect). The figure was created with BioRender.com. Licensing Right OG24OB731X, 21 November 2022.

**Figure 13 cancers-14-05839-f013:**
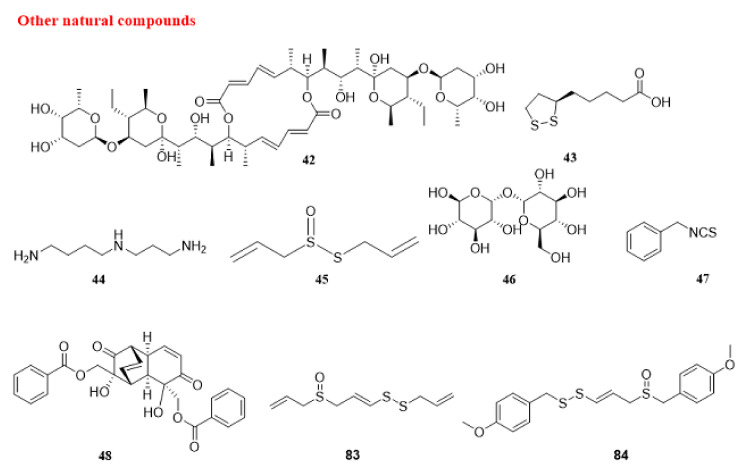
Chemical structures of unclassified natural compounds with anti-cancer activity. Elaiophylin (42), lipoic acid (43), spermidine (44), allicin (45), trehalose (46), benzyl isothiocyanate (47), (+)-grandifloracin (48), ajoene (83), bisPMB (84). The images were prepared using ChemDraw Professional 15.0.

**Figure 14 cancers-14-05839-f014:**
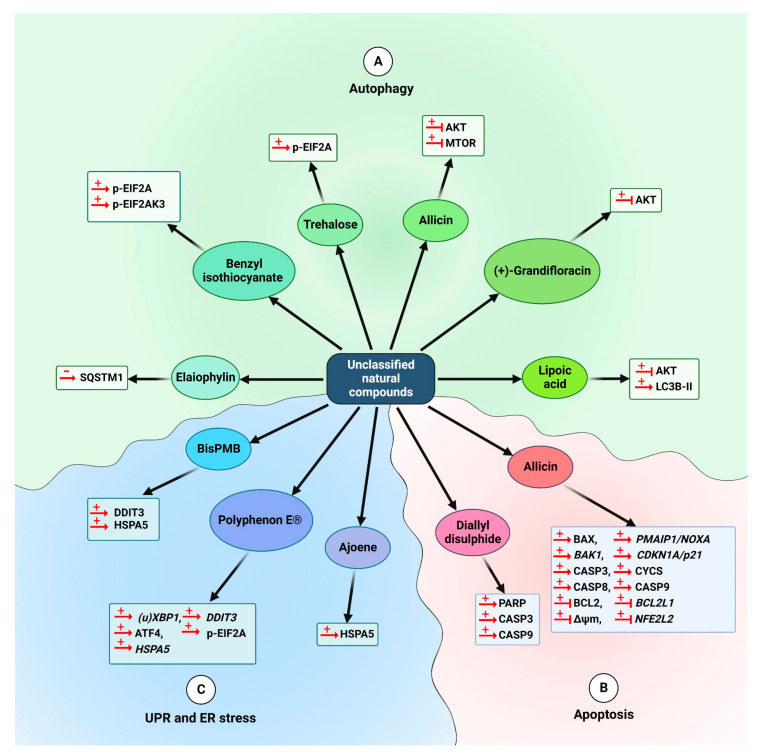
The schematic role of unclassified compounds on the proteins involved in three different pathways ((**A**): Autophagy, (**B**): Apoptosis and (**C**): The UPR and ER stress). (**A**): Six natural compounds are shown in this part of the figure as autophagy modulators. Upregulation of p-EIF2A, p-EIF2AK3 and LC3B-II and inhibition of AKT and MTOR result in autophagy in cancer cells. Conversely, the accumulation of SQSTM1 showed that elaiophylin acts as an autophagy inhibitor. (**B**): Apoptosis: Two compounds that induce apoptosis are illustrated in this part. Both compounds cause PARP, CASP3 and CASP9 activation in different cell lines. Moreover, allicin induces apoptosis via activation of CASP8 and upregulation of *CDKN1A/p21*, *PMAIP1/NOXA, BAK1,* BAX and CYCS. Moreover, downregulation of NFE2L2, BCL2 and BCL2L1 and loss of ∆Ψm are observed in allicin-treated cell lines. (**C**) UPR and ER stress: There are three compounds in this part that induce an ER accumulation of misfolded proteins and activate the UPR. Under ER stress induced by these compounds, some proteins, including p-EIF2A and ATF4, are overexpressed in different types of cancer cells. Moreover, the expression of some genes including *HSPA5*, *(u)XBP1* and *DDIT3* is increased. →: Indicates activation or an increase in the mentioned protein expression. 
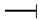
: Indicates deactivation or a decrease in the mentioned protein expression. − and +: Indicate the effect of the proteins on different pathways (− means negative effect and + means positive effect). The figure was created with BioRender.com. Licensing Right IP24OB76U3, 21 November 2022.

**Figure 15 cancers-14-05839-f015:**
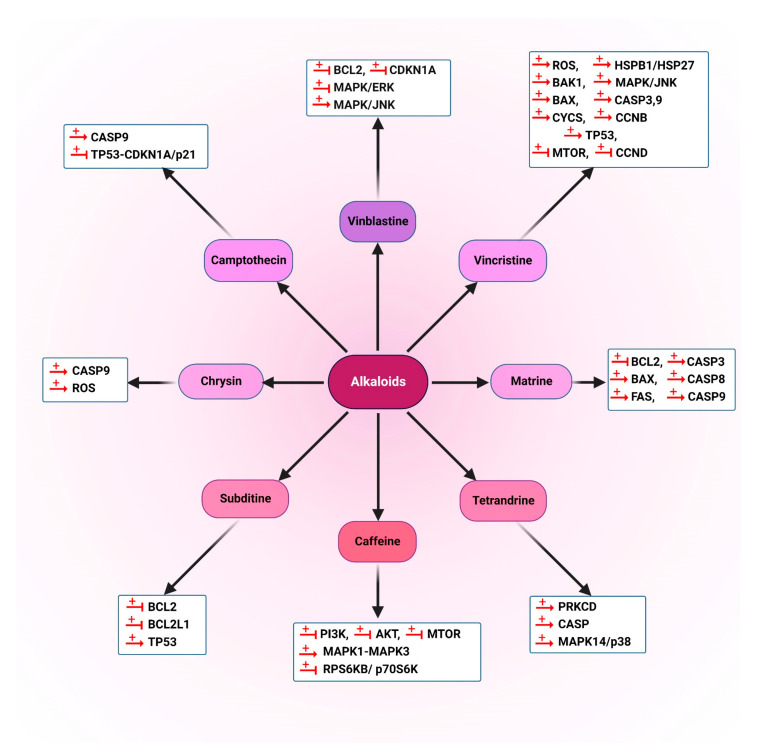
The schematic role of alkaloids on the proteins involved in the apoptosis pathway. Alkaloids induce apoptosis in distinct cells via downregulation of CDKN1A/p21/WAF1/CIP1, BCL2, BCL2L1 and CCND while promoting the expression of some other proteins like CASP3, CASP8, CASP9, CCNB, BAX, CYCS, BAK1 and FAS. Moreover, activation of MAPK/JNK, TP53, PRKCD and MAPK14/p38 signaling pathways, as well as inactivation of MAPK/ERK and RPS6KB/p70S6K, is observed in different cell lines treated with these alkaloids. →: Indicates activation or an increase in the mentioned protein expression. 
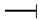
: Indicates deactivation or a decrease in the mentioned protein expression. +: means positive effect of the proteins on different pathways. The figure was created with BioRender.com. Licensing Right RD24OB7C3X, 21 November 2022.

**Figure 16 cancers-14-05839-f016:**
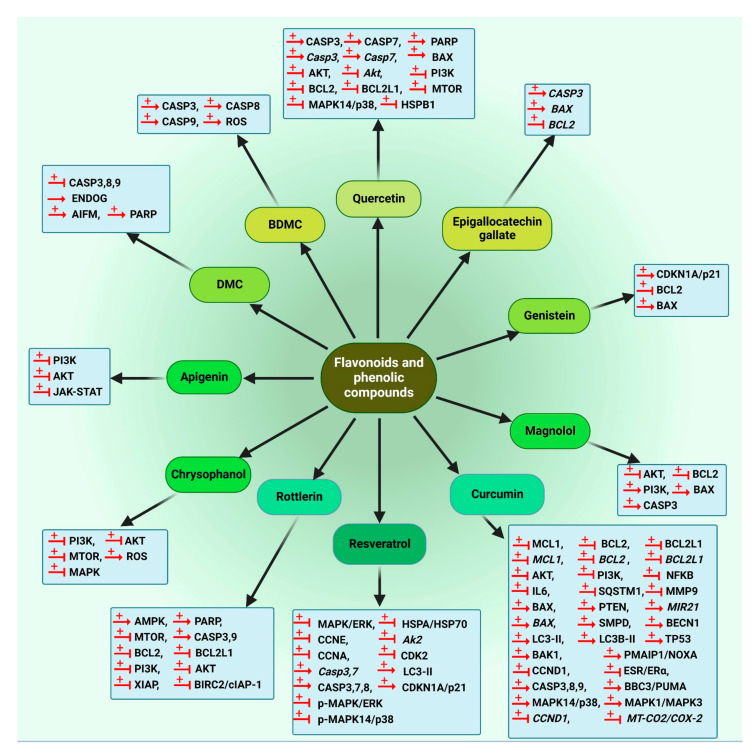
The schematic role of flavonoids and phenolic compounds on the proteins involved in the apoptosis pathway. PI3K inhibition, AKT dephosphorylation, activating caspases, dissociation of BAX from BCL2L1, increased PARP and generation of ROS are detectable changes in different cells treated with flavonoids and phenolic compounds. Additionally, these compounds regulate apoptosis-related proteins and genes such as MCL1, MAPK/ERK and AKT and genes such as MCL1 and ENDOG expression, while down-regulating CCNE, CCNA, BIRC2/cIAP-1 and CDK2. Moreover, activation of TP53, MAPK14/p38-HSPB1 and SMPD is observed in these cell lines. Moreover, NFKB and *MIR21* are some other important targets. Moreover, these compounds suppress cell survival and cell proliferative genes, including BCL2, CCND1, *MT-CO2/COX-2*, IL6 and MMP9. →: Indicates activation or an increase in the mentioned protein expression. 
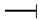
: Indicates deactivation or a decrease in the mentioned protein expression. +: means positive effect of the proteins on different pathways. The figure was created with BioRender.com. Licensing Right PC24OB7FPB, 21 November 2022.

**Figure 17 cancers-14-05839-f017:**
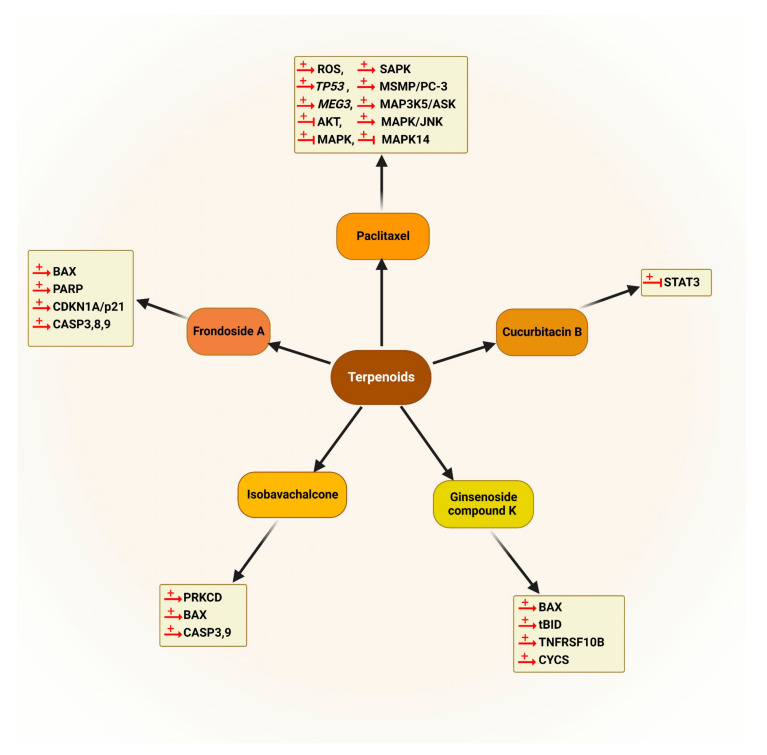
The schematic role of terpenoids on the proteins involved in the apoptosis pathway. These terpenoids increase the level of cleaved CASP3, CASP8 and CASP9 and induce the expression of pro-apoptotic proteins (BAX, truncated BID (tBID) and CYCS) and CDKN1A/p21. Moreover, inhibition of STAT3 phosphorylation, regulation of ROS and AKT-MAPK signaling and MAPK1/ERK2-MAPK3/ERK1, as well as activation of PRKCD is observed in cells treated with these natural compounds. Moreover, the compounds activate MAP3K5/Ask1-MAPK14/p38 and MAPK/JNK-MAPK/SAPK pathways and increase the expression of *MEG3* and *TP53*. →: Indicates activation or an increase in the mentioned protein expression. 
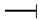
: Indicates deactivation or a decrease in the mentioned protein expression. +: means positive effect of the proteins on different pathways. The figure was created with BioRender.com. Licensing Right MV24OB7IC0, 21 November 2022.

**Figure 18 cancers-14-05839-f018:**
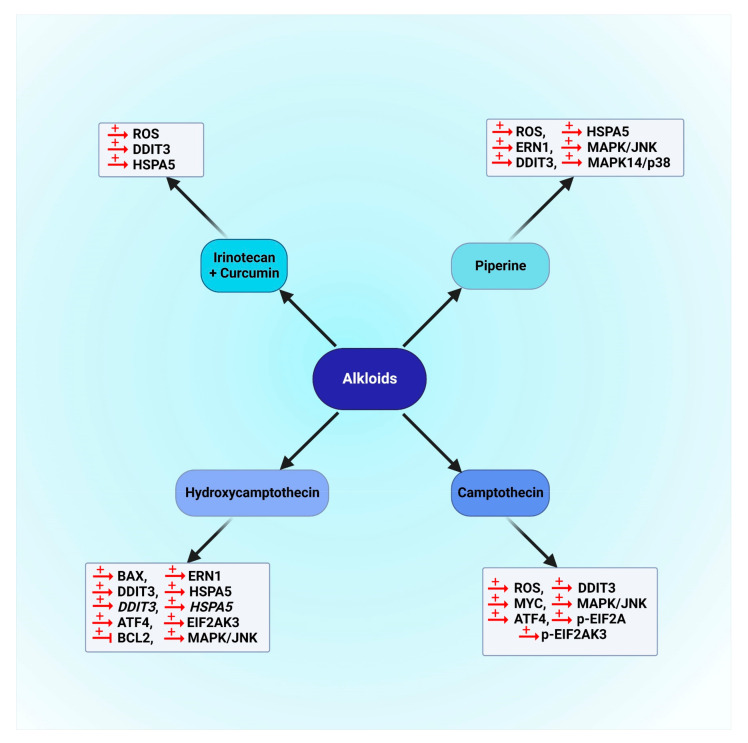
The schematic role of alkaloids on the proteins involved in the UPR and ER stress pathways. Alkaloids result in the UPR and ER stress in different cell lines through effects on a variety of proteins. These compounds increase the expression of some proteins associated with ER stress, including ERN1, DDIT3, HSPA5, EIF2AK3, MYC, EIF2A, ATF6 and ATF4. Activation of MAPK/JNK and MAPK14/p38 and generation of ROS are other considerable changes in the treated cell. Activation of MAPK/JNK and MAPK14/p38 and generation of ROS are other considerable changes in the treated cell, including changes to the mRNAs level of *DDIT3* and *HSPA5* proteins. →: Indicates activation or an increase in the mentioned protein expression. 
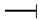
: Indicates deactivation or a decrease in the mentioned protein expression. +: means positive effect of the proteins on different pathways. The figure was created with BioRender.com. Licensing Right VE24OB7LHH, 21 November 2022.

**Figure 19 cancers-14-05839-f019:**
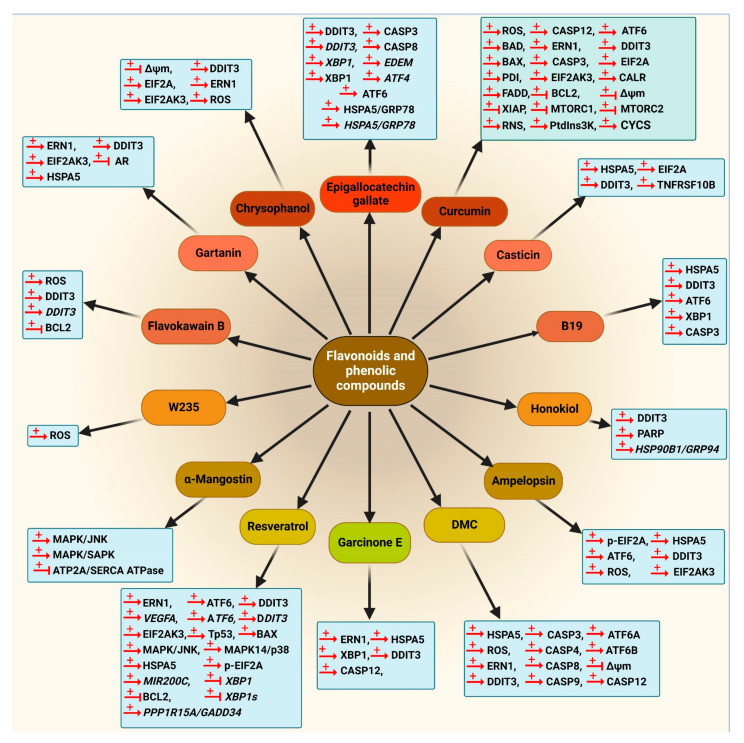
The schematic role of flavonoids and phenolic compounds on the proteins involved in the UPR and ER stress pathways. Most of the compounds mentioned in this part result in an increase in ER stress-related proteins, including HSPA5, EIF2AK3 and DDIT3. Moreover, they activate ATF6, PPP1R15A/GADD34, XBP1 splicing, DDIT3 and EDEM genes. Moreover, these compounds increase phosphorylation of MAPK/JNK, expression of CYCS, BAX and FADD, as well as activation of caspases and upregulation of TNFRSF10B, triggering ROS generation. Suppression of the transcriptional activity of XBP1 via SIRT1 was reported in some cell lines treated with these compounds. In one case, the inhibitory activity of the compound against AR was observed. Moreover, contrary to other compounds, resveratrol promotes cell death by inhibiting the transcriptional activity of XBP1s, which leads to impairment of the cell survival functions. →: Indicates activation or an increase in the mentioned protein expression. 
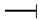
: Indicates deactivation or a decrease in the mentioned protein expression. +: means positive effect of the proteins on different pathways. The figure was created with BioRender.com. Licensing Right PA24OB7OP8, 21 November 2022.

**Figure 20 cancers-14-05839-f020:**
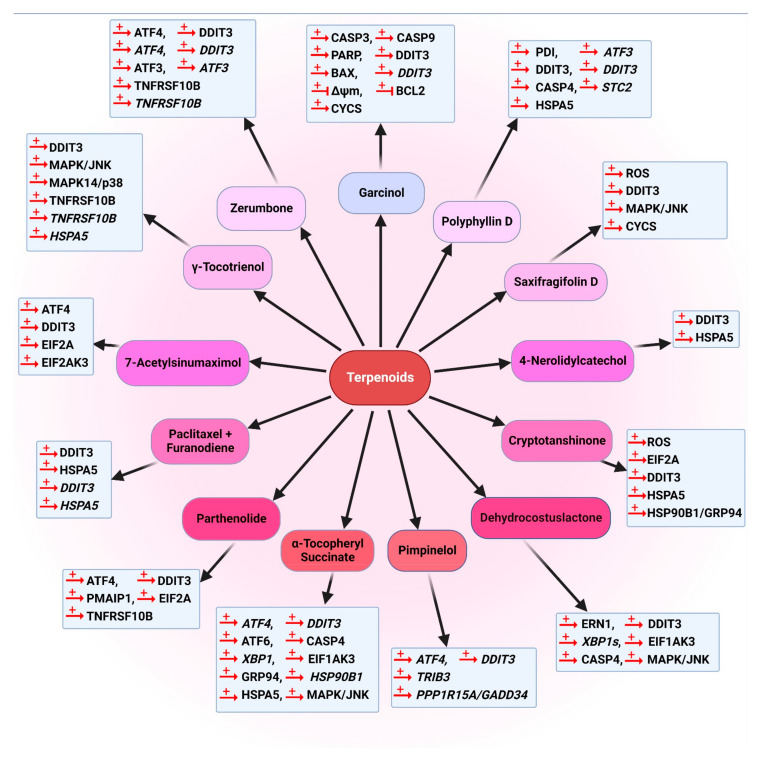
The schematic role of terpenoids on the proteins involved in the UPR and ER stress pathways. Terpenoid treatment results in upregulation of the mRNA levels of HMOX1, ATF3, DDIT3, HSPA5, HSP90B1/GRP94 and STC2. Moreover, typical ER stress-related proteins such as HSPA5, PDI, DDIT, EIF2AK3, ERN1, ATF6, ATF4 and HSP90B1/GRP94 increased in the cells treated with these compounds. Moreover, these compounds increase protein aggregation and mRNA expression of ATF4, PPP1R15A/GADD34 and TRIB3. In some cases, the expression of TNFRSF10B and increasing protein aggregation of PPP1R15A/GADD34 and TRIB3 are observed. →: Indicates activation or an increase in the mentioned protein expression. 
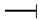
: Indicates deactivation or a decrease in the mentioned protein expression. +: means positive effect of the proteins on different pathways. The figure was created with BioRender.com. Licensing Right IB24OB7ST2, 21 November 2022.

**Figure 21 cancers-14-05839-f021:**
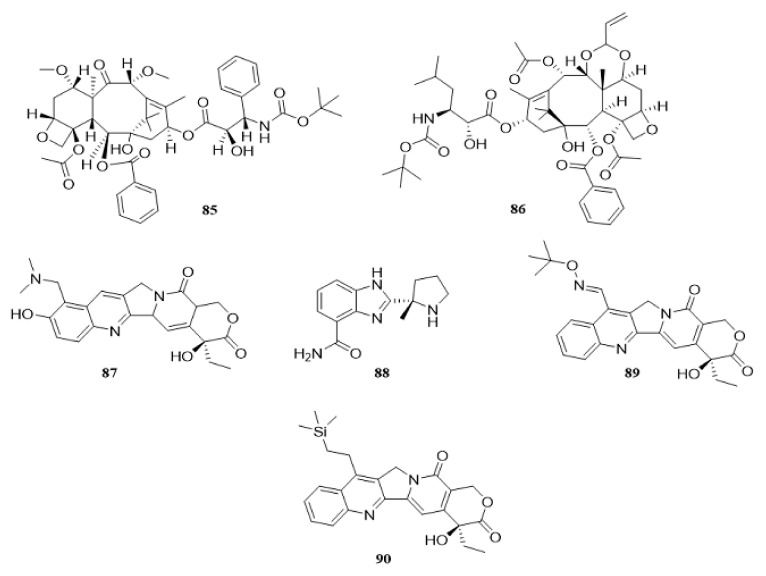
Chemical structures of natural compounds that have been used in clinical trials. Cabazitaxel (85), TPI 287 (86), topotecan (87), veliparib (88), gimatecan (89), karenitecin (90). The images were prepared using ChemDraw Professional 15.0.

**Table 1 cancers-14-05839-t001:** Natural autophagy modulators and their effect on signal transduction pathways.

Cancer Type	Compound	Cell Line	Target	Reference
Human colon cancer	Synthesized derivatives of berberine (7)	HCT116, SW613-B3	↑ LC3-II	[149]
	Oridonin (16)	SW480	↑ p-AMPK, ↓ SLC2A1/GLUT1	[161]
	Vinblastine (1)	LS174T	↑ LC3-II, ↑ autophagic vacuole,	[134]
	Camptothecin (5)	HCT116, RKO	AMPK-TSC2-MTOR	[144]
	B-group soyasaponins	HCT-15	↑LC3-II, ↑ autophagic vacuoles	[157]
	Helenalin (20)	RKO	↓ RELA, ↑ *LC3B, ATG12* and caspases	[170]
	Ursolic acid (21)	HCT15	MAPK/JNK, ↑ LC3, SQSTM1	[172]
	Ginsenoside compound K (22)	HCT15	ROS-MAPK/JNK, ↑ TNFRSF10A	[173]
	Quercetin (25)	DLD-1, HT-29	↑ EGFP-LC3	[180]
	Curcumin (33)	HCT116	↑ ROS-MAPK1-MAPK3-MAPK14/p38	[220]
			↓ AKT-MTOR, ↑TFEB	[226]
	Bis-dehydroxycurcumin (36)	HCT116	ER stress	[231]
	Salvianolic acid B (40)	HCT116, HT-29	↓ AKT-MTOR	[245]
	Lipoic acid (43)	HCT116	↓ MGMT, AKT ↑ LC3-II	[248]
	Resveratrol (32) and spermidine (44)	HCT116	AMPK-MTOR independent, Change phosphoproteome and acetylproteome	[250]
Neuroblastoma	Salvigenin (28)	SH-SY5Y	↑ LC3-II, ATG7, ATG12	[191]
Hepatocellular carcinoma	Tetrandrine (9)	Huh7, HCCLM9, Hep3B	WNT-CTNNB1 pathway and ↓ MTA1	[152]
	Resveratrol (32)	Huh-7	↑ ATG5, ATG7, ATG9, ATG12	[209]
	Baicalin (30)	MHCC97L	↑ RELB/p52, ↓ TRAF2	[194]
Thyroid cancer	Allicin (45)	SW1736, HTh-7	↓ AKT-MTOR	[253]
Oral squamous cell carcinoma	Carbazole alkaloids	CLS-354	↑ SQSTM1, LC3B-II, cleaved CASP3	[155]
Breast cancer	Resveratrol (32)	MCF-7, SUM159	↓ WNT-CTNNB1 ↑ LC3-II, BECN1, ATG7	[198]
		MCF-7	↓ AKT-MTOR	[205]
		MCF-7	WIPI1, ↑ LC3-II, ATG7, ATG5	[208]
	Bisleuconothine A (11)	MCF-7	AMPK-MTOR, ↑ LC3-II,	[154]
	Helenalin (20)	MCF-7	↓ RELA, ↑ *LC3B, ATG12* and caspases	[170]
	Qauercetin (25)	MCF-7	Triggering macroautophagy, ↓ MTOR	[182]
	Rottlerin (38)	MCF-7	↑ TSC2,MTORC1, ↑ EGFP-LC3-II, EGFP	[238]
			↑ LC3-II, ↓ SQSTM1	[239]
			MTORC1 inhibition	[241]
			↑ LC3, BECN1, ATG12, AMPK	[240]
	Isoliquiritigenin (41)	MCF-7	↓ *MIR25*, ↑ ULK1	[246]
	Juglanin (31)	MCF-7, SKBR3,	↑ MAPK/JNK, ROS	[195]
	chrysin (29)	MCF-7	↑ LC3-II	[192]
Lung cancer	Hernandezine (10)	A549, H1299	↑ AMPK, ↑ ATG7	[153]
	Curcumin (33)	A549	↑ LC3-II, p-AMPK, p-ACAC ↓ SQSTM1, AMPK	[222]
	Resveratrol (32)	HFL1	SIRT1, PARP	[213]
	Bisleuconothine A (11)	A549	AKT-MTOR, ↑LC3-II	[154]
	Camptothecin (5)	H1299	↑ Autophagosome formation	[146]
	Paclitaxel (17)	A549, Calu-3	↑ BECN1, ↓ *MIR216B*	[166]
	Capsaicin (13)	H1299	↑ WT TP53	[156]
	Benzyl isothiocyanate (47)	A549	↑ EIF2AK3, EIF2A, LC3-II-ATG5, AVOs	[255].
Osteosarcoma	Curcumin (33)	MG63	↑ MAPK/JNK	[171]
Human fibrosarcoma	Rottlerin (38)	HT1080	↑ LC3-II	[236]
Gastric cancer (GC)	Matrine (6)	SGC-7901	blocking autophagic degradation	[147]
	Quercetin (25)	AGS, BGC-823, SGC-7901, MKN-28	PI3K-AKT-MTOR, HIF1A	[181]
	Curcumin (33)	SGC-7901, BGC-823	↓ PI3K-AKT-MTOR ↑ TP53, CDKN1A/p21	[230]
		MKN-28	PI3K-AKT-MTOR	[255]
	Magnolol (35)	SGC-7901	↓ PI3K-AKT	[234]
Uterine leiomyosarcoma	Curcumin (33)	SKN, SK-UT-1	↑ MAPK1, MAPK3	[223]
Pancreatic cancer	Curcumin (33)	PANC-1, BxPC3	↑ LC3-II, ↓ MTOR	[229]
	(+)-Grandifloracin (48)	PANC-1	↓ AKT	[256]
	Rottlerin (38)	MDA-Panc28	↓ PRKCD, TGM2	[237]
	Gossypol (39)	PANC-1	↑ LC3-II, BNIP3 ↓ EHMT2	[244]
Nasopharyngeal carcinoma	Neoalbaconol (14)	C666, HK1, CNE1	PI3K-AKT	[158]
prostate cancer	Resveratrol (32)	PC-3, DU145	↑ AMPK ↓ STIM1, AKT, MTOR	[201]
	Camptothecin (5)	LNCaP	↑ EIF2AK3, EIF2A, ATF4, DDIT3, MYC, ROS, MAPK/JNK	[145]
	Rottlerin (38)	Cancer stem cells	↓ PI3K-AKT-MTOR ↑ LC3-II, ATG5, ATG7, ATG12, BECN1	[242]
leukemia	Tanshinone IIA (19)	KBM-5	↑ AMPK, LC3-II, MAPK/ERK, RAF, RPS6KA1/p90RSK ↓ MTOR, RPS6KB/p70S6K	[169]
	Resveratrol (32)	K562	MAPK/JNK ↑ SQSTM1, AMPK	[204]
	Tetrahydrocurcumin (37)	HL-60	↓ PI3K-AKT-MTOR-RPS6KB/p70S6K, GSK3B-MAPK14/p38, ↑ MAPK1-MAPK3,MAPK8/JNK1-MAPK9/JNK2	[232]
	Apigenin (26)	TF1	↓ MTOR, RPS6KB/p70S6K, JAK-STAT	[187]
Cervical cancer	Guttiferone K (15)	Hela	↑ MAPK/JNK, LC3-II ↓ AKT-MTOR, SQSTM1	[159]
	Paclitaxel (17)	SiHa	↑ LC3-II, ATG7	[167]
		HeLa-R	↑ *ATG* genes *(ATG7, ATG12–ATG5), BECN1, MAP1LC3B, HIF1A*	[165]
	Resveratrol (32)	SiHa, Hela	↑ LC3-II, autophagosomes	[214]
		Hela	↑ Lysosomal permeability	[199]
Oral squamous cell carcinoma	Curcumin (33)	YD10B	↑ LC3-II, AVOs	[221]
Glioma	Resveratrol (32)	U373	↑ MAPK14/p38-MAPK1-MAPK3	[203]
		U-87, U-251, U-138	↑ LC3-II, ATG5, BECN1	[211]
	Curcumin (33)	U87-MG, U373-MG	↓ AKT-MTOR-RPS6KB/p70S6K, ↑ MAPK1-MAPK3	[219]
Ovarian	Helenalin (20)	A2780	↓ RELA, ↑ *LC3B, ATG12* and caspases	[170]
	Quercetin (25)	OVCAR3	↑ BECN1, LC3B-II	[183]
	Resveratrol (32)	A2780, CaOV3	↑ BCL2L1, BCL2	[197]
		SKOV3, CaOV3	↓ AKT-MTOR	[202]
	Genistein (27)	A2780, CaOV3, ES2	↓ p-AKT	[189]
		CHO	↓ PDE4A4-SQSTM1	[190]
	Elaiophylin (42)	SKOV3, A2780	↑ SQSTM1	[247]
Epidermoid carcinoma	Resveratrol (32)	A431	↓ RICTOR, RHOA GTP ↑ GLB1/β-gal	[206]
Urothelial carcinoma	Frondoside A (23)	RT112	↑ LC3-I/II, SQSTM1	[175]
Skin cancer (melanoma)	Resveratrol (32)	B16	↑ Ceramide, ↓ AKT-MTOR pathway	[200]
	Honokiol (34)	B16-F10	↓ NOTCH1, AKT-MTOR	[233]
Human melanoma	Curcumin (33)	A375,C8161	↓ AKT-MTOR	[228]

Up and down arrows indicate increase or decrease, respectively.

**Table 2 cancers-14-05839-t002:** Natural apoptosis modulators and their effect on signal transduction pathways.

Cancer Type	Compound	Cell Line	Target	Reference
Pancreatic cancer	Matrine (6)	BxPC-3, PANC-1	↑ BCL2-BAX, FAS, CASP3, CASP8, CASP9	[266]
	Rottrelin (38)	stem cells (CD44^+^ CD24^+^ ESA^+^)	PI3K-AKT-MTOR inhibition	[317]
Cervical cancer	Allicin (45)	SiHa	↑ CASP3, CASP8, CASP9	[334]
Leukemia	Tetrandrine (9)	U937	↑ Caspases, PRKCD	[151]
	Diallyl disulfide	HL-60	↑ CASP3, ↓ PARP	[340]
Murine colon cancer	Tetrandrine (9)	CT-26	↑ MAPK14/p38	[267]
Human prostate cancer	Subditine (49)	LNCaP, PC-3	↓ BCL2-BCL2L1	[268]
	Curcumin (33)	LNCaP	NFKB, NFKBIA/IκBα	[302]
		PC-3	↓ p-AKT and NFKB	[303]
		PC-3	↓ BCL2, BCL2L1 ↑ BAX, BAK1, BBC3/PUMA, PMAIP1/NOXA, CASP3, CASP9	[291]
	Quercetin (25)	DU-145, LNCaP	↓ AKT, ↑ Activation	[275]
		LNCaP	↓ BCL2L1 ↑ BAX, caspases	[277]
	Resveratrol (32)	DU-145	↑ CASP3 activity by HSPA8/HSP70 involvement	[316]
	Rottlerin (38)	Cancer stem cells	↓ PI3K-AKT-MTOR ↑ AMPK	[242]
	Caffeine (50)	PC12D, Hela, SH-SY5Y	↓ PI3K-AKT-MTOR-RPS6KB/p70S6K ↑ MAPK1-MAPK3	[270]
Human lung cancer	Camptothecin (5)	H1299, H460	↑ CASP9	[146]
	Bisdemethoxycurcumin (BDMC) (56)	NCI-H460	↑ CASP3, CASP8 and CASP9	[310]
	Demethoxycurcumin (DMC) (55)	NCI-H460	↑ CASP3, CASP8 and CASP9, AIFM, ENDOG, PARP	[311]
	Curcumin analog (WZ35) (57)	H1975	ROS, ER stress, mitochondrial dysfunction	[312]
	Cucurbitacins (A, B, I and Q)	A549	Inhibition of STAT3	[321]
Human laryngeal cancer	Cucurbitacin B (60)	Hep-2	↓ p-STAT3	[325]
Human renal cancer	Curcumin (33)	Caki	↑ ROS, TNFRSF10B	[304]
Human tenon’s capsule fibroblasts	Hydroxycamptothecin(51)	HTCFs	↑ EIF2AK3/PERK	[271]
Human colon cancer	Camptothecin (5)	HCT116, RKO	↑ BECN1, TP53	[144]
	Curcumin (33)	HCT116	↓ NFKB	[283]
	Resveratrol (32)	HT-29, COLO 201	ROS, CASP3 and CASP8 activation	[314]
	Allicin (45)	HCT116	↑ CYCS, BAX ↓ BCL2	[335]
	Genistein (27)	HT-29	↑ CDKN1A, BAX-BCL2	[281]
	Ginsenoside compound K (22)	HT-29, HCT 116	↑ BAX, tBID, CYCS	[173]
	Diallyl disulfide	HT-29	↑ CASP9, CASP3	[342]
Urothelial carcinoma	Frondoside A (23)	RT112 RT4,HT-1197, TCC-SUP, T-24	↑ CASP3, CASP8 and CASP9, PARP, BAX, CDKN1A, DNA fragmentation	[175]
Glioma	Resveratrol (32)	U251	Inhibiting autophagy	[315]
Ovarian cancer	Curcumin (33)	SKOV3 ES-2	Activation of intrinsic and extrinsic pathways	[305]
	Curcumin (33)	HEY OVCA429OCC1, SKOV3	AKT signaling. ↑ CASP3, CASP8 and CASP9, CYCS, BID, ↓ BCL2	[307]
Human pancreatic cancer	Curcumin (33)	PANC1,BxPC3	↑ *BAX*, ↓ *BCL2*	[229]
Human hepatocellular carcinoma	Resveratrol (32)	Huh-7	↑ CDKN1A, ↓ CCNE, CCNA, CDK2, p-MAPK/ERK, p-MAPK14/p38	[209]
Brain cancer	Triazolyl curcumins (53, 54)	CRT-MG	Increased cytotoxicity and apoptosis	[306]
Gastric cancer	Curcumin (33)	MKN-28	↓ BCL2, ↑ BAX, CASP3 and CASP9 activation	[255]
	Curcumin (33) + quercetin (25)	MGC-803	↑ CYCS, ↓ phosphorylation of AKT and MAPK/ERK	[309]
	Quercetin (25)	BGC-82	↓ BCL2:BAX ratio, ↑ CASP3	[278]
	Magnolol (35)	SGC-7901	↓ PI3K-AKT, ↑ BAX- BCL2 and CASP3	[234]
	Allicin (45)	SGC-7901	↑ FAS, BAX, CYTC, CASP3, CASP8, CASP9	[338]
	Curcumin (33)	SGC-7901	↓ NFKB, *BCL2*, *BCL2L1*	[284]
Osteosarcoma	Curcumin (33)	MG63	MAPK/JNK pathway, inhibition of autophagy	[171]
Human breast cancer	Allicin (45)	MCF-7, HCC-70	↑ CASP3, CASP8 and CASP9, *CDKN1A*/p21 *PMAIP1/NOXA, BAK1* ↓ ∆Ψm, *BCL2L1*	[339]
	Chrysophanol (58)	MCF-7, BT-474	PI3K-AKT and MAPK	[318]
	Curcumin (33)	MCF-7	↑ TP53, BAX	[288]
		MCF-7	↓ *MIR21*, AKT ↑ PTEN	[286]
		T-47D	↓ TP53, ESR1/ERα	[290]
		MCF-10F	↑ ROS	[308]
	Quercetin (25)	Cancer stem cells (CSCs)	↑ BAX, ↓ BCL2, PI3K-AKT-MTOR,	[279]
	Quercetin (25)	BT-20, MCF-7	↑ PARP cleavage, CASP3 and CASP7	[276]
	Chrysin (29)	MCF-7	↑ ROS, CASP3	[192]
	Rottlerin (38)	MCF-7	↑ CASP9, CASP3, PARP cleavage	[239]
		cancer stem cells (CSCs)	AKT-MTOR, AMPK, ↓ BCL2, BCL2L1, XIAP and BIRC2/cIAP-1	[240]
Human fibrosarcoma	Rottlerin (38)	HT1080	PRKCD-independent pathway	[236]
Triple-negative breast cancer	Curcumin (33)	MDA-MB-231	↓ p-MAPK1-MAPK3, EGFR inhibition	[297]
	Epigallocatechin gallate (52)	Hs578T	*↓ BCL2* *↑ CASP3, BAX*	[280]
Human multiple myeloma	Curcumin (33)	MM RPMI 8226	↑ TP53, BAX	[289]
Head and neck squamous cell cancer	Curcumin (33)	MDA 686LN	↓ NFKB, *BCL2, CCND1, IL6, MT-CO2, MMP9*	[292]
Oral squamous cancer	Quercetin (25)	Cancer stem cell (CSCs)	↓ MAPK14/p38-HSPB1	[278]
Human leukemia	Curcumin (33)	HL60	↑ SMPD, ceramide generation	[293]
			Ceramide generation, CASP8 and inhibition of BCL2L1	[294]
		K562	↑ BECN1, LC3-II, promoting apoptosis by autophagy	[295]
	Apigenin (26)	HL60	↓ PI3K-AKT, JAK-STAT	[187]
Multiple myeloma	Isobavachalcone (59)	H929	Activation of PRKCD	[320]
Leiomyosarcoma	Curcumin (33)	SKN, SK-UT-1	↑ MAPK1-MAPK3, LC3B-II, ↓ SQSTM1	[223]

EPCAM, epithelial surface antigen. Up and down arrows indicate increase or decrease, respectively.

**Table 3 cancers-14-05839-t003:** Natural compounds with ER stress-mediated anti-cancer activity.

Cancer Type	Compound	Cell Line	Target	Reference
Lung cancer	Polyphyllin D (71)	NCI-H460	↑ DDIT3, HSPA5, PDI, CASP4, *ATF3*, *DDIT3*, *STC2*	[381]
	Dehydrocostuslactone (73)	NCI-H460 A549	↑ EIF2AK3, DDIT3, ERN1, MAPK/JNK, ROS, *XBP1s*	[383]
	Resveratrol (32)	NCI-H460	↑ DDIT3, HSPA5, RECK, *MIR200C*	[370]
	Bisdemethoxycurcumin (BDMC) (56)	NCI-H460	↑ HSPA5; ERN1-ERN2; DDIT3; ATF6; ATF6B; CASP4	[310]
	Demethoxycurcumin (DMC) (55)	NCI-H460	↑ HSPA5, ERN2, DDIT3, ATF6, ATF6B, CASP4, ROS ↓ ΔΨm	[311]
	WZ35 (57)	HI975	↑ p-EIF2A; ATF4; DDIT3	[312]
	Resveratrol (32) + arsenic trioxide	A549	↑ HSPA5, DDIT3, CASP12	[372]
	TMS Resveratrol (32) analog	A579; H1975	↑ EIF2A, EIF2AK3, DDIT3, Ca^2+^, AMPK	[373]
	Furanodiene (76)	A549, 95-D	↑ DDIT3, HSPA5, *DDIT3*	[386]
	Parthenolide (77)	A549, Calu-1, H1299, H1792	↑ ATF4, DDIT3, EIF2A,	[387]
Human multiple myeloma	Resveratrol (32)	ANBL-6	↑ ERN1, EIF2AK3, *ATF6*, *DDIT3*, MAPK/JNK, *PPP1R15A/GADD34* ↓ *XBP1s, VEGFA*	[367]
Breast cancer	Cryptotanshinone (74)	MCF-7	↑ EIF2A, HSP90B1/GRP94, HSPA5, DDIT3, ROS	[384]
	Pimpinelol (75)	MCF-7	↑ *ATF4*, DDIT3, PPP1R15A, *TRIB3*	[385]
	Ajoene (83)	MDA-MB-231	↑ HSPA5	[396]
	Chrysophanol (58)	MCF-7; BT-474	↑ ROS, EIF2AK3, EIF2A, DDIT3, ERN1	[318]
	Saxifragifolin D (72)	MDA-MB-231, MCF-7	↑ MAPK/JNK, Ca^2+^, ROS, DDIT3	[382]
	Ampelopsin (64)	MDA-MB-231, MCF-7	↑ HSPA5, p-EIF2AK3, p-EIF2A, cleaved ATF6, DDIT3, ROS	[360]
	γ-tocotrienol (82)	MDA-MB-231, MCF-7	↑ DDIT3, HSPA5, XBP1, TNFRSF10B, *TNFRSF10B*, MAPK/JNK¸CASP3, CASP8	[394]
Colon cancer	Curcumin (33)	AGS, HT-29	↑ DDIT3, MAPK/JNK-FADD	[359]
	Curcumin (33)+ sildenafil	HCT116, HT-29	↑ p-EIF2A; DDIT3	[363]
	Curcumin (33)+ irinotecan (62)	LoVo, HT-29	↑ DDIT3, PDI, HSPA5	[357]
	Piperine (61)	HT-29	↑ ERN1, DDIT3, HSPA5, ROS, MAPK/JNK	[355]
	Zerumbone (78)	HCT116-p53null, SW480	↑ ATF4, DDIT3, ATF3, *ATF4, DDIT3, ATF3*, EIF2AK3, EIF2A, *TNFRSF10B*, TNFRSF10B	[389]
	Flavokawain B (66)	HCT116	↑ DDIT3, *DDIT3*, ROS, BCL2	[365]
Gastric cancer (GC)	α-Tocopheryl succinate (79)	SGC-7901	↑ *DDIT3*, MAPK/JNK, EIF2AK3, ATF6, *XBP1*, *ATF4* HSP90B1/GRP94, HSPA5, CASP4	[390]
	7-Acetylsinumaximol B (81)	NCI-N87	↑ EIF2AK3, EIF2A, ATF4, DDIT3	[393]
	α-Tocopheryl succinate (79)	SGC-7901	↑ *DDIT3*, MAPK/JNK, HSPA5, ROS, CASP4	[391]
	Honokiol (34)	MKN-45 SCM-1, AGS, N87	↑ DDIT3, CAPN2, PARP and *HSP90B1/GRP94*, cleavage	[379]
	Casticin (63)	BGC-823	↑ DDIT3, p-EIF2A, EIF2A, HSPA5	[359]
	Casticin (63)	SGC-7901 MGC-803	↑ DDIT3, TNFRSF10B, ROS	[359]
Malignant mesothelioma	Epigallocatechin gallate (52)	MM98	↑ HSPA5; DDIT3; ATF4; EDEM; XBP1	[358]
Liver cancer	Curcumin (33)+ sildenafil	HEPG2	↑ EIF2A, BECN1; ↓ MTORC1 and MTORC2 activity	[363]
Murine Myeloma	Curcumin (33)	WEHI-3	↑ ATF6; DDIT3; ERN1; CASP12; BCL2, ROS, Ca^2+^, ↓ ΔΨm	[361]
Human melanoma	4-nerolidylcatechol (80)	SK-MEL-28; BRAi/MEKi	↑ HSPA5, DDIT3	[392].
	Vincristine (3)	B16	↑ AMPK, TP53 ↓MTORC1	[264]
	Resveratrol (32)	A375SM	↑ p-EIF2A, DDIT3, ROS, MAPK14/p38, TP53, BAX, ↓ BCL2	[368]
Pheochromocytoma	α-Mangostin (67)	PC12	↓ ATP2A/Ca^2+^ ATPase activity ↑ MAPK/JNK, MAPK/SAPK	[376]
Human ovarian cancer	Garcinone E (68)	HEY; A2780, A2780/Taxol cells	↑ HSPA5, ERN1, XBP1, DDIT3, CASP12	[377]
	B19 (65)	HO8910	↑ PDI, HSPA5, DDIT3, ATF6, XBP1, CASP3	[359]
Human hepatoblastoma	Resveratrol (32) + palmitate	HepG2	↑ *XBP1s*, *DDIT3*, ↓ ROS	[369]
	RES006 Resveratrol (32) analog	HepG2	↑ ROS	[374]
Human nasopharyngeal cancer	Resveratrol (32)	NPC-TW076; NPC-TW039	↑ ERN1, DDIT3, ATF6, p-EIF2AK3	[371]
Human prostate cancer	Gartanin (69)	LNCaP; 22RV1	↑ DDIT3, HSPA5, ERN1, EIF2AK3	[378].
	Polyphenon E	PNT1a	↑ *DDIT3*, *HSPA5* p-EIF2A and ATF4	[398]
		PC3	↑ *DDIT3*, *(u)XBP1* p-EIF2A and ATF4	[398]
	Garlic extract	DU-145 DU145, U2OS,67NR	↑ HSPA5	[395]
	Curcumin (33)	PC3	↑ ERN1; ROS; PDI; CALR	[362]
Human hepatocellular carcinoma	Garcinol (70)	Hep3B	↑ DDIT3, *DDIT3*, ROS, BAX, CASP3, CASP9, PARP; ↓ ΔΨm, BCL2	[380]
Human esophageal cancer	Ajoene (83)	WHCO1	↑ HSPA5	[396]
	BisPMB (84)	WHCO1	↑ HSPA5, DDIT3	[397]

Up and down arrows indicate increase or decrease, respectively.

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
