# Peer review of "New Visions on Natural Products and Cancer Therapy: Autophagy and Related Regulatory Pathways"

_cancers, 2022, doi:10.3390/cancers14235839_

Round 1
Reviewer 1 Report
Running title: Natural Products and Cancer Therapy: How Autophagy Regulates Response of Cancer Cells to Natural Products
This is an interesting article of vital relevance to the role of autophagy in cancer and how modulation of autophagy could improve cancer therapy. The manuscript is well written, orderly and with focused results that are important for the field of cancer research. I suggest that the authors correct the following minor issues.
• Page 6, line 265: define BCL2
• In the chapter “Autophagy in advanced tumors” and figure 2, I consider that a very important role of autophagy with respect to therapy response should be addressed. Many previous papers have described that after treatment, autophagy induction is responsible for worst prognosis of cancer patients, and therapy resistance (eg: 10.1016/j.semcdb.2019.05.029)
• Page 8, line 329: define ACD
• Page 9, line 360: Eif2ak3 in capital letter? It is a human gen.
• The figure captions of the following figures should be better explained (Figure 1, and figure 6, figure 8, etc.). Figures 6 and 8 should include the name of the compounds more than the number of them.
• Figure 7, 9, etc: clearly explain the meaning of the arrows and signs (-) and (+), it can lead to confusion
• Page 17, line 557 “or glycolysis” instead of “or of glycolysis”
• Figure 9: Lc3-II and BECN1 are repeated in the paclitaxel box. In the Oblongifolin C box should appear “-“ autophagy inhibition?
Author Response
- Page 6, line 265: define BCL2
The definition “B-cell lymphoma 2” was added to the related part (Page 6, line 267) and the abbreviation list.
- In the chapter “Autophagy in advanced tumors” and figure 2, I consider that a very important role of autophagy with respect to therapy response should be addressed. Many previous papers have described that after treatment, autophagy induction is responsible for worst prognosis of cancer patients, and therapy resistance (eg: 10.1016/j.semcdb.2019.05.029)
Thanks for the important note. Some complementary sentences were added to the text using the mentioned paper and some more references: (Page 7, line 312-320)
“The plasticity of cancer cell metabolism makes limitations for anticancer treatments that could lead to therapy resistance. Evidences suggest an important role of mitophagy in tumor growth, metastasis and therapy resistance depending on tumor type, stage, or metabolic activity. Therefore, pharmacological modulation of mitophagy in tumors could be a promising anticancer strategy [98]. Researchers demonstrated that autophagy plays dual roles in drug resistance of gastric cancer [99], and it is activated in response to chemotherapy in neuroblastoma cells that confers chemoresistance [100]. However, the proposition of inhibition or activation of this pathway is still limited in preclinical models and human tumors [98].”
- Page 8, line 329: define ACD
The definition “accidental cell death” was added. (Page 9, line 351)
- Page 9, line 360: Eif2ak3 in capital letter? It is a human gene.
Kindly, the names of the genes and proteins all over the text were double checked. They are mentioned based on the “Guidelines for Formatting Gene and Protein Names”:
Human protein: all capital, no italic (BECN1)
Mouse protein: all capital, no italic (BECN1)
Human gene: all capital, all italic (BECN1)
Mouse gene: first letter capital, all italic (Becn1)
- The figure captions of the following figures should be better explained (Figure 1, and figure 6, figure 8, etc.). Figures 6 and 8 should include the name of the compounds more than the number of them.
-Figure 1 caption was explained more: “Schematic illustration of the main distinct autophagy mechanisms in mammals (A) macroautophagy (autophagy), which include initiation, phagophore expansion, autophagosome fusion with the lysosome, developing an autolysosome, and cargo degradation; (B) microautophagy in which lysosome takes up small soluble particulates by invagination, septation, and bulging of the lysosomal membrane, and (C) CMA, which is a selective mechanism for proteins with specific amino acid motif (KFERQ sequence). The figure was created with BioRender.com.”
-All compounds images were prepared using ChemDraw Professional 15.0, and there is no need for reproduction permission. The captions of Figures 6, 8, 10, 11, 13 and 21 were improved by adding the name of each compound beside its number.
- Figure 7, 9, etc: clearly explain the meaning of the arrows and signs (-) and (+), it can lead to confusion
All schematic figures were created with BioRender.com and there is no need for reproduction permission. In all schematic figures these explanations were added:
“→ : Indicates activation or an increase in the mentioned protein expression. ---│: Indicates deactivation or a decrease in the mentioned protein expression ─ and + : Indicates the effect of the proteins on different pathways (─ means negative effect and + means positive effect)”.
- Page 17, line 557 “or glycolysis” instead of “or of glycolysis” done
The typo mistake was corrected. (Page 17, line 601)
- Figure 9: Lc3-II and BECN1 are repeated in the paclitaxel box. In the Oblongifolin C box should appear “-“ autophagy inhibition Should be correct
Thanks for the accuracy of sight. Amendment was applied on the image.
Reviewer 2 Report
This is a well written and meticulously researched review article that will be very useful for the field of cancer and autophagy researchers. The authors performed a true Sysiphus work summarizing the vast number of studies on autophagy-modulating natural compounds in the context of cancer therapy. I have only a couple of minor suggestions.
Autophagy is a rather general stress response and most, if not all cancer drugs trigger autophagy to some extent. The authors should try to better discriminate between compounds that exert their effects mainly through modulation of autophagy and those that induce autophagy rather as a bystander effect (e.g. DNA damaging agents). This is a critical point that needs to be addressed in regard to the possible exploitation of autophagy in the clinical setting.
Regarding the role of autophagy in cell survival/cell death, there is no attempt to explain/discriminate between the crosstalk between autophagy and apoptosis and true autophagic cell death, i.e. cell death executed by autophagy in the absence of other cell death modalities according to the nomenclature committee on cell death (PMID: 29362479) in the article. This is another important point and should be incorporated, perhaps in one of the figures.
Fig. 6, Fig.8, Fig. 10, Fig. 11, Fig. 21: what do they add to the article? There is no explanatory legend to these figures.
Spelling mistakes should be checked (e.g. Fig. 7, Fig. 18 “alkloids” instead of alkaloids).
Author Response
-Autophagy is a rather general stress response and most, if not all cancer drugs trigger autophagy to some extent. The authors should try to better discriminate between compounds that exert their effects mainly through modulation of autophagy and those that induce autophagy rather as a bystander effect (e.g. DNA damaging agents). This is a critical point that needs to be addressed in regard to the possible exploitation of autophagy in the clinical setting.
Thanks for the important note. Some complementary sentences were added to the text: (Page 61, 62 line 2003-2015)
“Natural compounds use different mechanisms to affect autophagy pathway and distinction between the survival-supporting and/or death-promoting roles of them on autophagy process need more deep study for therapeutic response. For example, magnolol (35) that can induce autophagy can affect the morphological and cellular events such as ATP level, cells blebbing and DNA fragmentation without leading to cell death in itself [425]. Most of the natural compounds mentioned here for clinical trials are alkaloids. As mentioned previously, alkaloids like vinblastine (1) and vincristine (3) act as autophagy inhibitors [128, 142]. Whereas, camptothecin (5), another alkaloid, was reported to induce autophagy [144]. Paclitaxel (17), an important anticancer agent, is a diterpenoid. This natural compound inhibits the progression of cervical cancer by inhibiting autophagy [167]. However, cabazitaxel (85) is a derivative of paclitaxel (17) that has been used in different phases of clinical trial. This compound was reported as an autophagy [426].”
-Regarding the role of autophagy in cell survival/cell death, there is no attempt to explain/discriminate between the crosstalk between autophagy and apoptosis and true autophagic cell death, i.e. cell death executed by autophagy in the absence of other cell death modalities according to the nomenclature committee on cell death (PMID: 29362479) in the article. This is another important point and should be incorporated, perhaps in one of the figures.
Thanks for the important note. Some complementary sentences were added to the text: (Page 8, line 327-335)
“Cancer control may not be achieved by targeting only a single cell death program. Each pathway has its own characteristics. For example, apoptosis exhibits cytoplasmic shrinkage, chromatin condensation, nuclear fragmentation, plasma membrane blebbing, and formation of apoptotic bodies; however, autophagy shows extensive cytoplasmic vacuolization and similarly culminating of phagocytic uptake and consequent lysosomal degradation [103]. UPR is a stress pathway that can be triggered by the abnormal accu-mulation of unfolded proteins in the ER caused by genetic or environmental changes, hypoxia or altered glycosylation. It makes balance in ER homeostasis by lowering the number of unfolded proteins present in the cell [104].”
-Fig. 6, Fig.8, Fig. 10, Fig. 11, Fig. 21: what do they add to the article? There is no explanatory legend to these figures. ???
The captions of Figures 6, 8, 10, 11, 13 and 21 were improved by adding the name of each compound beside its number. The images are prepared in the highest resolution. The original ones are submitted in attach.
Spelling mistakes should be checked (e.g. Fig. 7, Fig. 18 “alkloids” instead of alkaloids).
Spelling mistakes were double-checked. There is no spelling mistake in this regard.
Reviewer 3 Report
In this review, the authors presented natural plant products that were used in cancer therapy and their molecular pathways. Also, a nice and comprehensive explanation for autophagy, apoptosis, and UPR was recorded in this review. The review is interesting. I think it covers all the needed information about several plant products. Only a few points must be revised as follows:
- The illustrations' resolution must be clarified for the reader to understand.
- The authors write Becn1 and BECN1, Atg and ATG, and also, Eif2ak3 and EIF2ak3! They must use the written abbreviations uniformly.
- The text is missing the second part of the abbreviations BBC3/PUMA and PMAIP1/NOXA. The full name of MAPK8/JNK1 is also missing from the text.
- Catharanthus roseus must be italicized, as well as all the scientific plant names. Also, in vivo and in vitro as they are Latin.
- In Figure 8, the a and b letters must be written alone in the figure without repeating (Figure 8).
- In line no. 621, the authors wrote, "Treatment with 3-methyladenine (3-MA) inhibits quercetin-induced cell death," but the role of 3-MA in inhibiting autophagy hasn’t been mentioned.
- Figure 14's legend must be revised with the figure illustration.
- In line no. 1548, the spelling of protein, the n letter is missing.
- In line no. 1829, cancer spelling in the National Cancer Institute, the "C" letter is missing.
Author Response
- The illustrations' resolution must be clarified for the reader to understand.
The images are prepared in the highest resolution. The original ones are submitted in attach.
- The authors write Becn1 and BECN1, Atg and ATG, and also, Eif2ak3 and EIF2ak3! They must use the written abbreviations uniformly.
Kindly, the names of the genes and proteins all over the text were double checked. They are mentioned based on the “Guidelines for Formatting Gene and Protein Names”:
Human protein: all capital, no italic (BECN1)
Mouse protein: all capital, no italic (BECN1)
Human gene: all capital, all italic (BECN1)
Mouse gene: first letter capital, all italic (Becn1)
- The text is missing the second part of the abbreviations BBC3/PUMA and PMAIP1/NOXA. The full name of MAPK8/JNK1 is also missing from the text.
The corrections were applied in the appropriate places and also in the abbreviation list.
(Page 8, line 347), (Page 9, 348), (Page 5, line 221)
- Catharanthus roseusmust be italicized, as well as all the scientific plant names. Also, in vivo and in vitro as they are Latin.
-Catharanthus roseus was italicized. (Page 13, line 476)
-The scientific plant names were italicized all over the text. (Cases in green highlights)
-in vivo and in vitro words were italicized all over the text. (Cases in green highlights)
- In Figure 8, the a and b letters must be written alone in the figure without repeating (Figure 8).
Thanks for the accuracy of sight. Amendment was applied on the image.
- In line no. 621, the authors wrote, "Treatment with 3-methyladenine (3-MA) inhibits quercetin-induced cell death," but the role of 3-MA in inhibiting autophagy hasn’t been mentioned.
The role of 3-MA as “an autophagy inhibitor” was added. (Page 20, line 677)
- Figure 14's legend must be revised with the figure illustration.
Figure caption was explained more in detail.
- In line no. 1548, the spelling of protein, the n letter is missing.
The typo mistake was corrected. (Page 52, line 1633)
- In line no. 1829, cancer spelling in the National Cancer Institute, the "C" letter is missing.
The typo mistake was corrected. (Page 60, line 1918).
Reviewer 4 Report
1. The review is well written, It cover all the drugs and mechanism of action.
2. Please can you write few paragraph for autophagy and immunology in cancer and drug.
3. How drug is related to autophagy in hematological cancer.
4. If possible can you rewrite the title, the title is not very through in understanding.
Author Response
- The review is well written. It covers all the drugs and mechanism of action.
Thanks for the positive point of view
- Please can you write few paragraphs for autophagy and immunology in cancer and drug.
Some complementary sentences were added to the text: (Page 13, line 447-460)
“Autophagy plays role in immune cells; T-cells, B cells, macrophages, myeloid de-rived suppressor cells, and dendritic cells. It can alter tumor immunity as well as the ef-ficacy of immunotherapy [129]. The tumor microenvironment (TME), an integral complex component of cancer, can significantly influence the therapeutic response and determine tumor fate by its composition and dynamics [130]. There is a complex interaction between autophagy and TME, which modulates immunotherapy. Autophagy-mediated regulation of tumor-associated immunity may counteract or enhance the efficacy of immunotherapy. On the one hand, autophagy can promote immune response by enhancing the inhibitory role of immune cells on tumor cells and the release of cytokines, which leads to the enhanced antitumor immunotherapy responses. On the other hand, it can reduce immune response by immunosuppressive Tregs and cytokines, contributing to attenuated antitumor immunotherapy effects and accelerated tumor development. The lack of sufficient specificity of autophagy activators or inhibitors limits their clinical applications [129].”
- How drug is related to autophagy in hematological cancer.
Thanks for this comment. Throughout the text we have mentioned about the hematological and non-hematological toxicities of the mentioned compounds:
non-hematological toxicities (Page 59, line 1896)
hematological toxicity was dose-limiting (Page 60, line 1926)
hematological adverse reactions (Page 60, line 1933)
frequency and severity of the hematological toxicities (Page 60, line 1946)
non-hematological toxicities (Page 61, line 1971)
As hematological cancer is not specifically in the scope of this review paper, we decided not to make the paper long but added this sentence:
“Even, in hematologic/blood cancers, autophagy can either perform as a chemo-resistance mechanism or have tumor suppressive functions, depending on the context.” (Page 7, line 309-311)
- If possible can you rewrite the title, the title is not very through in understanding.
The title was changed:
New Visions on Natural Products and Cancer Therapy: Autophagy and Related Regulatory Pathways